# Predicting Kernel Regression Learning Curves from Only Raw Data Statistics

**Dhruva Karkada**$^{\circ,\star}$, **Joseph Turnbull**$^{\circ,\star}$, **Yuxi Liu**$^{\circ}$, **James B. Simon**$^{\circ,\text{\textcircled{89}}}$

{dkarkada,joeyturnbull,yuxi_liu,jsi}@berkeley.edu

## Abstract

We study kernel regression with common rotation-invariant kernels on real datasets including CIFAR-5m, SVHN, and ImageNet. We give a theoretical framework that predicts learning curves (test risk vs. sample size) from only two measurements: the empirical data covariance matrix and an empirical polynomial decomposition of the target function $f_*$. The key new idea is an analytical approximation of a kernel's eigenvalues and eigenfunctions with respect to an anisotropic data distribution. The eigenfunctions resemble Hermite polynomials of the data, so we call this approximation the *Hermite eigenstructure ansatz* (HEA). We prove the HEA for Gaussian data, but we find that real image data is often "Gaussian enough" for the HEA to hold well in practice, enabling us to predict learning curves by applying prior results relating kernel eigenstructure to test risk. Extending beyond kernel regression, we empirically find that MLPs in the feature-learning regime learn Hermite polynomials in the order predicted by the HEA. Our HEA framework is a proof of concept that an end-to-end theory of learning which maps dataset structure all the way to model performance is possible for nontrivial learning algorithms on real datasets.

## 1 Introduction

The quest to understand machine learning is largely motivated by a desire to predict and explain learning behavior in realistic settings. This means that, sooner or later, scientists of machine learning must develop theory that works for real datasets, somehow incorporating task structure into predictions of model performance, optimal hyperparameters, and other objects of interest. This necessity has been the elephant in the room of much of deep learning theory for some time: despite much progress in the study of neural network training and generalization, it has proven difficult to move beyond simplistic models of data and make analytical predictions applicable to real data distributions.

The central difficulty is of course the complexity of real data. There can be no full analytical description of any real data distribution, so it is difficult to see how we might develop mathematical theory that describes how such a dataset is learned. How might we hope to proceed?

One way forward may be to identify a comparatively succinct "reduced description" of a data distribution that characterizes its structure, at least insofar as a particular class of learner is concerned. We would like this reduced description to be sufficient to predict quantities of interest yet minimal enough to be a significant reduction in complexity. Ideally, we would like the theory that makes predictions from this reduced description to be mathematically simple, and we would like the description itself to give some insight into how the class of learner in question sees the data.

In this paper, we present such a reduced description of high-dimensional datasets that is suitable for describing their learning by kernel ridge regression (KRR) with rotation-invariant kernels. We find that just the data covariance matrix $\boldsymbol{\Sigma} := \mathbb{E}\big[\boldsymbol{x}\boldsymbol{x}^\top\big]$, together with a Hermite decomposition of the target function,[1] is sufficient to characterize learning by rotation-invariant kernels. We obtain this reduced description, which we term "Hermite eigenstructure," from a study of Gaussian data, but we nonetheless find it predictive for complex image datasets including CIFAR-5m, SVHN, and

---

$^{\circ}$UC Berkeley  $^{\text{\textcircled{89}}}$Imbue  $^{\star}$Joint primary authorship. Work completed during summer internship at Imbue. Code: https://github.com/JoeyTurn/hermite-eigenstructure-ansatz

[1]See Section 3.2 and Appendix A for a review of Hermite polynomials.

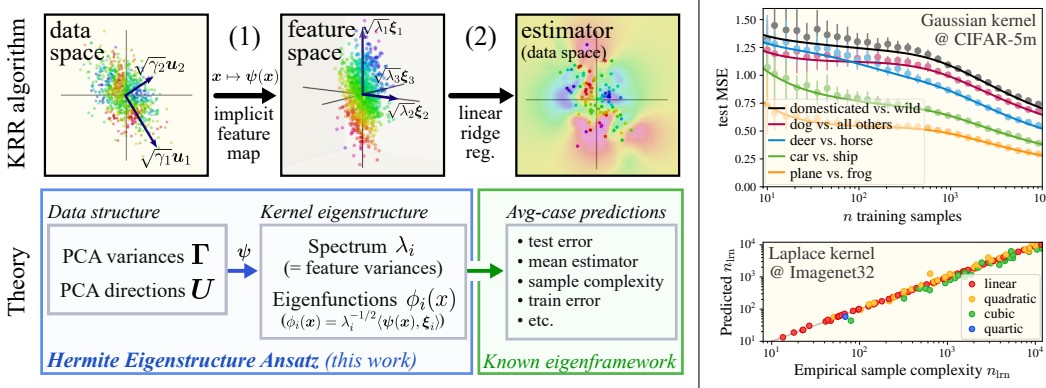

Figure 1: **We provide an end-to-end theory of learning for kernel ridge regression (KRR) that maps minimal statistics of the data distribution to test-time performance. (Top left.)** KRR implicitly consists of two steps: (1) the kernel maps the data to high-dimensional nonlinear features $x \mapsto \psi(x)$, then (2) it fits a linear estimator to these features. Let the covariance in data space be $\mathbb{E}[xx^\top] = U\Gamma U^\top$ and let the covariance in feature space be $\mathbb{E}[\psi(x)\psi(x)^\top] = \Xi\Lambda\Xi^\top$. **(Lower left.)** We introduce an ansatz (blue box) that predicts the *feature* covariance statistics $(\Xi, \Lambda)$ from only the *data* covariance $(U, \Gamma)$ and the functional form of $\psi(\cdot)$. This is sufficient to predict average-case test error using known results (green box). **(Top right.)** We are able to predict learning curves for KRR on image tasks without requiring omniscient knowledge of the feature statistics (i.e., without ever constructing or diagonalizing a kernel matrix). **(Bottom right.)** We are able to accurately predict the KRR sample complexity, including constant prefactors, for learning polynomials of the ImageNet dataset. See Figure 3 for additional plots and Appendix D for experimental details.

ImageNet. From just the covariance matrix, we can predict kernel eigenstructure and learning curves for synthetic functions. Using the true labels to estimate the target function's Hermite decomposition, we are additionally able to predict KRR learning curves on real tasks. Unlike previous approaches for predicting learning curves, this method does not require numerically constructing or diagonalizing a kernel matrix to find the kernel eigensystem.

Our approach relies on recent results in the theory of KRR which assert that knowledge of the kernel's eigenstructure with respect to a data measure is sufficient to predict learning behavior (Sollich (2001); Bordelon et al. (2020); Jacot et al. (2020); Simon et al. (2021)). These works provide a set of equations that map this eigenstructure to predictions of test-time error. Our central observation is that, despite the complexity of high-dimensional datasets and the great variety of rotation-invariant kernels, this kernel eigenstructure is often very close to a simple analytical form expressible in terms of Hermite polynomials in the original data space. We term this claim the "Hermite eigenstructure ansatz," and we identify a (broad) set of conditions under which it empirically holds.

Concretely, our contributions are as follows:

- We propose the *Hermite eigenstructure ansatz* (Section 4), a closed-form expression for the eigensystem of rotation-invariant kernels on real datasets. We find empirically that it holds to an excellent approximation for real image datasets (Figure 2).

- We prove that the HEA holds in the case of Gaussian data for two limiting cases of the kernel function (Theorems 1 and 2).

- We use the HEA to predict KRR learning curves on CIFAR-5m, SVHN, and ImageNet from only the sample covariance and a Hermite decomposition of the target function (Figures 1 and 3).

- We empirically find that MLPs in the feature-learning regime learn Hermite polynomials of CIFAR-5m in the same order as the HEA predicts for KRR (Figure 4).

## 2 RESEARCH CONTEXT AND RELATED WORKS

**Kernel models as proxies for neural networks.** Our motivation for studying KRR comes from the *"neural tangent kernel" (NTK)* line of work, which finds suitably parametrized infinite-width networks are equivalent to KRR, and that for MLPs, the kernel function is rotation-invariant (Neal, 1996; Lee et al., 2018; Jacot et al., 2018). Kernel methods have proven useful as models of network dynamics and optimization (Chizat et al., 2019; Du et al., 2019; Bordelon & Pehlevan, 2021).

**Learning curves for KRR.** Motivated by the NTK, many recent works have converged on a set of equations which predict KRR's test-time error from the *kernel and task eigenstructure* (Sollich, 2001; Bordelon et al., 2020; Jacot et al., 2020; Simon et al., 2021). This "KRR eigenframework" depends on the kernel's eigenvalues and eigenfunctions with respect to the data distribution. Our main result is an approximate *analytical* expression for these eigenvalues and eigenfunctions, permitting the inductive bias of KRR to be studied directly in the data space. Appendix C reviews this eigenframework.

**Exactly-solved cases of kernel eigenstructure.** Exact kernel diagonalizations with machine-learning-relevant kernels are known in many highly-symmetric settings, including stationary kernels on the torus $\mathbb{T}^d$ and rotation-invariant kernels on the sphere $\mathbb{S}^d$ (Mei & Montanari, 2019). Moving to anisotropic domains, Ghorbani et al. (2020) gave the eigenstructure of rotation-invariant kernels when the measure is a "product of spheres" of different radii. The case of a Gaussian kernel on a Gaussian measure was solved exactly by Zhu et al. (1997). Our Hermite eigenstructure ansatz is consistent with all these results and unifies them in a limiting case.

**Eigenstructure of dot-product kernels on anisotropic Gaussian data.** Concurrent work by Wortsman & Loureiro (2025) takes up the same mathematical problem as us: finding the eigenstructure of dot-product kernels on anisotropic Gaussian data. They prove upper and lower bounds on kernel eigenvalues that agree with the eigenvalues we propose in our Equation (5). When studying the generalization of KRR, they switch from dot-product kernels to Hermite polynomial kernels. Our work provides a justification for this replacement: our HEA states that this swap changes the eigensystem only slightly.

**Modeling data with a Gaussian measure.** A developing body of literature argues that MLPs' learning of complex data distributions is similar to the behavior one would see if the data were Gaussian with the same covariance (Goldt et al., 2020; Refinetti et al., 2023). We broadly adopt this lens in the study of KRR and find that, indeed, the data is well-modeled as Gaussian. We note that this is distinct from the "Gaussian universality" assumption typically made in the derivation of the KRR eigenframework. See Appendix F for a discussion of the difference.

**Single- and multi-index models.** Much recent literature has studied MLPs' learning of *single- and multi-index functions* which depend only on a rank-one or rank-$k$ projection of the input $\boldsymbol{x}$ (Dudeja & Hsu, 2018; Bietti et al., 2022; Dandi et al., 2023; Lee et al., 2024; Mousavi-Hosseini et al., 2024). This work partially motivated our study, and the multidimensional Hermite basis we use in this work is a basis of multi-index functions. Prior work in this vein has found that higher-order Hermite polynomials require more samples or gradient steps to learn. Two ways in which we depart from this body of work are that (a) we seek to predict the *value* of the test error (including constant prefactors), not just asymptotics or scaling laws, and (b) we study *anisotropic* data, which allows application of our results to real datasets.

**Analytical models of data.** Several recent works have proposed theoretical models for the hierarchical structure in image and text data with the aim of understanding neural network performance on such datasets (Cagnetta et al., 2024; Sclocchi et al., 2025; Cagnetta & Wyart, 2024). Our work in this paper is undertaken in a similar spirit.

## 3 PRELIMINARIES

We will work in a standard supervised setting: our dataset consists of $n$ samples $\mathcal{X} = \{\boldsymbol{x}_i\}_{i=1}^{n}$ drawn i.i.d. from a measure $\mu$ over $\mathbb{R}^d$, and we wish to learn a target function $f_*$ from noisy training labels $\boldsymbol{y} = \{y_i\}_{i=1}^{n}$ where $y_i = f_*(\boldsymbol{x}_i) + \mathcal{N}(0, \epsilon^2)$ with noise level $\epsilon \geq 0$. We will assume with minimal loss of generality that $\mu$ has mean zero: $\mathbb{E}_{\boldsymbol{x} \sim \mu}[\boldsymbol{x}] = \boldsymbol{0}$. Once a learning rule returns a predicted function $\hat{f}$, we evaluate its test mean-squared error $\text{MSE}_{\text{te}} = \mathbb{E}_{\boldsymbol{x} \sim \mu}\left[(f_*(\boldsymbol{x}) - \hat{f}(\boldsymbol{x}))^2\right] + \epsilon^2$. We write $\langle g, h \rangle_\mu := \mathbb{E}_{\boldsymbol{x} \sim \mu}[g(\boldsymbol{x})h(\boldsymbol{x})]$ and $\|g\|_\mu^2 := \langle g, g \rangle_\mu$ for the $L^2$ inner product and norm with respect to

$\mu$. We write $(a_i)_{i \in \mathcal{I}}$ to denote an ordered sequence with index set $\mathcal{I}$, and we write only $(a_i)$ when the index set is clear from context.

## 3.1 KERNEL REGRESSION AND KERNEL EIGENSYSTEMS

KRR is a learning rule specified by a positive-semidefinite "kernel function" $K : \mathbb{R}^d \times \mathbb{R}^d \to \mathbb{R}$ and a ridge parameter $\delta \geq 0$. Given a dataset $(\mathcal{X}, \boldsymbol{y})$, KRR returns the predicted function

$$\hat{f}(\boldsymbol{x}) = \boldsymbol{k}_{\boldsymbol{x}\mathcal{X}}(\boldsymbol{K}_{\mathcal{X}\mathcal{X}} + \delta \boldsymbol{I}_n)^{-1}\boldsymbol{y}, \tag{1}$$

where the vector $[\boldsymbol{k}_{\boldsymbol{x}\mathcal{X}}]_i = K(\boldsymbol{x}, \boldsymbol{x}_i)$ and matrix $[\boldsymbol{K}_{\mathcal{X}\mathcal{X}}]_{ij} = K(\boldsymbol{x}_i, \boldsymbol{x}_j)$ contain evaluations of the kernel function. In this paper, we will restrict our attention to two special classes of kernel:

**Definition 1** (Rotation-invariant kernel). A kernel function is *rotation-invariant* if it takes the form $K(\boldsymbol{x}, \boldsymbol{x}') = K(\|\boldsymbol{x}\|, \|\boldsymbol{x}'\|, \boldsymbol{x}^\top \boldsymbol{x}')$.

Such a kernel $K$ is called "rotation-invariant" because $K(\boldsymbol{U}\boldsymbol{x}, \boldsymbol{U}\boldsymbol{x}') = K(\boldsymbol{x}, \boldsymbol{x}')$ for any orthonormal matrix $\boldsymbol{U}$. Many widely-used kernels are rotation-invariant, including the Gaussian kernel $K(\boldsymbol{x}, \boldsymbol{x}') = e^{-\frac{1}{2\sigma^2}\|\boldsymbol{x}-\boldsymbol{x}'\|^2}$, the Laplace kernel $K(\boldsymbol{x}, \boldsymbol{x}') = e^{-\frac{1}{\sigma}\|\boldsymbol{x}-\boldsymbol{x}'\|}$, and the Neural Network Gaussian Process (NNGP) kernels and NTKs of infinite-width MLPs. We will be particularly interested in a subset of rotation-invariant kernels which discard the explicit radial dependence:

**Definition 2** (Dot-product kernel). A kernel function is a *dot-product kernel* if it takes the form $K(\boldsymbol{x}, \boldsymbol{x}') = K(\boldsymbol{x}^\top \boldsymbol{x}')$.

For a dot-product kernel to be positive-semidefinite on all domains, it must admit a Taylor series $K(\boldsymbol{x}^\top \boldsymbol{x}') = \sum_{\ell \geq 0} \frac{c_\ell}{\ell!}(\boldsymbol{x}^\top \boldsymbol{x}')^\ell$ with nonnegative *level coefficients* $c_\ell \geq 0$ (Schoenberg, 1942). We will find it useful to describe dot-product kernels in terms of their level coefficients $(c_\ell)_{\ell \geq 0}$.

We would like to study arbitrary rotation-invariant kernels, but it is easier to study dot-product kernels, which admit the above series expansion. Fortunately, a rotation-invariant kernel is a dot product kernel when the domain is restricted to a sphere, and if we know that our data has typical norm $r$, we may approximate the rotation-invariant kernel as the dot-product kernel which matches on $r\mathbb{S}^{d-1}$:

**Definition 3** (On-sphere level coefficients). The *on-sphere level coefficients* of a rotation-invariant kernel $K$ at a radius $r > 0$ are the nonnegative sequence $\mathsf{coeffs}(K, r) := (c_\ell)_{\ell \geq 0}$ such that

$$K(\boldsymbol{x}, \boldsymbol{x}') = \sum_{\ell \geq 0} \frac{c_\ell}{\ell!}(\boldsymbol{x}^\top \boldsymbol{x}')^\ell \quad \text{for all } \boldsymbol{x}, \boldsymbol{x}' \text{ such that } \|\boldsymbol{x}\| = \|\boldsymbol{x}'\| = r. \tag{2}$$

We give the on-sphere level coefficients for various kernels in Appendix B.

An *eigenfunction* of a kernel $K$ with respect to a measure $\mu$ is a function $\phi$ such that $\mathbb{E}_{\boldsymbol{x}' \sim \mu}[K(\boldsymbol{x}, \boldsymbol{x}')\phi(\boldsymbol{x}')] = \lambda \phi(\boldsymbol{x})$ for some $\lambda \geq 0$. By Mercer's Theorem (Mohri et al., 2018, Theorem 6.2), any compact kernel admits a complete basis of orthonormal eigenfunctions with $\langle \phi_i, \phi_j \rangle_\mu = \delta_{ij}$ and may be spectrally decomposed[2] as $K(\boldsymbol{x}, \boldsymbol{x}') = \sum_i \lambda_i \phi_i(\boldsymbol{x})\phi_i(\boldsymbol{x}')$. We will write $\mathsf{eigensystem}(\mu, K) = (\lambda_i, \phi_i)_{i=1}^\infty$ to denote the sequence of all eigenpairs, indexed in decreasing eigenvalue order ($\lambda_i \geq \lambda_{i+1}$) unless otherwise specified. It will prove useful to decompose the target function in the kernel eigenbasis as $f_*(\boldsymbol{x}) = \sum_i v_i \phi_i(\boldsymbol{x})$, where $(v_i)$ are eigencoefficients.

## 3.2 HERMITE POLYNOMIALS AS A NATURAL BASIS FOR GAUSSIAN DATA

Throughout, we write $(h_k)_{k \geq 0}$ for the normalized probabilist's Hermite polynomials. These are the orthogonal polynomials for the standard Gaussian measure, satisfying $\mathbb{E}_{x \sim \mathcal{N}(0,1)}[h_k(x)h_m(x)] = \delta_{km}$. The first few such polynomials are $h_0(x) = 1$, $h_1(x) = x$, $h_2(x) = \frac{1}{\sqrt{2}}(x^2 - 1)$. See Appendix A for a review of Hermite polynomials.

---

[2]Here is an intuitive description of a kernel eigensystem which is shown visually in Figure 1. Any kernel function $K$ may be viewed as an inner product in a high-dimensional feature space: $K(\boldsymbol{x}, \boldsymbol{x}') = \langle \boldsymbol{\psi}(\boldsymbol{x}), \boldsymbol{\psi}(\boldsymbol{x}') \rangle$. Consider mapping the dataset into this high-dimensional space and then computing the principal components of $\boldsymbol{\Sigma}_{\boldsymbol{\psi}} := \mathbb{E}_{\boldsymbol{x}}[\boldsymbol{\psi}(\boldsymbol{x})\boldsymbol{\psi}^\top(\boldsymbol{x})] = \boldsymbol{\Xi}\boldsymbol{\Lambda}\boldsymbol{\Xi}^\top$. Each eigenvalue $\lambda_i$ is a kernel eigenvalue. The corresponding eigenfunction is a projection onto the $i$-th principal direction: $\phi_i(\boldsymbol{x}) = \lambda_i^{-1/2}\langle \boldsymbol{\psi}(\boldsymbol{x}), \boldsymbol{\xi}_i \rangle$.

We can use these 1D Hermite polynomials to construct an orthonormal basis for a multivariate Gaussian measure $\boldsymbol{x} \sim \mathcal{N}(0, \boldsymbol{\Sigma})$ with positive-definite covariance $\boldsymbol{\Sigma} \succ \boldsymbol{0}$. First we diagonalize the covariance as $\boldsymbol{\Sigma} = \boldsymbol{U}\boldsymbol{\Gamma}\boldsymbol{U}^\top$ with orthogonal matrix $\boldsymbol{U} = [\boldsymbol{u}_1 \cdots \boldsymbol{u}_d]$ and diagonal matrix $\boldsymbol{\Gamma} = \operatorname{diag}(\gamma_1, \ldots, \gamma_d)$. Then for any multi-index $\boldsymbol{\alpha} \in \mathbb{N}_0^d$, we define

$$h_{\boldsymbol{\alpha}}^{(\boldsymbol{\Sigma})}(\boldsymbol{x}) := \prod_{i=1}^d h_{\alpha_i}(z_i), \qquad \text{where} \qquad \boldsymbol{z} = \boldsymbol{\Gamma}^{-1/2}\boldsymbol{U}^\top \boldsymbol{x}. \tag{3}$$

In Equation (3), the elements of $\boldsymbol{\alpha}$ specify the order of the Hermite polynomial along each principal direction of data covariance. These multidimensional Hermite polynomials are orthonormal, satisfying $\mathbb{E}_{\boldsymbol{x} \sim \mathcal{N}(0, \boldsymbol{\Sigma})}\left[h_{\boldsymbol{\alpha}}^{(\boldsymbol{\Sigma})}(\boldsymbol{x}) h_{\boldsymbol{\alpha}'}^{(\boldsymbol{\Sigma})}(\boldsymbol{x})\right] = \delta_{\boldsymbol{\alpha}\boldsymbol{\alpha}'}$. In the next section, we assert that this naïve guess of basis is in fact often close to the *true basis of kernel eigenfunctions* for synthetic and real datasets.

## 4 THE HERMITE EIGENSTRUCTURE ANSATZ: THEORY AND EXPERIMENT

Prior work has shown that predicting kernel regression learning curves boils down to understanding the kernel eigensystem. With this as motivation, we are ready to introduce our primary mathematical object: an *explicit functional form* for the kernel eigensystem, suitable for rotation-invariant kernels and high-dimensional datasets.

**Plan of attack.** First we will write down this explicit functional form (Definition 4). Then we will state our assertion that this functional form approximates the true kernel eigensystem (HEA). Next, we will demonstrate that the HEA holds to an excellent approximation for several rotation-invariant kernels on several real image datasets (Figure 2). We will then give an intuitive justification for the HEA and give two formal theorems which state that the HEA holds for Gaussian data as kernel width grows (Theorems 1 and 2). Next, we will characterize the factors that make the HEA work better or worse (Section 4.2). Finally, we will use the HEA to predict KRR learning curves in Section 5. We begin by explicitly defining our Hermite eigensystem:

---

**Definition 4** (Hermite eigensystem). Given a data covariance matrix $\boldsymbol{\Sigma} = \boldsymbol{U}\boldsymbol{\Gamma}\boldsymbol{U}^\top$ and a sequence of level coefficients $(c_\ell)$, we define the $(\boldsymbol{\Sigma}, (c_\ell))-$*Hermite eigensystem* to be the set of (scalar, function) pairs

$$\mathcal{HE}(\boldsymbol{\Sigma}, (c_\ell)) = \{(\lambda_{\boldsymbol{\alpha}}, \phi_{\boldsymbol{\alpha}}) \text{ for all } \boldsymbol{\alpha} \in \mathbb{N}_0^d\} \tag{4}$$

where for each multi-index $\boldsymbol{\alpha}$ the proposed eigenvalue and eigenfunction are constructed as

$$\lambda_{\boldsymbol{\alpha}} = c_{|\boldsymbol{\alpha}|} \cdot \prod_{i=1}^d \gamma_i^{\alpha_i} \quad \text{and} \quad \phi_{\boldsymbol{\alpha}} = h_{\boldsymbol{\alpha}}^{(\boldsymbol{\Sigma})}, \tag{5}$$

where $|\boldsymbol{\alpha}| = \sum_i \alpha_i$ and $h_{\boldsymbol{\alpha}}^{(\boldsymbol{\Sigma})}$ is the multivariate Hermite polynomial given in Equation (3).

---

The $(\boldsymbol{\Sigma}, (c_\ell))-$Hermite eigensystem is a set of Hermite polynomials $\phi_{\boldsymbol{\alpha}}$ and associated positive scalars $\lambda_{\boldsymbol{\alpha}}$, one for each multi-index $\boldsymbol{\alpha} \in \mathbb{N}_0^d$. The eigenvalues $(\lambda_{\boldsymbol{\alpha}})$ are monomials in the data covariance eigenvalues $(\gamma_i)$, rescaled by the appropriate level coefficient $c_{|\boldsymbol{\alpha}|}$. We now present the Hermite eigenstructure ansatz, which asserts that this set of (scalar, function) pairs is in fact a close match to the true kernel eigensystem.

---

Let $K$ be a rotation-invariant kernel and let $\mu$ be a measure over $\mathbb{R}^d$ with zero mean. Then let:

- $\boldsymbol{\Sigma} = \mathbb{E}_{\boldsymbol{x} \sim \mu}[\boldsymbol{x}\boldsymbol{x}^\top]$ be the data covariance matrix,
- $r = \operatorname{Tr}[\boldsymbol{\Sigma}]^{\frac{1}{2}}$ be the root-mean-squared data norm, and
- $(c_\ell) = \operatorname{coeffs}(K, r)$ be the level coefficients of $K$ restricted to the sphere $r\mathbb{S}^{d-1}$.

The ***Hermite eigenstructure ansatz*** asserts that

$$\operatorname{eigensystem}(\mu, K) \approx \mathcal{HE}(\boldsymbol{\Sigma}, (c_\ell)). \tag{HEA}$$

That is, the true kernel eigensystem is approximately equal to the $(\boldsymbol{\Sigma}, (c_\ell))$-Hermite eigensystem given in Definition 4.

---

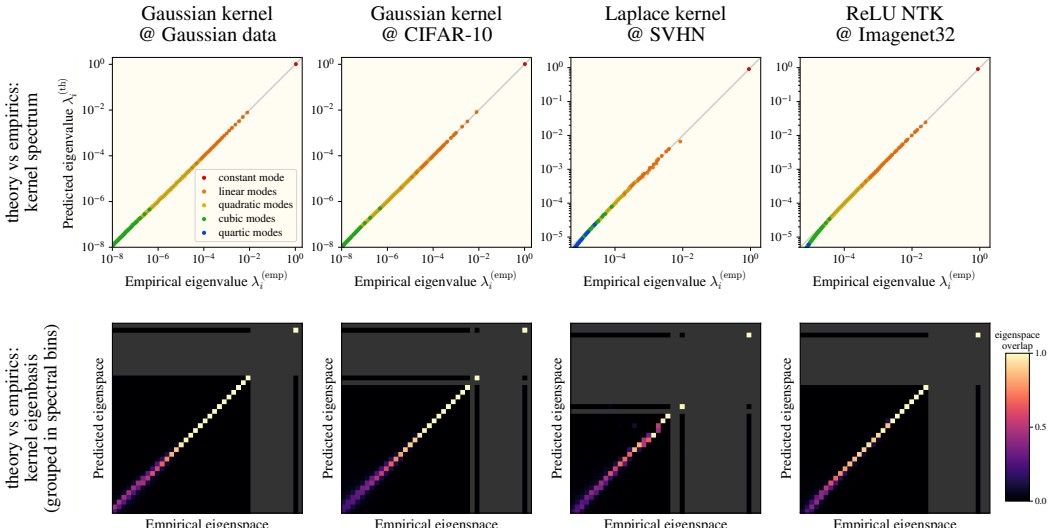

Figure 2: **The Hermite eigenstructure ansatz (HEA) accurately predicts the eigenvalues (top) and eigenfunctions (bottom) of various kernel/dataset combinations.** For four kernel/dataset settings (columns), we compute the empirical kernel eigensystem $\{(\lambda_i^{(\text{emp})}, \phi_i^{(\text{emp})})\}$ and compare to the theoretical eigenpairs $\{(\lambda_i^{(\text{th})}, \phi_i^{(\text{th})})\}$ obtained from Definition 4 and indexed in order of decreasing $\lambda_i^{(\text{th})}$. In the top plot in each column, the $i$-th point from the top right has coordinates $(\lambda_i^{(\text{emp})}, \lambda_i^{(\text{th})})$ and its color indicates the polynomial degree of $\phi_i^{(\text{th})}$. In the bottom plot in each column, we bin both the predicted and empirical eigenfunctions into logarithmic spectral bins and visualize the pairwise subspace overlap (Equation (44)), with axes matching the top plot. Grey pixels indicate bins with no eigenvalues. In all plots, concentration along the diagonal indicates theory-experiment match. See Appendix D.2 for further explanation and experimental details.

The HEA is a strong claim: it asserts that the kernel eigensystem, to a good approximation, has a simple analytical form which depends only on the second-order statistics of $\mu$ and no higher moments and that the kernel eigenfunctions are multivariate Hermite polynomials *independent* of the kernel chosen (so long as it is rotation-invariant).

*Rather than a provable fact, the HEA should be treated as a falsifiable claim that may hold well or poorly in any given setting.* In practice, with rotation-invariant kernels and high-dimensional image datasets, we will find that it often holds quite well. Figure 2 examines four such settings, finding that in each, both the kernel spectrum and eigenfunctions are well-predicted by the HEA.

Before moving on, we make a note about the meaning of the symbol "$\approx$" in the HEA. Equality would mean that the $i$-th largest eigenvalues of the true eigensystem$(\mu, K)$ and the synthetic Hermite eigensystem $\mathcal{HE}(\mathbf{\Sigma}, (c_\ell))$ are equal, and the corresponding eigenfunctions are also equal (up to eigenspace degeneracy). Intuitively, approximate agreement means that the eigenvalues approximately match, and the $i$-th eigenfunction from one set has significant overlap only with eigenfunctions from the other set with eigenvalues close to $\lambda_i$. This fuzzier notion of alignment is enough to ensure that learning curves computed using the two eigensystems are a good match, which is the practical outcome of interest. It is this notion of eigenspace alignment that we test for empirically in Figure 2.

## 4.1 THE HEA FOR GAUSSIAN DATA: SOME INTUITION AND TWO THEOREMS

When might we expect the HEA to hold? To gain the central intuition, it is sufficient to consider a simple case of univariate Gaussian data $x \sim \mu = \mathcal{N}(0, \gamma)$ and the Gaussian kernel $K_\sigma(x, x') = e^{\frac{-1}{2\sigma^2}(x-x')^2}$. This kernel admits the feature map $K_\sigma(x, x') = \langle \boldsymbol{\psi}_\sigma(x), \boldsymbol{\psi}_\sigma(x') \rangle$ where

$$\boldsymbol{\psi}_\sigma(x) = e^{\frac{-x^2}{2\sigma^2}} \cdot \begin{bmatrix} 1 & \dfrac{x}{\sigma} & \dfrac{x^2}{\sqrt{2}\sigma^2} & \cdots & \dfrac{x^\ell}{\sqrt{\ell!}\sigma^\ell} & \cdots \end{bmatrix}, \tag{6}$$

We would like to find the directions of principal covariance of $\boldsymbol{\psi}_\sigma(x)$. Let us suppose that $\sigma^2 \gg \gamma$: the kernel width dominates the width of the data distribution. Examining Equation (6), we can make two observations. First, the exponential prefactor will be close to one, and we may approximate $\boldsymbol{\psi}_\sigma$ componentwise as $[\boldsymbol{\psi}_\sigma(\boldsymbol{x})]_\ell \approx \frac{x^\ell}{\sqrt{\ell!}\sigma^\ell}$. This amounts to approximating our kernel as $K_\sigma(x, x') = \sum_\ell \frac{(xx')^\ell}{\sigma^{2\ell} \cdot \ell!}$ — that is, as a dot-product kernel with coefficients $c_\ell = \sigma^{-2\ell}$. Second, each component of $\boldsymbol{\psi}_\sigma$ will dominate all subsequent components:

$$\mathbb{E}_x\big[[\boldsymbol{\psi}_\sigma(\boldsymbol{x})]_\ell^2\big] \propto \sigma^{-2\ell}\gamma^\ell \quad \gg \quad \mathbb{E}_x\big[[\boldsymbol{\psi}_\sigma(\boldsymbol{x})]_{\ell+1}^2\big] \propto \sigma^{-2(\ell+1)}\gamma^{\ell+1}. \tag{7}$$

Since the first element of $\boldsymbol{\psi}_\sigma$ is by far the largest (and since we do not center $\boldsymbol{\psi}_\sigma$ before computing eigendirections), the first direction of principal variation will correspond to $\phi_0(x) \approx 1$, with variance $\lambda_0 \approx 1$.[3] The next direction will correspond to $\phi_1(x) \approx \gamma^{-1/2}x$ with variance $\lambda_1 \approx \sigma^{-2}\gamma$. The next eigenfunction must incorporate the $x^2$ direction, but we have a problem: $x^2$ is not orthogonal to $\phi_0(x) = 1$. We must therefore orthogonalize it with respect to $\phi_0$ against our measure $\mu$ in the usual Gram-Schmidt fashion. This yields $\phi_2(x) \approx \frac{1}{\sqrt{2}}(\gamma^{-1}x^2 - 1)$, which using standard formulas for Gaussian integrals gives an eigenvalue

$$\lambda_2 = \int K(x, x')\phi_2(x)\phi_2(x')d\mu(x)d\mu(x') \approx \sigma^{-4}\gamma^2. \tag{8}$$

Continuing this process to higher orders, we find that the kernel eigensystem matches that predicted by the HEA: $\phi_\ell$ is the $\ell$-th orthogonal polynomial with respect to our Gaussian measure — that is, the Hermite polynomial $h_\ell(\gamma^{-1/2}x)$ — and the $\ell$-th eigenvalue is $\lambda_\ell \approx \sigma^{-2\ell}\gamma^\ell = c_\ell\gamma^\ell$. The corrections hidden by every "$\approx$" in this derivation are of relative size $O(\sigma^{-2}\gamma)$ and thus vanish as $\sigma$ grows.[4]

This same analysis holds for any dot-product kernel $K(x, x') = \sum_\ell \frac{c_\ell}{\ell!}(xx')^\ell$ with level coefficients $(c_\ell)$ such that $\frac{c_{\ell+1}\gamma}{c_\ell} \ll 1$. It can, with some difficulty, be further extended to apply to multivariate Gaussian data $\boldsymbol{x} \sim \mathcal{N}(0, \boldsymbol{\Sigma})$. But what if the kernel is not a dot-product kernel, such as the Laplace kernel $K(\boldsymbol{x}, \boldsymbol{x}') = e^{\frac{1}{\sigma}\|\boldsymbol{x}-\boldsymbol{x}'\|}$? Unlike the Gaussian kernel, the Laplace kernel is not well-approximated by a dot-product kernel even at large width because of its nonanalyticity at zero. However, like all rotation-invariant kernels, the Laplace kernel *is* a dot-product kernel when restricted to a sphere (and will be close to a dot-product kernel when restricted to a spherical shell whose thickness is not too big). In such a case, we will require the data to be high-dimensional: for data with a high effective dimension $d_{\text{eff}} := \frac{\text{Tr}[\boldsymbol{\Sigma}]^2}{\text{Tr}[\boldsymbol{\Sigma}^2]} \gg 1$, samples will tend to concentrate in norm.[5] We may thus safely approximate *any* rotation-invariant $K$ as a dot-product kernel.

Having given an intuitive derivation of the HEA for Gaussian data, we now move to formal statements. Our first theorem states that the HEA holds for the Gaussian kernel at large width.

---

**Theorem 1** (The HEA holds for a wide Gaussian kernel on a Gaussian measure)**.**
*Let $\mu = \mathcal{N}(0, \boldsymbol{\Sigma})$ be a multivariate Gaussian measure and let $K_\sigma(\boldsymbol{x}, \boldsymbol{x}') = e^{-\frac{1}{2\sigma^2}\|\boldsymbol{x}-\boldsymbol{x}'\|^2}$ be the Gaussian kernel with width $\sigma$. Let $r = \text{Tr}[\boldsymbol{\Sigma}]^{1/2}$ and let $(c_\ell) = \text{coeffs}(K_\sigma, r)$, which yields $c_\ell = \sigma^{-2\ell}e^{-\frac{r^2}{2\sigma^2}}$. Then:*

$$\text{as } \sigma \to \infty, \quad \text{eigensystem}(\mu, K_\sigma) \to \mathcal{HE}(\boldsymbol{\Sigma}, (c_\ell)).$$

---

*Proof sketch (full proof in Appendix I).* Mehler's formula can be used to express the Gaussian kernel's eigensystem exactly (Mehler, 1866). Taking $\sigma \to \infty$ in the resulting expressions yields agreement with the HEA.

Our second theorem applies to dot-product kernels with fast-decaying level coefficients.

---

[3]We index eigenmodes from 0 instead of 1 here to match the polynomial order $\ell$.

[4]Were we to repeat this calculation with a different measure $\mu$ for $x$, we would obtain the orthogonal polynomials with respect to $\mu$ as eigenfunctions. For example, if $\mu = U[-1, 1]$, we get the Legendre polynomials.

[5]For Gaussian data $\boldsymbol{x} \sim \mathcal{N}(0, \boldsymbol{\Sigma})$, the relative variance of the norm is $\text{Var}\big[\|\boldsymbol{x}\|^2\big] / \mathbb{E}\big[\|\boldsymbol{x}\|^2\big]^2 = 2/d_{\text{eff}}$, which falls to zero as $d_{\text{eff}}$ grows.

---

**Theorem 2** (The HEA holds for a fast-decaying dot-product kernel on a Gaussian measure).
*Let $\mu = \mathcal{N}(0, \boldsymbol{\Sigma})$ be a multivariate Gaussian measure with variance $\boldsymbol{\Sigma} \succ \mathbf{0}$ and let*

$$K_{(c_\ell)}(\boldsymbol{x}, \boldsymbol{x}') = \sum_{\ell=0}^{\infty} \frac{c_\ell}{\ell!} (\boldsymbol{x}^\top \boldsymbol{x}')^\ell$$

*be a dot-product kernel with coefficients $c_\ell \geq 0$ such that $c_{\ell+1} \leq \epsilon \cdot c_\ell$ for some $\epsilon > 0$. Then:*

*as $\epsilon \to 0$, $\mathsf{eigensystem}(\mu, K_{(c_\ell)}) \to \mathcal{HE}(\boldsymbol{\Sigma}, (c_\ell))$ linearly in $\epsilon$.*

---

*Proof sketch:* Our proof formalizes the intuitive "Gram-Schmidt" derivation of the HEA given above. We use perturbation theory to show that the kernel eigenstructure splits into exponentially-separated segments, with the $\ell$-th segment eigenstructure determined almost fully by the $\ell$-th order term of $K_{(c_\ell)}$. Due to the complexity of the proof, we break it up into stages: we rigorously state and prove the one-dimensional case in Appendix J, then state and prove the general case in Appendix K.

What are the meanings of the "wide kernel" and "fast decay" limits in Theorems 1 and 2? Instead of taking a wider (or faster-decaying) kernel on fixed data, one might equivalently fix the kernel and scale down the data as $\boldsymbol{x} \mapsto \epsilon \boldsymbol{x}$ with $\epsilon \to 0$, and these theorems will still hold.[6] These are thus the limits in which the effective length scale of the kernel dominates that of the data.

An astute observer might note that in the limits taken in Theorems 1 and 2, the kernel converges to a constant: $K(\boldsymbol{x}, \boldsymbol{x}') \propto 1$. Are these theorems then trivial? No: even though the zeroth-order eigenvalue dominates all subsequent eigenvalues, the *relative* size of these smaller eigenvalues is nontrivial (and determines the inductive bias of KRR so long as one has $n > 1$ samples). These theorems state that, in these limits, the HEA gives the relative sizes of *all* eigenvalues, the constant mode and all subsequent smaller eigenvalues.

### 4.2 Conditions for success: fast decay of $c_\ell$, high data dimension, and a "Gaussian enough" data distribution

The intuitive theory above suggests three conditions required for the HEA to hold reasonably well. Here we list these conditions and give empirical confirmation that breaking any one usually causes the HEA to fail.

**1) Fast decay of level coefficients.** As discussed in Section 4.1, we need $c_\ell \gg \gamma_1 c_{\ell+1}$ for the Gram-Schmidt process underlying the HEA to work. In Figure 13, we show that as we decrease the Gaussian kernel's width (and thus increase $\frac{c_{\ell+1}}{c_\ell}$) on a fixed dataset, the HEA eventually breaks.

**2) High data dimension (for some kernels).** As previously discussed, concentration of norm (via high $d_{\text{eff}}$) is required if we are to approximate an arbitrary rotation-invariant kernel as a dot-product kernel. In Figure 14, we show that for the Laplace kernel and ReLU NTK, agreement with the HEA worsens as $d_{\text{eff}}$ decreases. However, since the Gaussian kernel is smooth at $\boldsymbol{x} = \boldsymbol{x}'$, it does not require concentration of norm, and low $d_{\text{eff}}$ is fine (Figure 15).

**3) "Gaussian enough" data distribution.** Common image datasets are complex enough to roughly satisfy simple tests of Gaussianity, such as coordinatewise Gaussian marginals. As we make the dataset simpler (CIFAR → SVHN → MNIST → tabular), these marginals become less Gaussian, and HEA agreement degrades (Figures 16 and 17). It is noteworthy that our theory empirically works *better on more complex datasets* thanks to the blessings of dimensionality.

## 5 The HEA Allows Prediction of KRR Learning Curves

Under the conditions outlined in the previous section, we expect the HEA to accurately predict kernel eigenstructure. We aim to plug these results directly into the aforementioned KRR eigenframework (of e.g. Simon et al. (2021)) to predict the final test risk of KRR. However, a key challenge remains: using the eigenframework requires knowing the coefficients of the target function in the kernel

---

[6]This is essentially the physicist's duality between "scaling down your system" and "scaling up your measuring stick."

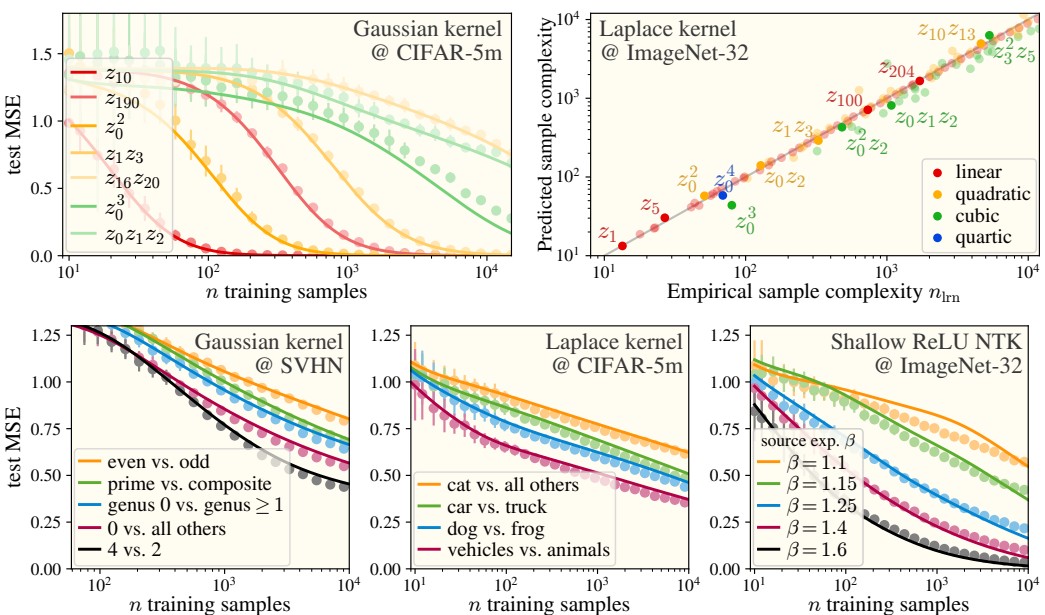

Figure 3: **Using only the empirical data covariance and a polynomial decomposition of the target function, we predict learning curves across a variety of kernels, datasets, and targets. (Top row.)** We predict test error on synthetic target functions of real datasets. Each target is the multi-Hermite polynomial (Equation (3)) whose leading term is indicated in the plots. (Recall that $z_i = u_i^\top x / \sqrt{\gamma_i}$ are the rescaled PCA coordinates.) For each of these targets, our ansatz predicts both learning curves (top left) and the sample complexity required to achieve $\text{MSE} \leq 0.5$ (top right). **(Bottom left and center.)** We train on binarized true target functions. See Appendix D.1 for details. **(Bottom right.)** We construct synthetic targets by drawing from a Gaussian process. The source exponent controls the difficulty of the target. See Appendix D.1 for details. For all learning curve predictions, we estimate the coefficients of the target function in the predicted eigenbasis using the Gram-Schmidt process described in Appendix D.3. See Appendix D.4 for full experimental details.

eigenbasis, $f_\star(x) = \sum_i v_i \phi_i(x)$. We must measure these coefficients $v_i$ from finitely many samples of the target function.

Were the data perfectly Gaussian, the multi-Hermite polynomials would be an orthonormal basis with respect to the measure. We could then estimate the coefficients by simply taking inner products between the target vector and generated Hermite polynomials and expect the estimation error to decay as $\mathcal{O}(N^{-1/2})$ with the total number of samples $N$. However, small amounts of non-Gaussianity in the data introduce cascading non-orthogonality in the Hermite basis. As a result, the naïve method overestimates the power in the overlapping modes. To rectify this effect, we modify our measurement technique by re-orthogonalizing the sampled Hermite polynomials via the Gram-Schmidt process:[7]

$$\text{iterate over increasing } i: \quad h_i^{(\text{GS})} = \text{unitnorm}\left( h_i - \sum_{j<i} \left\langle h_j^{(\text{GS})}, h_i \right\rangle h_j^{(\text{GS})} \right) \tag{9}$$

where $h_i := h_i(\mathcal{X})$ is the $i^{\text{th}}$ multi-Hermite polynomial evaluated on the samples. The $\{h_i(\cdot)\}_i$ are ordered by increasing degree; there is no dependence on the level coefficients $c_\ell$ and thus our measurement is *kernel-independent*. We proceed to estimate the coefficients as $\hat{v}_i = \langle h_i^{(\text{GS})}, y \rangle$. As we show in Figure 3, with this single estimate of the target's near-Hermite orthonormal decomposition, we can reliably predict learning curves on a variety of tasks and kernels.

We summarize in Appendix F the various assumptions and simplifications we have made in the derivation of our theory and our successful prediction of learning curves.

---

[7]For a full discussion of this method, see Appendix D.3.

## 6  MLPs Learn Hermite Polynomials in the Order Predicted by the HEA

One consequence of our theory is that there exists a canonical learning order in which KRR learns Hermite polynomials as sample size increases: each polynomial's learning priority is given by its associated HEA eigenvalue. Here, we check whether this order also predicts the *training time* learning order of *feature-learning MLPs* (Yang & Hu, 2021). We train MLPs online on multi-Hermite polynomial target functions of Gaussian data and CIFAR-5m and count the number of steps $n_{iter}$ required to reach online loss $\texttt{MSE} \leq 0.1$. We find that the effective optimization time $\eta \cdot n_{iter}$ is well predicted by the HEA eigenvalue (Figure 4).

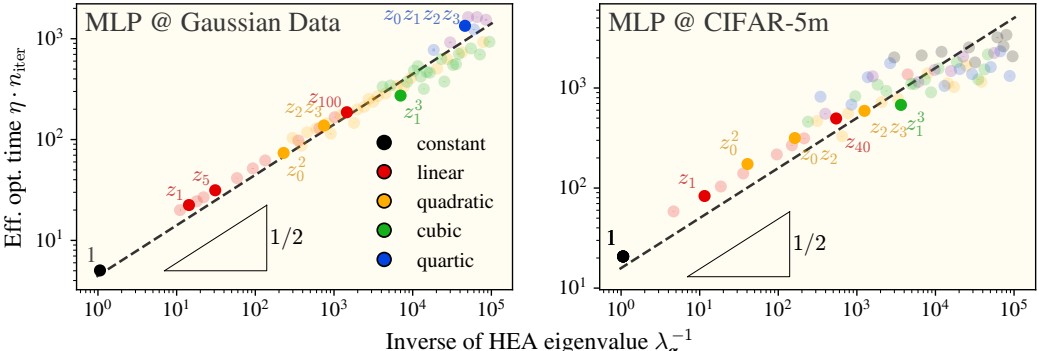

Figure 4: **The HEA accurately predicts polynomial learning order in MLPs in the feature-learning regime.** We measure the amount of time it takes to train three-layer ReLU MLPs, each dot being a trained MLP on one multi-Hermite polynomial target once a test error of $\texttt{MSE} \leq 0.1$ is reached. We find that the optimization time is well-predicted by the *inverse square root* of the HEA eigenvalue $\lambda_{\alpha}^{-1/2}$.

Details of our exact MLP setup experiments can be found in Appendix G, detailing validation of the feature-learning regime, insight into hyperparameter choices, and model performance when taken into the NTK/lazy and ultra-rich regimes.

## 7  Discussion

We have presented a theoretical framework which describes how KRR "sees" complex natural datasets: namely, as a nearly-Gaussian measure with Hermite eigenstructure as per Definition 4. This is a proof of concept that end-to-end theories of learning — mapping dataset structure all the way to model performance — are possible for a nontrivial learning algorithm on real datasets. Theories of this sort applicable to more general algorithms may be a good end goal for learning theory.

### Author Contributions

DK led the scientific development of the KRR empirics from early exploration through final experiments, wrote most of the codebase, and together with JT wrote the empirical sections of the main text. JT developed the MLP empirics and determined the conditions under which the HEA holds for KRR. YL provided formal statements of Theorems 1 and 2 and developed the proofs appearing in Appendices I to K. JS guessed the HEA idea, developed preliminary empirics and the conjectures that became our theorems, wrote most of the main text, and led the team logistically.

### Acknowledgments

The authors thank Boris Hanin for giving us an introductory schooling in Hermite polynomials. We are grateful to Berkan Ottlik for many clarifying discussions and detailed comments on the manuscript, Anthony Thomas for comments on the manuscript, and Evan Ellis for useful comments on the introduction. DK thanks the many friendly faces at Imbue for welcoming us, especially Evan Ellis for interesting conversations, Matthew Schallenkamp for useful tips, and Bowei Liu for being our GPU genie. JT thanks Kanjun Qiu and Josh Albrecht for supporting this research financially. YL acknowledges GPT-5 for multiple wrong attempts at proofs which gave salutary motivation to produce correct proofs, if only to prove someone on the internet wrong. JS thanks Parthe Pandit, Margalit Glasgow, and Jonathan Shi for useful early feedback and Josh Albrecht and Kanjun Qiu

for encouragement, guidance and support during the development of this work and the process of learning to lead. JS additionally thanks the residents of Fort Jones, CA for providing a peaceful and welcoming environment for a November 2024 stay in which the ideas developed in this paper took form.

This work was principally funded by Imbue under the Feature Lab (FLAB) initiative. This work was supported in part by the U.S. Army Research Laboratory and the U.S. Army Research Office under Contract No. W911NF-20-1-0151 awarded to Michael R. DeWeese. YL was supported by a gift to the Center for Human-Compatible AI at Berkeley from Coefficient Giving (formerly Open Philanthropy) during the development of this work.

STATEMENT ON THE USE OF LARGE LANGUAGE MODELS

We used large language models (LLMs) for analytical computations, writing code, detailed literature search on narrow topics, and the Taylor expansions of Laplace and ReLU kernels appearing in Appendix B. We found that LLMs performed certain tasks faster than us but with a propensity for miscommunication or overconfidence, especially when fashioning proofs, so we sought or performed independent verification of everything useful we got from an LLM.

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

## A  REVIEW OF HERMITE POLYNOMIALS

Our ansatz for kernel eigenstructure is constructed from Hermite polynomials, and we use several key properties throughout the paper. In this appendix, we give a brief review of Hermite polynomials.

Let us write $\mathrm{He}_\ell$ for the *probabilist's* Hermite polynomials.[8] These are the unique set of polynomials satisfying the following properties:

(i) **Degree.** $\mathrm{He}_\ell$ is a polynomial of degree $\ell$.

(ii) **Monic.** The leading coefficient is 1, so $\mathrm{He}_\ell(x) = x^\ell + \cdots$ (in particular $\mathrm{He}_0(x) = 1$).

(iii) **Orthogonality w.r.t. $\mathcal{N}(0, 1)$.** For $\ell \neq m$, it holds that $\mathbb{E}_{x \sim \mathcal{N}(0,1)}[\mathrm{He}_\ell(x)\mathrm{He}_m(x)] = 0$.

The probabilist's Hermite polynomials have squared norm $\mathbb{E}_{x \sim \mathcal{N}(0,1)}\left[\mathrm{He}_\ell^2(x)\right] = \ell!$. Since we will use the Hermite polynomials as a basis in which to express other functions, we will prefer them to have unit norm, so we use the *normalized* probabilist's Hermite polynomials $h_\ell(x) := \frac{1}{\sqrt{\ell!}}\mathrm{He}_\ell(x)$, which satisfy $\mathbb{E}_{x \sim \mathcal{N}(0,1)}[h_k(x)h_\ell(x)] = \delta_{k\ell}$. The first several such polynomials are given by

$$
\begin{aligned}
h_0(x) &= 1, \\
h_1(x) &= x, \\
h_2(x) &= \frac{1}{\sqrt{2}}\left(x^2 - 1\right), \\
h_3(x) &= \frac{1}{\sqrt{6}}\left(x^3 - 3x\right), \\
h_4(x) &= \frac{1}{\sqrt{24}}\left(x^4 - 6x^2 + 3\right), \\
h_5(x) &= \frac{1}{\sqrt{120}}\left(x^5 - 10x^3 + 15x\right).
\end{aligned}
$$

The Hermite polynomials obey a remarkable number of useful mathematical relations. In particular, single- and multidimensional integrals of Hermite polynomials against exponentials and other polynomials are often computable in closed form. The Wikipedia page (https://en.wikipedia.org/wiki/Hermite_polynomials) is a good first reference. Here we state one such integral which is useful for understanding the intuitive derivation of the HEA in Section 4.1:

$$
\mathbb{E}_{x \sim \mathcal{N}(0,1)}[h_\ell(x)\, x^m] = \begin{cases} \dfrac{m!}{\sqrt{\ell!}\, 2^{(m-\ell)/2}\left((m-\ell)/2\right)!}, & m \geq \ell \text{ and } m \equiv \ell \pmod{2}, \\ 0, & \text{otherwise.} \end{cases} \tag{10}
$$

That is, the $\ell$-th Hermite polynomial is orthogonal to all monomials $x^m$ whose order is less than $\ell$ (or whose order is simply of a different parity from $\ell$). In particular, this implies that $\mathbb{E}_{x' \sim \mathcal{N}(0,1)}[(xx')^m h_\ell(x')] = 0$ when $m < \ell$: the function $h_\ell$ lies in the nullspace of the rank-one kernel term $K(x, x') = (xx')^m$.

For future reference, we also note the multiplication and differentiation formulas:

$$
\mathrm{He}_n\mathrm{He}_m = \sum_{j=0}^{\min(m,n)} \binom{m}{j}\binom{n}{j} j!\,\mathrm{He}_{n+m-2j} \tag{11}
$$

$$
\mathrm{He}_n' = n\mathrm{He}_{n-1} \tag{12}
$$

---

[8]Some references use the *physicist's* Hermite polynomials $\mathrm{H}_\ell$. These are related to the probabilist's Hermites by $\mathrm{He}_\ell$ by $\mathrm{H}_\ell(x) = 2^{\ell/2}\mathrm{He}_\ell(\sqrt{2}\,x)$. When using Hermite polynomials, be sure you know which ones you're talking about!

# B ON-SPHERE LEVEL COEFFICIENTS FOR COMMON ROTATION-INVARIANT KERNELS

In this appendix, we tabulate the on-sphere level coefficients for various kernels. We also use this appendix as the place where we recall the functional forms of the ReLU NNGP kernel and NTK.

We begin by recalling Definition 3.

**Definition 3** (On-sphere level coefficients). The *on-sphere level coefficients* of a rotation-invariant kernel $K$ at a radius $r > 0$ are the nonnegative sequence $\mathrm{coeffs}(K, r) := (c_\ell)_{\ell \geq 0}$ such that

$$K(\boldsymbol{x}, \boldsymbol{x}') = \sum_{\ell \geq 0} \frac{c_\ell}{\ell!} (\boldsymbol{x}^\top \boldsymbol{x}')^\ell \quad \text{for all } \boldsymbol{x}, \boldsymbol{x}' \text{ such that } \|\boldsymbol{x}\| = \|\boldsymbol{x}'\| = r. \tag{13}$$

## B.1 GAUSSIAN KERNEL

For the Gaussian kernel $K(\boldsymbol{x}, \boldsymbol{x}') = e^{-\frac{1}{2\sigma^2}\|\boldsymbol{x}-\boldsymbol{x}'\|^2}$, the level coefficients $(c_\ell) = \mathrm{coeffs}(K, r)$ may be found by noting that, when $\|\boldsymbol{x}\| = \|\boldsymbol{x}'\| = r$, then

$$K(\boldsymbol{x}, \boldsymbol{x}') = e^{-\frac{r^2}{\sigma^2}} \cdot e^{\frac{\boldsymbol{x}^\top \boldsymbol{x}'}{\sigma^2}} = e^{-\frac{r^2}{\sigma^2}} \cdot \sum_{\ell \geq 0} \frac{1}{\sigma^{2\ell}\ell!} (\boldsymbol{x}^\top \boldsymbol{x}')^\ell. \tag{14}$$

Pattern-matching to Definition 3, we may then observe that

$$c_0 = e^{-\frac{r^2}{\sigma^2}},$$
$$c_1 = e^{-\frac{r^2}{\sigma^2}} \cdot \sigma^{-2},$$
$$c_2 = e^{-\frac{r^2}{\sigma^2}} \cdot \sigma^{-4},$$
$$\vdots$$
$$c_\ell = e^{-\frac{r^2}{\sigma^2}} \cdot \sigma^{-2\ell}.$$

## B.2 EXPONENTIAL KERNEL

Let the *exponential kernel* be $K(\boldsymbol{x}, \boldsymbol{x}') = e^{\frac{1}{\sigma^2}\boldsymbol{x}^\top \boldsymbol{x}'}$. We do not use this kernel in experiments or theory reported in this paper, but we nonetheless include it here because it is arguably the nicest kernel for the study of the HEA, and we used it extensively in our initial experiments. The blessing of the exponential kernel is that it admits the Taylor expansion

$$K(\boldsymbol{x}, \boldsymbol{x}') = e^{\frac{1}{\sigma^2}\boldsymbol{x}^\top \boldsymbol{x}'} = \sum_{\ell} \frac{1}{\sigma^{-2\ell}\ell!} (\boldsymbol{x}^\top \boldsymbol{x}')^\ell. \tag{15}$$

That is, the exponential kernel is exactly a dot-product kernel with coefficients $c_\ell = \sigma^{-2\ell}$. When $\sigma = 1$ and thus $c_\ell = 1$ for all $\ell$, this is in some sense the "platonic ideal" dot-product kernel (though since we must then take $\gamma_1 \ll 1$ for the HEA to hold as per the intuition developed in Section 4.1). When the domain is restricted to a sphere, the Gaussian kernel and exponential kernel are equal up to a global factor of $e^{-\frac{r^2}{\sigma^2}}$.

## B.3 LAPLACE KERNEL

Here we obtain the on-sphere level coefficients for the Laplace kernel $K(\boldsymbol{x}, \boldsymbol{x}') = e^{-\frac{1}{\sigma}\|\boldsymbol{x}-\boldsymbol{x}'\|}$.[9] Let

$$s := \frac{\boldsymbol{x}^\top \boldsymbol{x}'}{r^2} \in [-1, 1], \qquad \beta := \frac{\sqrt{2}\,r}{\sigma}. \tag{16}$$

Since $\|\boldsymbol{x} - \boldsymbol{x}'\| = \sqrt{2}\,r\,\sqrt{1-s}$ on the sphere,

$$K(\boldsymbol{x}, \boldsymbol{x}') = \exp\left(-\beta\sqrt{1-s}\right). \tag{17}$$

---

[9] Note for users of our codebase: as defined in our repo, the Laplace kernel is actually $K(\boldsymbol{x}, \boldsymbol{x}') = e^{-\frac{1}{\sqrt{2}\sigma}\|\boldsymbol{x}-\boldsymbol{x}'\|}$; that is, there is an extra factor of $\sqrt{2}$. We later moved away from this convention, but it remains in code.

**Closed form for on-sphere level coefficients.** Let $y_n(x)$ denote the reverse Bessel polynomials, with exponential generating function

$$\sum_{\ell \geq 0} y_{\ell-1}(x) \frac{t^\ell}{\ell!} = \exp\Big(\frac{1}{2x}\big(1 - \sqrt{1-2xt}\big)\Big). \tag{18}$$

Applying Equation (18) with $x = \frac{1}{2\beta}$ and $t = \frac{\beta}{2} s$ gives

$$\exp\big(\beta(1 - \sqrt{1-s})\big) = \sum_{\ell \geq 0} y_{\ell-1}\Big(\frac{1}{2\beta}\Big) \frac{(\beta s/2)^\ell}{\ell!}. \tag{19}$$

Multiplying by $e^{-\beta}$ and substituting $s = (\boldsymbol{x}^\top \boldsymbol{x}')/r^2$, the definition $K(\boldsymbol{x}, \boldsymbol{x}') = \sum_{\ell \geq 0} \frac{c_\ell}{\ell!}(\boldsymbol{x}^\top \boldsymbol{x}')^\ell$ implies

$$\boxed{c_\ell = \frac{e^{-\beta}}{r^{2\ell}} y_{\ell-1}\Big(\frac{1}{2\beta}\Big) \Big(\frac{\beta}{2}\Big)^\ell, \qquad \beta = \frac{\sqrt{2}\,r}{\sigma}.} \tag{20}$$

**The first few coefficients.** With the convention $y_{-1} \equiv 1$, $y_0(x) = 1$, $y_1(x) = 1 + x$, $y_2(x) = 3 + 3x + x^2$, $y_3(x) = 15 + 15x + 6x^2 + x^3$, we obtain (for $\beta = \sqrt{2}r/\sigma$)

$$c_0 = e^{-\beta},$$

$$c_1 = \frac{e^{-\beta}}{r^2} \frac{\beta}{2},$$

$$c_2 = \frac{e^{-\beta}}{r^4}\Big(\frac{\beta^2}{4} + \frac{\beta}{8}\Big),$$

$$c_3 = \frac{e^{-\beta}}{r^6}\Big(\frac{\beta^3}{8} + \frac{3\beta^2}{16} + \frac{\beta}{16}\Big),$$

$$c_4 = \frac{e^{-\beta}}{r^8}\Big(\frac{\beta^4}{16} + \frac{3\beta^3}{16} + \frac{5\beta^2}{32} + \frac{5\beta}{128}\Big).$$

**Large-$\ell$ asymptotics.** The dominant singularity of $F(s) = e^{-\beta\sqrt{1-s}}$ is at $s = 1$, with $F(s) = 1 - \beta\sqrt{1-s} + O(1-s)$, yielding $[s^\ell]F(s) \sim \frac{\beta}{2\sqrt{\pi}}\ell^{-3/2}$, where the coefficient extraction operator $[s^\ell]F(s)$ returns the $\ell$-th coefficient in the power series of $F(s)$. Since $c_\ell = r^{-2\ell}\,\ell!\,[s^\ell]F(s)$,

$$\boxed{c_\ell \sim \frac{\beta}{2\sqrt{\pi}} \frac{\ell!}{r^{2\ell}\,\ell^{3/2}} \qquad (\ell \to \infty), \qquad \beta = \frac{\sqrt{2}\,r}{\sigma}.} \tag{21}$$

**Subtlety: fast-growing $c_\ell$ means diverging HEA eigenvalues**

Here we encounter a subtlety with the Laplace kernel coefficients: Equation (21) states that $c_\ell \to \infty$ *superexponentially* as $\ell$ grows. Recall that the largest eigenvalue in each level predicted by the HEA is $\lambda = c_\ell \gamma_1^\ell$. This superexponential growth means that for *any* value of $\gamma_1 > 0$, no matter how small, these largest levelwise eigenvalues will eventually begin to *increase* as $\ell$ grows and continue to grow without bound. A hacky fixed-point calculation using Stirling's formula suggests that the minimum occurs at $\ell_{\min} \approx r^2 \gamma_1^{-1}$. See Figure 5 for a numerical illustration of this.

What do we make of this? From a theoretical standpoint, this is the result of the fact that since the Laplace kernel on the sphere $r\mathbb{S}^{d-1}$ has a singularity at $\boldsymbol{x}^\top \boldsymbol{x}' = r^2$, our on-sphere dot-product kernel approximation to the Laplace kernel will *diverge* when attempting to evaluate $K(\boldsymbol{x}, \boldsymbol{x}')$ at points $\boldsymbol{x}$ with larger radius $\|\boldsymbol{x}\| > r$. Since our theory is designed to work with Gaussian data, and roughly half of a Gaussian distribution will spill outside its sphere of average radius into this divergent region, the HEA predicts growing eigenvalues and infinite trace.

While this might seem to spell the doom of the HEA insofar as the Laplace kernel is concerned, our experiments (e.g. Figures 2 and 3) attest that this is not the case. We find that we can still get good experimental agreement by (a) ensuring the data has high effective dimension so that $\gamma_1$ is small[10]

---

[10]Simply decreasing all eigenvalues does not help as that also decreases the sphere radius $r$ proportionally, which Equation (21) shows then *increases* each $c_\ell$ in a manner that compensates.

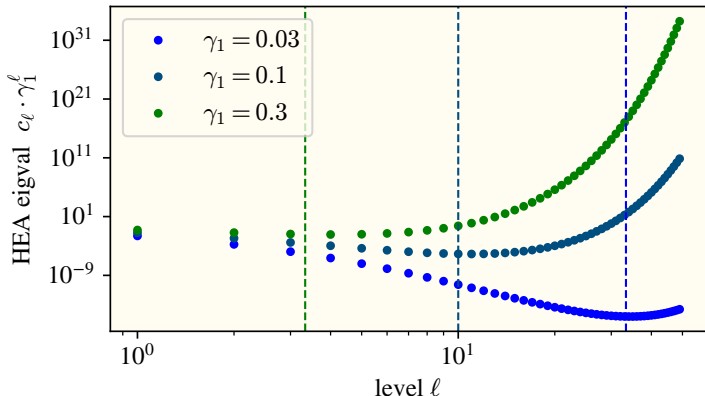

Figure 5: We compute the Laplace kernel on-sphere coefficients $c_\ell$ with $\sigma = r = 1$ and plot $c_\ell \gamma_1^\ell$ for various $\ell$. Dashed lines show $r^2 \gamma_1^{-1}$, the value of the predicted minimum.

and then (b) *truncating the HEA at finite order* $\ell$, usually $\ell \in [5, 10]$. It seems plausible to us that this series approximation to the Laplace kernel is essentially an *asymptotic expansion* rather than a true Taylor series, meaning that it gives a *good* approximation when truncated to a finite number of terms so long as a particular parameter (here $r^{-1}\gamma_1$) is small, but then later diverges, rather than giving a better and better, and ultimately perfect, approximation as the number of terms grows.

Essentially this same story also holds for the ReLU NNGP kernel and ReLU NTK, so we will not discuss it again.

### B.4 THE RELU NNGP KERNEL

For a one–hidden-layer ReLU network with first-layer weight variance $\sigma_w^2$ and bias variance $\sigma_b^2$ (and no output bias), the first-layer preactivation kernel is

$$K_1(\boldsymbol{x}, \boldsymbol{x}') = \sigma_w^2 \, \boldsymbol{x}^\top \boldsymbol{x}' + \sigma_b^2. \tag{22}$$

On the sphere $\|\boldsymbol{x}\| = \|\boldsymbol{x}'\| = r$, set

$$q := K_1(\boldsymbol{x}, \boldsymbol{x}) = \sigma_w^2 r^2 + \sigma_b^2, \qquad s := \frac{\boldsymbol{x}^\top \boldsymbol{x}'}{r^2}, \qquad \rho := \frac{K_1(\boldsymbol{x}, \boldsymbol{x}')}{\sqrt{K_1(\boldsymbol{x}, \boldsymbol{x}) K_1(\boldsymbol{x}', \boldsymbol{x}')}} = \frac{\sigma_w^2 r^2 \, s + \sigma_b^2}{q}. \tag{23}$$

The ReLU NNGP kernel is

$$K_2(\boldsymbol{x}, \boldsymbol{x}') = \frac{\sigma_w^2}{2\pi} q \left( \sqrt{1 - \rho^2} + (\pi - \arccos \rho) \rho \right) =: \frac{\sigma_w^2}{2\pi} q \, H(\rho). \tag{24}$$

We Taylor-expand $K_2$ in powers of $\boldsymbol{x}^\top \boldsymbol{x}'$ and write

$$K_2(\boldsymbol{x}, \boldsymbol{x}') = \sum_{\ell \geq 0} \frac{c_\ell}{\ell!} (\boldsymbol{x}^\top \boldsymbol{x}')^\ell. \tag{25}$$

A change of variables gives the coefficients

$$\boxed{c_\ell = \frac{\sigma_w^2}{2\pi} q \left( \frac{\sigma_w^2}{q} \right)^\ell H^{(\ell)}(a) \qquad \text{with} \qquad a := \frac{\sigma_b^2}{q}, \qquad q = \sigma_w^2 r^2 + \sigma_b^2,} \tag{26}$$

where $H(\rho) = \sqrt{1 - \rho^2} + (\pi - \arccos \rho)\rho$ and $H^{(\ell)}$ denotes the $\ell$-th derivative.

The first few coefficients follow from

$$H(a) = \sqrt{1 - a^2} + (\pi - \arccos a)\, a,$$
$$H'(a) = \pi - \arccos a,$$
$$H''(a) = \frac{1}{\sqrt{1 - a^2}},$$
$$H^{(3)}(a) = \frac{a}{(1 - a^2)^{3/2}},$$
$$H^{(4)}(a) = \frac{2a^2 + 1}{(1 - a^2)^{5/2}},$$

yielding

$$c_0 = \frac{\sigma_w^2}{2\pi}\, q \left[\sqrt{1 - a^2} + (\pi - \arccos a)\, a\right],$$
$$c_1 = \frac{\sigma_w^2}{2\pi}\, q \left(\frac{\sigma_w^2}{q}\right)(\pi - \arccos a),$$
$$c_2 = \frac{\sigma_w^2}{2\pi}\, q \left(\frac{\sigma_w^2}{q}\right)^2 \frac{1}{\sqrt{1 - a^2}}, \tag{27}$$
$$c_3 = \frac{\sigma_w^2}{2\pi}\, q \left(\frac{\sigma_w^2}{q}\right)^3 \frac{a}{(1 - a^2)^{3/2}},$$
$$c_4 = \frac{\sigma_w^2}{2\pi}\, q \left(\frac{\sigma_w^2}{q}\right)^4 \frac{2a^2 + 1}{(1 - a^2)^{5/2}}.$$

**Asymptotics.** As $\ell$ grows, the coefficient $c_\ell$ grows as

$$c_\ell = \Theta\left(\frac{\sigma_w^2}{2\pi}\, q \left(\frac{\sigma_w^2}{q}\right)^\ell \frac{\ell!}{\ell^{3/2}}\right), \qquad \ell \to \infty. \tag{28}$$

### B.5 THE RELU NTK

As in the previous subsection, we treat a shallow ReLU network. The ReLU NNGP kernel remains

$$K_2(\boldsymbol{x}, \boldsymbol{x}') = \frac{\sigma_w^2}{2\pi}\, q \left(\sqrt{1 - \rho^2} + (\pi - \arccos \rho)\, \rho\right) =: \frac{\sigma_w^2}{2\pi}\, q\, H(\rho). \tag{29}$$

The corresponding two-layer ReLU NTK is

$$\Theta_2(\boldsymbol{x}, \boldsymbol{x}') = \underbrace{\sigma_w^2\, K_1(\boldsymbol{x}, \boldsymbol{x}') \cdot \frac{1}{2\pi}(\pi - \arccos \rho)}_{\text{training first layer}} + \underbrace{K_2(\boldsymbol{x}, \boldsymbol{x}')}_{\text{training second layer}} \tag{30}$$

$$= \frac{\sigma_w^2}{2\pi}\, q \left(\sqrt{1 - \rho^2} + 2\rho(\pi - \arccos \rho)\right) =: \frac{\sigma_w^2}{2\pi}\, q\, J(\rho). \tag{31}$$

We Taylor-expand $\Theta_2$ in powers of $\boldsymbol{x}^\top \boldsymbol{x}'$ and write

$$\Theta_2(\boldsymbol{x}, \boldsymbol{x}') = \sum_{\ell \geq 0} \frac{c_\ell}{\ell!}\, (\boldsymbol{x}^\top \boldsymbol{x}')^\ell. \tag{32}$$

A change of variables gives the coefficients

$$\boxed{c_\ell = \frac{\sigma_w^2}{2\pi}\, q \left(\frac{\sigma_w^2}{q}\right)^\ell J^{(\ell)}(a) \qquad \text{with} \qquad a := \frac{\sigma_b^2}{q}, \qquad q = \sigma_w^2 r^2 + \sigma_b^2,} \tag{33}$$

where $J(\rho) = \sqrt{1 - \rho^2} + 2\rho(\pi - \arccos \rho)$ and $J^{(\ell)}$ denotes the $\ell$-th derivative.

The first few coefficients follow from the identities

$$J(a) = \sqrt{1 - a^2} + 2a\,(\pi - \arccos a),$$

$$J'(a) = 2(\pi - \arccos a) + \frac{a}{\sqrt{1 - a^2}},$$

$$J''(a) = \frac{3 - 2a^2}{(1 - a^2)^{3/2}},$$

$$J^{(3)}(a) = \frac{a\,(5 - 2a^2)}{(1 - a^2)^{5/2}},$$

$$J^{(4)}(a) = \frac{5 + 14a^2 - 4a^4}{(1 - a^2)^{7/2}},$$

yielding

$$c_0 = \frac{\sigma_w^2}{2\pi}\,q\Big[\sqrt{1 - a^2} + 2a\,(\pi - \arccos a)\Big],$$

$$c_1 = \frac{\sigma_w^2}{2\pi}\,q\left(\frac{\sigma_w^2}{q}\right)\Big[2(\pi - \arccos a) + \frac{a}{\sqrt{1 - a^2}}\Big],$$

$$c_2 = \frac{\sigma_w^2}{2\pi}\,q\left(\frac{\sigma_w^2}{q}\right)^2 \frac{3 - 2a^2}{(1 - a^2)^{3/2}},$$

$$c_3 = \frac{\sigma_w^2}{2\pi}\,q\left(\frac{\sigma_w^2}{q}\right)^3 \frac{a\,(5 - 2a^2)}{(1 - a^2)^{5/2}},$$

$$c_4 = \frac{\sigma_w^2}{2\pi}\,q\left(\frac{\sigma_w^2}{q}\right)^4 \frac{5 + 14a^2 - 4a^4}{(1 - a^2)^{7/2}}.$$

**Asymptotics.** As $\ell$ grows, the coefficient $c_\ell$ grows as

$$c_\ell = \Theta\left(\frac{\sigma_w^2}{2\pi}\,q\left(\frac{\sigma_w^2}{q}\right)^\ell \frac{\ell!}{\ell^{3/2}}\right), \qquad \ell \to \infty. \tag{34}$$

## C    REVIEW OF THE KRR EIGENFRAMEWORK FOR PREDICTING TEST PERFORMANCE FROM TASK EIGENSTRUCTURE

The central piece of existing theory which we use in this paper is a set of equations which give the average-case test MSE of KRR in terms of the task eigenstructure. In this appendix, we will review this KRR eigenframework.

This eigenframework has been derived many times by different means in both the statistical physics community and the classical statistics community (which usually phrases the result as applying to linear ridge regression). References studying KRR include Sollich (2001); Bordelon et al. (2020); Jacot et al. (2020); Simon et al. (2021); Loureiro et al. (2021); Wei et al. (2022); references studying linear ridge regression include Caponnetto & De Vito (2007); Dobriban & Wager (2018); Wu & Xu (2020); Hastie et al. (2022); Richards et al. (2021); Cheng & Montanari (2022). The result is essentially the same in all cases. Here we will adopt the terminology and notation of Simon et al. (2021).

Recall that we are studying KRR with a kernel function $K$ with data sampled $\boldsymbol{x}_i \sim \mu$, targets generated as $y_i = f_*(\boldsymbol{x}_i) + \eta$, and noise $\eta \sim \mathcal{N}(0, \epsilon^2)$ with variance $\epsilon^2 \geq 0$. The kernel admits an eigendecomposition $K(\boldsymbol{x}, \boldsymbol{x}') = \sum_i \lambda_i \phi_i(\boldsymbol{x})\phi_i(\boldsymbol{x}')$ with orthonormal eigenfunctions $(\phi_i)$. Let us decompose the target function in the eigenbasis as $f_*(\boldsymbol{x}) = \sum_i v_i \phi_i(\boldsymbol{x})$. Suppose we run KRR with $n$ samples and a ridge parameter $\delta \geq 0$, and we compute the population (i.e. test) and train MSEs as

$$\text{MSE}_{\text{te}} = \mathbb{E}_{\boldsymbol{x} \sim \mu}\left[(f_*(\boldsymbol{x}) - \hat{f}(\boldsymbol{x}))^2\right] + \epsilon^2, \tag{35}$$

$$\text{MSE}_{\text{tr}} = \frac{1}{n}\sum_i (y_i - \hat{f}(\boldsymbol{x}_i))^2. \tag{36}$$

### C.1    STATEMENT OF THE EIGENFRAMEWORK

We are now ready to state the eigenframework. Let $\kappa \geq 0$ be the unique nonnegative solution to

$$\sum_i \frac{\lambda_i}{\lambda_i + \kappa} + \frac{\delta}{\kappa} = n. \tag{37}$$

Then test risk is given approximately by

$$\text{MSE}_{\text{te}} \approx \mathcal{E}_{\text{te}} := \mathcal{E}_0 \mathcal{B}, \tag{38}$$

where the *overfitting coefficient* $\mathcal{E}_0$ is given by

$$\mathcal{E}_0 := \frac{n}{n - \sum_i \left(\frac{\lambda_i}{\lambda_i + \kappa}\right)^2} \tag{39}$$

and the *bias* is given by

$$\mathcal{B} = \sum_i \left(\frac{\kappa}{\lambda_i + \kappa}\right)^2 v_i^2 + \sigma^2. \tag{40}$$

Train risk is given by

$$\text{MSE}_{\text{tr}} \approx \mathcal{E}_{\text{tr}} \equiv \frac{\delta^2}{n^2 \kappa^2} \mathcal{E}_{\text{te}}, \tag{41}$$

What is meant by the "$\approx$" in Equations (38) and (41)? This result only becomes exact in certain stringent cases; it is formally derived under an assumption that the eigenfunctions are independent Gaussian (or sub-Gaussian) variables when $\boldsymbol{x}$ is sampled from $\mu$, and it is exact only in an asymptotic limit in which $n$ and the number of eigenmodes in a given eigenvalue range (or the number of duplicate copies of any given eigenmode) both grow large at a proportional rate (Hastie et al., 2022; Bach, 2023). These conditions emphatically do not apply to any realistic instance of KRR. Nonetheless, numerical experiments find that Equations (38) and (41) hold with small error even at modest $n$ (Canatar et al., 2021; Simon et al., 2021; Wei et al., 2022): though derived in an idealized setting and exact only in a limit, this eigenframework holds reliably in practical cases.

In this paper, we use this eigenframework as a tool to map predictions of task eigenstructure to predictions of learning curves. Since we are here using it in settings very similar to those tested by previous works (Bordelon et al., 2020; Jacot et al., 2020; Simon et al., 2021; Wei et al., 2022), we expect it to work well. Its use introduces some small error (as it is not perfect at finite $n$), but this is usually not the dominant source of error.

# D EXPERIMENTS CHECKING THE HEA: DETAILS AND FURTHER DISCUSSION

This appendix contains descriptions of the experimental stack used to verify the HEA, as well as a discussion of practical considerations for applying the HEA to real datasets. It is organized as follows:

- In Appendix D.1 we catalog the kernels, datasets, and target functions we use throughout our experiments.
- In Appendix D.2 we explain the experiments that directly check whether the kernel eigenstructure matches the HEA prediction (Figure 2).
- In Appendix D.3 we describe our method for estimating the decomposition of the target function in the Hermite eigenbasis. Unlike previous work, our method does not require constructing or diagonalizing an empirical kernel matrix.
- In Appendix D.4 we detail the experimental setups for each of the learning curve and sample complexity plots (Figures 1 and 3).
- Finally, in Appendix D.5 we show the results of various additional experiments.

## D.1 KERNELS, DATASETS, AND TARGET FUNCTIONS

**Kernels.** We use the Gaussian kernel, Laplace kernel, ReLU NNGP kernel, and ReLU NTK in our experiments. A detailed review of these kernels can be found in Appendix B.[11]

**Datasets.** We use the following datasets for the main experiments:

- **Mean-zero anisotropic Gaussian data.** This synthetic dataset is fully specified by its (diagonal) covariance. Different experiments set the data dimension and covariance decay rate differently; see experiment-specific details in Appendices D.2, E and G.
- **CIFAR-5m** (Nakkiran et al., 2020). This dataset consists of more than 5 million samples of synthetic images akin to CIFAR-10. These images were sampled using a deep generative model trained on CIFAR-10. Though the distributions of CIFAR-5m and CIFAR-10 may not be identical, they are typically considered close enough for research purposes.
- **SVHN** (Netzer et al., 2011). This dataset contains over 600,000 images of numerals, taken from house numbers found on Google Street View.
- **ImageNet-32** (Deng et al., 2009). This dataset contains downsampled ImageNet images ($32 \times 32$ pixels).
- **MNIST** (LeCun et al., 1998) **and Mushroom dataset** (Dua & Graff, 2017). We use MNIST and the UCI Mushroom tabular dataset in Appendix E as examples of insufficiently Gaussian datasets.

We sometimes employ *regularized ZCA whitening* to increase the effective dimension of the data. This is a data preprocessing technique parameterized by a ZCA strength $\omega^2$ which maps

$$\boldsymbol{X} = \boldsymbol{U}\boldsymbol{S}\boldsymbol{V}^\top \mapsto \boldsymbol{U}\boldsymbol{S}\overline{\left(\omega^2\overline{\boldsymbol{S}^2} + \boldsymbol{I}_d\right)^{-1/2}}\boldsymbol{V}^\top \tag{42}$$

where $\boldsymbol{X} \in \mathbb{R}^{d \times N}$ is the data matrix, $\boldsymbol{U}\boldsymbol{S}\boldsymbol{V}^\top$ is its SVD, and we use the normalization notation $\overline{\boldsymbol{A}} := \boldsymbol{A}/(\|\boldsymbol{A}\|_{\mathrm{F}}^2/d)$. As the ZCA strength $\omega^2 \to 0$, we get no spectral transformation apart from a scalar normalization $\boldsymbol{X} \to \boldsymbol{U}\overline{\boldsymbol{S}}\boldsymbol{V}^\top$. Conversely, when $\omega^2 \to \infty$, we get full whitening $\boldsymbol{X} \to \boldsymbol{U}\boldsymbol{V}^\top$. The crossover point of this behavior occurs at $\omega^2 \sim 1$. Note that although partially-whitened Gaussian data are slightly less anisotropic, they are still distributed as a multivariate Gaussian.

We sometimes employ sample normalization, $\boldsymbol{x} \to \boldsymbol{x}/\|\boldsymbol{x}\|$. Note that although the normalized data lie on the hypersphere, their distribution is still anisotropic.

Both sample normalization and ZCA whitening are preprocessing techniques that, on aggregate, shift high-dimensional data samples closer to the hypersphere. Since the HEA relies on an expansion

---

[11]A note for users of our codebase: in this paper, we define the Laplace kernel to be $K(\boldsymbol{x}, \boldsymbol{x}') = e^{-\frac{1}{\sigma}\|\boldsymbol{x}\boldsymbol{x}'\|}$, but because we initially used a different convention, in code it is $K(\boldsymbol{x}, \boldsymbol{x}') = e^{-\frac{1}{\sigma\sqrt{2}}\|\boldsymbol{x}\boldsymbol{x}'\|}$. When we report a kernel width in this paper, this is the width in the parameterization we use *in the paper*, not in code.

of kernel functions in terms of on-sphere coefficients (see Appendix B), these methods move any experimental setup closer to the regime well-described by the HEA. See Appendix E for further discussion of this point.

**Targets.** We use a variety of synthetic and real target functions in our experiments. All targets are scalar; the synthetic targets take continuous values, whereas the real targets are binarized ($y_i \in \{+1, -1\}$). All targets are mean-zero; for real targets, this means that the binary (super)classes are always balanced (even if the binary superclasses contain a differing number of base classes).

We use the following targets for the main experiments:

- **(Synthetic.) Multi-Hermite targets.** A single (normalized) multi-Hermite polynomials of the PCA components (Equation (3)).

- **(Synthetic.) Powerlaw targets.** We draw a random sample of the Gaussian process

$$f_\star(\boldsymbol{x}) = \sum_i^P c_i h_i(\boldsymbol{x}) + \epsilon \cdot (\text{white noise}), \tag{43}$$

  where $h_i(\boldsymbol{x})$ is shorthand for the $i^{\text{th}}$ multi-Hermite polynomial $h_{\boldsymbol{\alpha}_i}(\boldsymbol{z})$ and $c_i$ is a mean-zero Gaussian random variable with variance $(i+6)^{-\beta}$. The so-called *source exponent* $\beta$ satisfies $\beta > 1$ and controls the Sobolev smoothness of the target: the larger $\beta$ is, the smoother and easier-to-learn the target. We choose a numerical truncation threshold $P = 30,000$ for convenience, choosing the target noise level $\epsilon$ to ensure that the target is unit norm $\mathbb{E}[y_i] = 1$. Our results are empirically insensitive to the randomness in the target.

- **(Real.) `class` vs. `class`.** A binarization in which samples are only drawn from two classes.

- **(Real.) `class` vs. `all others`.** A binarization similar to a single output element of practical neural networks with one-hot label encodings. A key difference here is that samples are drawn from each binary superclass in equal proportion so that $\mathbb{E}[y_i] = 0$.

- **(Real.) Domesticated vs. wild animals.** CIFAR-5m binarization: `[cat, dog, horse]` vs. `[bird, deer, frog]`.

- **(Real.) Vehicles vs. animals.** CIFAR-5m binarization: `[plane, car, ship, truck]` vs. `[bird, cat, deer, dog, frog, horse]`. Samples are drawn from each superclass in equal proportion so that $\mathbb{E}[y_i] = 0$.

- **(Real.) Even vs. odd.** SVHN binarization based on parity: `[0, 2, 4, 6, 8]` vs. `[1, 3, 5, 7, 9]`.

- **(Real.) Prime vs. composite.** SVHN binarization based on primality: `[2, 3, 5, 7]` vs. `[4, 6, 8, 9]`. We leave out `[0]` and `[1]`, numerals whose primality is undefined.

- **(Real.) Genus $0$ vs. genus $\geq 1$.** SVHN binarization based on the numeral's topological genus: `[1, 3, 5, 7]` vs. `[0, 6, 8, 9]`. We leave out `[2]` and `[4]`, numerals whose topological genus is font-dependent.

## D.2 Direct eigenstructure checks

What is the appropriate way to numerically compare two eigensystems?

The spectra are easy to compare – one can simply check whether $\frac{|\lambda_i - \hat{\lambda}_i|}{\lambda_i}$ is small for all $i$. An easy visual check is to simply scatter one spectrum against the other on a log-log plot; if the points remain close to the $y = x$ line, then the spectra agree.

Comparing the eigenbases, on the other hand, is a subtler matter. One must be careful when the eigensystems have small eigenvalue gaps. This issue is most easily understood by considering the limit: what happens when comparing two diagonalizations of a degenerate matrix? In this case, numerical eigendecomposition is undefined since the true eigenvectors are not unique; the computed eigenvectors are thus arbitrary. Simply comparing $\hat{\phi}_i$ with $\phi_i$ for all $i$ is therefore insufficient.

Clearly, any good eigenbasis comparison must be spectrum-aware. In particular, differences between eigenvectors belonging to (near-)degenerate subspaces should not be strongly penalized.

A coarse but simple way to make this comparison is with spectral binning. We divide $\mathbb{R}^+$ into logarithmically-spaced bins; then, for each eigenbasis, we treat the modes whose eigenvalues fall within the same bin as a single near-degenerate eigenspace. Applying this procedure to the HEA[12] yields a set of disjoint Hermite eigenspaces $\{\boldsymbol{\Phi}_i^{(\text{th})}\}_{i=1}^{\texttt{nbins}}$, and likewise for the empirical basis. Let $d_i^{(\text{th})} = \dim(\boldsymbol{\Phi}_j^{(\text{th})})$ and likewise for $d_i^{(\text{emp})}$. Note that in general we do not expect $d_i^{(\cdot)}$ to equal $d_j^{(\cdot)}$ for $i \neq j$; indeed, some bins may contain no modes at all. However, we *do* expect $d_i^{(\text{th})} = d_i^{(\text{emp})}$ for all $i$ (if the theory is accurate).

Having handled any issues of spectral near-degeneracy, we may directly compare the two eigenbases by computing the pairwise overlaps between the binned eigenspaces:

$$\text{Overlap}(i, j) = \begin{cases} \dfrac{1}{d_j^{(\text{emp})}} \left\| \boldsymbol{\Phi}_i^{(\text{th})\top} \boldsymbol{\Phi}_j^{(\text{emp})} \right\|_{\text{F}}^2, & d_i^{(\text{th})} \neq 0 \text{ and } d_j^{(\text{emp})} \neq 0, \\ \text{undefined}, & \text{otherwise.} \end{cases} \tag{44}$$

where again $1 \leq i, j \leq \texttt{nbins}$ enumerate the bins.

To visualize this in Figure 2, we plot the overlaps in a heatmap, graying out pixels whose spectral bins do not contain any eigenmodes (and thus have undefined overlap). We note that discrepancies in the tails are primarily caused by distortions in the *empirical* eigenbasis caused by finite-size effects, rather than genuine disagreement between the theory and the true eigensystem (see Figure 6).

The experiments that generated Figure 2 used the following hyperparameters:

- **Gaussian kernel, gaussian data.** Kernel width $\sigma = 6$. Data $\boldsymbol{x} \in \mathbb{R}^{200}$ drawn i.i.d. Gaussian with diagonal covariance $\mathbb{E}\left[x_i^2\right] = (i + 6)^{-3.0}$. This results in $d_{\text{eff}} \approx 7$. We choose a steep covariance decay exponent to ensure that cubic modes are clearly present in the plot.
- **Gaussian kernel, CIFAR-5m.** Kernel width $\sigma = 6$. No ZCA nor sample normalization. This results in $d_{\text{eff}} \approx 9$.
- **Laplace kernel, SVHN.** Kernel width $\sigma = 8\sqrt{2}$. Whiten data with ZCA strength $\omega^2 = 5 \times 10^{-3}$ and then unit-normalize. This results in $d_{\text{eff}} \approx 21$. We generally observed that $d_{\text{eff}} \geq 20$ is a reliable rule of thumb for obtaining good agreement with the HEA using the Laplace kernel.
- **ReLU NTK, ImageNet-32.** Bias variance $\sigma_b^2 = 1.68$ and weight variance $\sigma_w^2 = 0.56$. Whiten data with ZCA strength $\omega^2 = 5 \times 10^{-3}$ and then unit-normalize. This results in $d_{\text{eff}} \approx 40$. Examining the on-sphere coefficients, we see that $\sigma_b^2/\sigma_w^2 \gg 1$ is the ReLU NTK equivalent for the wide kernel condition for Gaussian kernels.

## D.3 How to decompose the target function

We are interested in recovering the coefficients of the target $f_\star$ in the kernel eigenbasis:

$$\text{Recover } v_i \text{ from samples of } f_\star(\boldsymbol{x}) = \sum_i v_i \phi_i(\boldsymbol{x}), \tag{45}$$

where $i$ is the mode index. According to the HEA, this amounts to expanding $f_\star$ in the multi-Hermite basis:

$$f_\star(\boldsymbol{x}) = \sum_i \tilde{v}_i h_i(\boldsymbol{x}). \tag{46}$$

where $h_i(\boldsymbol{x})$ is shorthand for the $i^{\text{th}}$ multi-Hermite polynomial $h_{\boldsymbol{\alpha}_i}(\boldsymbol{z})$. We use different notation for the Hermite coefficients since the HEA will not hold exactly in practical settings.

Of course, obtaining any full expansion of $f_\star$ from $N$ samples is exactly as hard as the original learning problem. However, we *can* hope to obtain an expansion that is good enough to predict the behavior of KRR trained with up to $N$ samples.

Let us define $\boldsymbol{y} \in \mathbb{R}^N$ as the vector of target samples and $\boldsymbol{h}_i \in \mathbb{R}^N$ as the $i^{\text{th}}$ multi-Hermite polynomial evaluated on the samples. Let us stack the top $P < N$ modes $[\boldsymbol{h}_1 \ \boldsymbol{h}_2 \ \cdots \boldsymbol{h}_P]^\top$ and call

---

[12]Here, we abuse terminology by referring to the proposed Hermite basis as an eigenbasis; evaluated on finitely many samples of real data, these basis vectors may not be truly orthonormal.

the resulting matrix $\boldsymbol{H} \in \mathbb{R}^{P \times N}$.[13] We would like to recover the top $P$ coefficients $\hat{\boldsymbol{v}} \in \mathbb{R}^P$ from $\boldsymbol{y}$ and $\boldsymbol{H}$.

**Naïve first try.** As a first pass, let's simply run linear regression: $\hat{\boldsymbol{v}} = \boldsymbol{H}^\dagger \boldsymbol{y}$, where $\boldsymbol{H}^\dagger$ is the pseudo-inverse. Empirically, the recovered coefficients are not very good. There are two main reasons.

- The true target contains Hermite modes which are not represented in our top $P$ list. This is an essential model misspecification which appears to linear regression as target noise. This is not a problem in itself; the learning curve eigenframework can handle this kind of noise, *if* it knows how much noise power there is. However, it is unclear how to measure the magnitude of this misspecification noise, since it gets mixed in with the sampling "noise."

- Linear regression has a flat inductive bias: it tends to fit the samples using each of its regressors with equal enthusiasm. Here, the regressors are functions, arranged (roughly) in order of decreasing smoothness. On the other hand, natural targets are relatively smooth; their power tends to concentrate on the early modes. This leads to large estimation errors in the top coefficients.

These two challenges, left unresolved, result in poorly predicted learning curves.

**Second pass.** Let us instead consider the following greedy iterative algorithm:

---

**Algorithm 1** Greedy Residual Fitting (GRF)

---

**Require:** Target vector $\boldsymbol{y} \in \mathbb{R}^N$, Hermite feature matrix $\boldsymbol{H} \in \mathbb{R}^{P \times N}$.

1: $\boldsymbol{y}^{(0)} \leftarrow \boldsymbol{y}$
2: **for** $p = 1$ **to** $P$ **do**
3: $\quad \boldsymbol{h}_p \leftarrow \boldsymbol{H}_{p,:}$
4: $\quad \hat{v}_p \leftarrow \dfrac{\langle \boldsymbol{h}_p, \boldsymbol{y}^{(p-1)} \rangle}{\|\boldsymbol{h}_p\|_2^2}$ $\qquad\qquad\qquad\qquad$ ▷ find the coefficient of $\boldsymbol{h}_p$ in $\boldsymbol{y}^{(p-1)}$
5: $\quad \boldsymbol{y}^{(p)} \leftarrow \boldsymbol{y}^{(p-1)} - \hat{v}_p\, \boldsymbol{h}_p$ $\qquad\qquad\qquad$ ▷ subtract off the projection onto $\boldsymbol{h}_p$
6: **end for**
7: **return** $\hat{\boldsymbol{v}} = (\hat{v}_1, \dots, \hat{v}_P)$ **and** $\hat{\epsilon}^2 = \|\boldsymbol{y}^{(P)}\|_2^2$

---

This is essentially the Kaczmarz method for solving a linear system (Kaczmarz, 1937). This target recovery algorithm has a few notable virtues. First, it is *direct*; it does not require matrix inverses, so it runs in quadratic rather than cubic time. Second, since it is iterative and stateful, it automatically prioritizes correctly estimating the top coefficients, which is where the target power tends to lie. Third, it naturally estimates the noise power (it is simply the norm of the final residual $\|\boldsymbol{y}^{(\mathrm{P})}\|^2$). Fourth, it estimates the correct amount of total power: by the Pythagorean theorem, $\sum_p \hat{v}_p^2 + \hat{\epsilon}^2 = \|\boldsymbol{y}\|$ if $\|\boldsymbol{h}_p\| = \sqrt{N}$, that is, if the Hermite polynomials are indeed unit norm.[14]

We empirically find that this algorithm works well for synthetic targets on synthetic Gaussian data. However, real data is not perfectly Gaussian. Even small deviations from Gaussianity introduce *systematic* correlations between the Hermite polynomials $\{\boldsymbol{h}_i\}$; these overlaps cause the greedy algorithm to systematically overestimate the target power in the affected modes.

We empirically found that if the target is sufficiently "dense" in the Hermite modes (i.e., not dominated by very few coefficients), then these errors "average out" (loosely speaking) and the predicted learning curves remain accurate. Unfortunately, this is rarely the case in real targets. Real targets often contain much power in a few early modes, many of which suffer from this overlap problem. As a result, the predicted learning curves are often far off the mark. See Figures 9 and 11 for empirics.

---

[13]In our experiments we choose $P = 30{,}000$ and $N = 80{,}000$ since manipulating this tensor maximizes our GPU VRAM utilization.

[14]If they are not unit norm, then for the purposes of the KRR eigenframework, one usually wants to take $\hat{v}_p \mapsto N^{-1/2} \|\boldsymbol{h}_p\| \hat{v}_p$ to estimate the total amount of power in Hermite direction $p$. After adopting this scaling, GRF again conserves total power by the Pythagorean theorem: $\sum_p \hat{v}_p^2 + \hat{\epsilon}^2 = \|\boldsymbol{y}\|$.

**Third pass.** To rectify these modes, we simply squeeze out the non-orthogonality, starting from the top Hermite modes. A standard technique for iteratively eliminating overlaps is to use the Gram-Schmidt process. In practice, this simply amounts to performing a QR decomposition, $QR = H$, and estimating the coefficients from the orthogonal component as $\hat{v} = Qy$.

The major strength of this coefficient recovery technique is that it accurately predicts learning curves. Another strength is that since the regressors are orthogonal, the noise power is easily obtained as $1 - \hat{v}^\top \hat{v}$. The main drawback is that we must once again resort to a cubic-time algorithm. A natural question is then: if we are going to run a cubic-time algorithm anyways, why not simply diagonalize the empirical kernel matrix? There are several reasons:

- *Universal measurement of target.* This procedure only needs to be run once for a given target function. The resulting coefficients can then be reused to predict learning behavior across a variety of kernels and hyperparameters.

- *Finite-size effects.* Diagonalizing the finite-sample empirical kernel matrix typically introduces distortions in the tail modes. The HEA avoids this issue by constructing the basis directly.

- *Numerical conditioning.* The (non-orthogonalized) Hermite polynomial matrix $H$ is very well-conditioned on natural datasets. Performing Gram-Schmidt orthogonalization tends to be numerically robust, even at 32-bit floating-point precision. On the other hand, the numerical conditioning of kernel diagonalization worsens as the number of samples (and retrievable modes) increases; in practice, we typically need 64-bit precision to obtain reliable eigenvectors. As a consequence, the prefactors for GPU VRAM space complexity are friendlier for Gram-Schmidt.

- *Theoretical insight.* Kernel diagonalization is an opaque numerical computation; it does not expose the functional form of the eigenfunctions. The perturbed closed-form expression offered by HEA + Gram-Schmidt reveals the analytical structure of learning in kernels.

**Takeaway.** The HEA conceptually holds; the kernel eigenfunctions are small perturbations of the multi-Hermite polynomials, even for complex real data. If the target function is sufficiently dense, the perturbations are not important to model, and the target coefficients can be estimated using a direct greedy method. However, for some real targets with prominent coefficients, we must be careful to account for their perturbations.

### D.4    LEARNING CURVES AND SAMPLE COMPLEXITIES

In this section, we discuss additional considerations for evaluating learning curves. We end by cataloging the experimental parameters used to generate learning curve plots.

Evaluating the theoretical predictions requires summing over all the eigenmodes. In practice, we estimate this sum by truncating at some large $P \gg N$ and discarding the tail. For kernels with fast spectral decay, the neglected tail contributes negligibly and the truncation introduces negligible error. However, for slow-decaying kernels (e.g., Laplace kernel or ReLU NTK) the tail sum is non-negligible and we must account for it. We do this by modifying the ridge as

$$\tilde{\delta} = \delta + \sum_{i=P}^{\infty} \lambda_i \approx \delta + \left( \mathrm{Tr}\,[K] - \sum_{i=0}^{P} \lambda_i \right). \tag{47}$$

This heuristic arises from examining the eigenframework conservation law

$$n = \frac{\delta}{\kappa} + \sum_{i=0}^{\infty} \frac{\lambda_i}{\lambda_i + \kappa} \tag{48}$$

$$= \frac{\delta}{\kappa} + \sum_{i=P}^{\infty} \frac{\lambda_i}{\lambda_i + \kappa} + \sum_{i=0}^{P-1} \frac{\lambda_i}{\lambda_i + \kappa} \tag{49}$$

$$\approx \frac{\delta}{\kappa} + \sum_{i=P}^{\infty} \frac{\lambda_i}{\kappa} + \sum_{i=0}^{P-1} \frac{\lambda_i}{\lambda_i + \kappa} \tag{50}$$

where the final approximation follows from the fact that $P \gg n$ so $\kappa \gg \lambda_i$ for the tail modes. Combining terms yields the tail-corrected ridge.

Finally, we detail the parameters used in each of our main experiments. All kernel regression experiments use a ridge of $\delta = 10^{-3}$ and run up to 50 trials, each with a new test set of size 5,000. All target function estimates use 80,000 samples.

- **Figure 1, top right.** Gaussian kernel, width $\sigma = 4$. No ZCA nor normalization.

- **Figure 1, bottom right.** Laplace kernel, width $\sigma = 4\sqrt{2}$. ZCA strength $\omega^2 = 10^{-3}$ with data sample normalization. Sample complexities are obtained by computing learning curves (empirical curves averaged over 20 trials), performing logarithmic interpolation, and finding where the test risk falls below $0.5$.

- **Figure 3, top left.** Gaussian kernel, width $\sigma = 8$. No ZCA nor normalization. We use a wide kernel for better agreement at targets of high degree; for comparison with a narrower kernel, see Figure 7 below.

- **Figure 3, top right.** Same as Figure 1, bottom right.

- **Figure 3, bottom left.** Gaussian kernel, width $\sigma = 10$. No ZCA, but the samples are normalized. For comparison with a narrow kernel and no sample normalization, see Figure 8 below.

- **Figure 3, bottom center.** Laplace kernel, width $\sigma = 4\sqrt{2}$. ZCA strength $\omega^2 = 5 \times 10^{-3}$ with data sample normalization. For comparison with no sample normalization, see Figure 10.

- **Figure 3, bottom right.** ReLU NTK, bias and weight variances $\sigma_b^2 = 1.96$ and $\sigma_w^2 = 0.49$. ZCA strength $\omega^2 = 10^{-2}$ with data sample normalization. We found that the HEA for the ReLU NTK is particularly sensitive to insufficient effective dimension.

## D.5 ADDITIONAL EXPERIMENTS

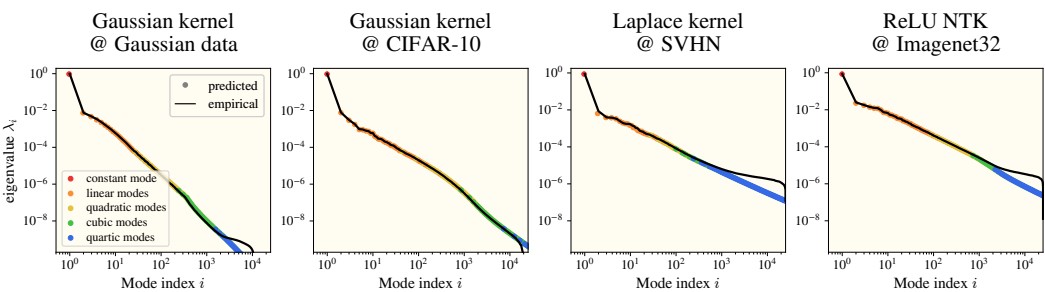

Figure 6: We show an alternative visualization of the top row of Figure 2, comparing the predicted and empirical spectra of various kernel/dataset combinations. We see that the HEA accurately predicts the minute details of the kernel spectrum. Furthermore, tail deviations are indeed caused by finite-kernel effects in the empirical spectrum rather than a failure of the HEA. We plot the empirical spectrum with a curve and the predicted spectrum with dots, instead of vice versa, so the colors of the dots can denote the polynomial order of the predicted eigenmode.

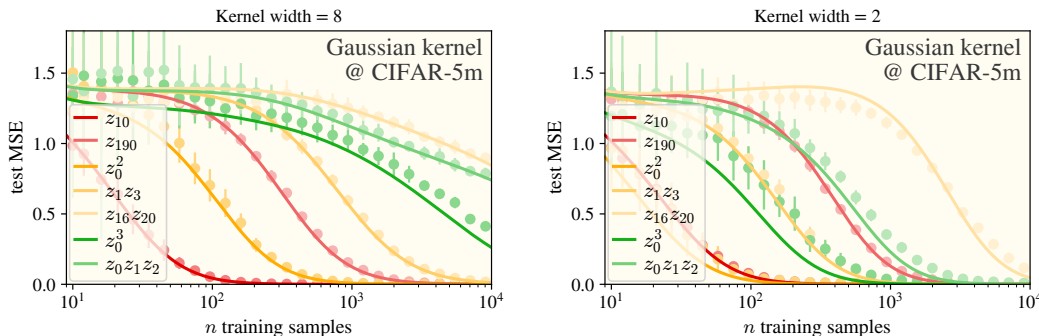

Figure 7: We compare the original plot from Figure 3 with kernel width 8 (left) to the same experimental setup except for a kernel width 2 (right). The true eigenfunctions of the narrower kernel deviate from the HEA prediction, especially for the high-degree modes. For a similar comparison, see Figure 13.

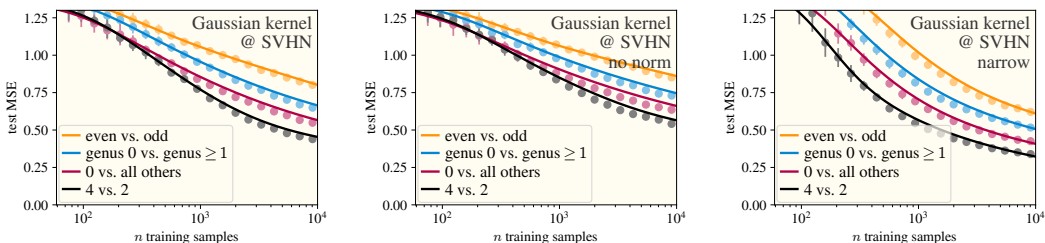

Figure 8: We compare the original experimental setup from Figure 3 (sample normalization, kernel width 10) on the left, to two similar setups: no sample normalization (center) and a narrower kernel width of 4 (right). Interestingly, the narrow kernel width does not change the agreement much; removing sample normalization worsens agreement slightly.

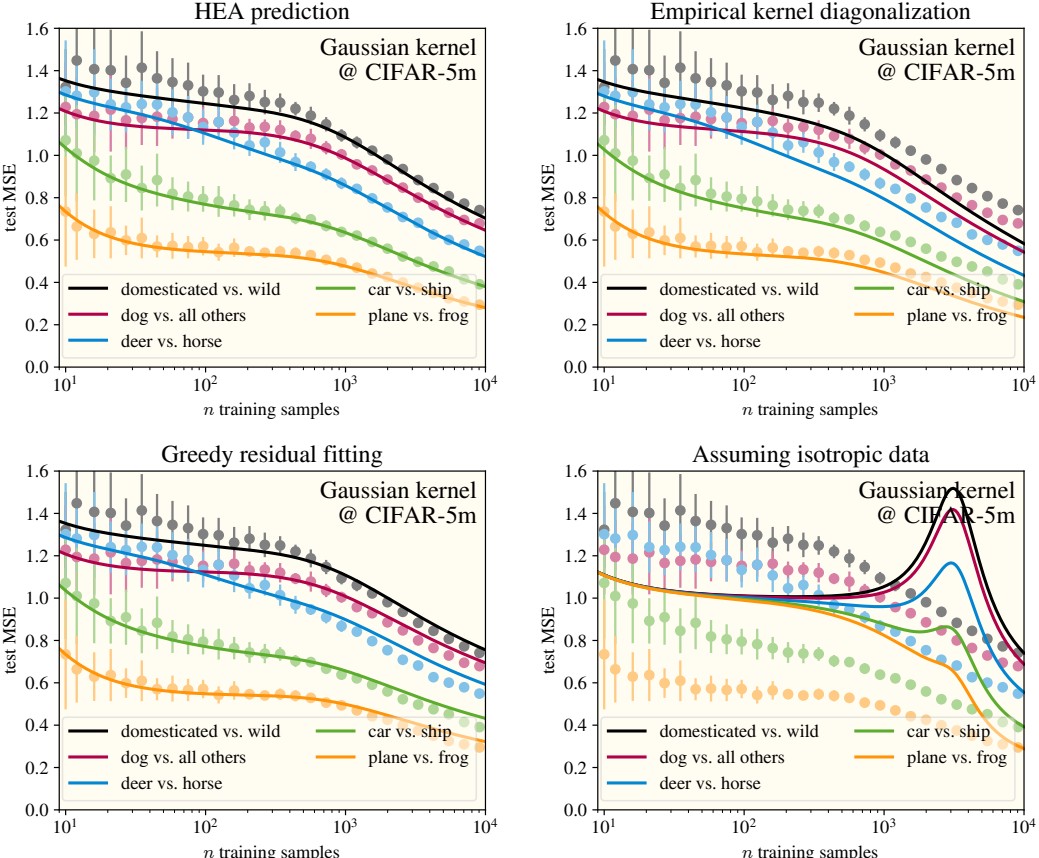

Figure 9: Here we show four plots in which the experimental setup does not change; instead, we compare different techniques for estimating the required theoretical quantities. **(Upper left.)** HEA prediction, identical to Figure 1. **(Upper right.)** We numerically diagonalize the kernel matrix (size $25000 \times 25000$) and use the obtained eigenstructure to make predictions. The accuracy of the prediction degrades as the number of training samples approaches the size of the kernel matrix used to estimate the true eigenstructure. **(Lower left.)** We use the greedy algorithm described in Algorithm 1 to estimate the target coefficients. We see good initial agreement, but it degrades due to accumulating non-orthogonality. **(Lower right.)** For comparison, we include the predictions one would obtain if one modeled the data distribution as an isotropic Gaussian. Clearly, a major contribution of our work is to handle the anisotropy in natural data, since it strongly affects the resulting learning curves.

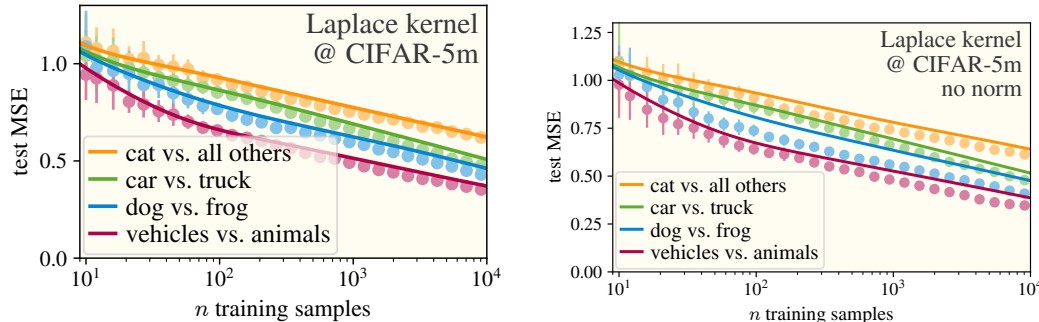

Figure 10: We compare the plot from Figure 3 on the left with an identical experimental setup, except without sample normalization. We see that normalization is necessary for obtaining agreement, since the HEA is derived using the kernel's on-sphere level coefficients.

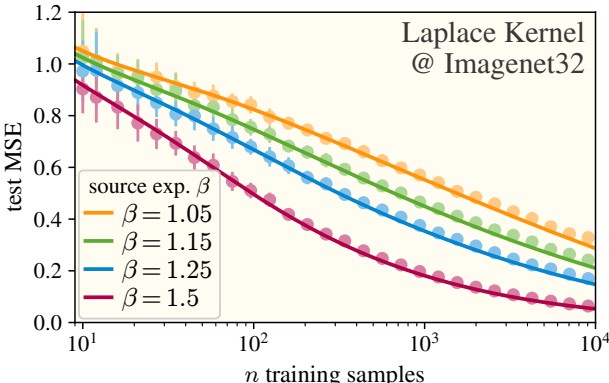

Figure 11: For "dense" synthetic targets (i.e., targets whose power is split among many eigenfunctions), the HEA predicts the performance of the Laplace kernel very well. Contrasting this with the predictions for Laplace kernel on real targets (Figure 10), we conclude that dense targets are more forgiving of errors in target coefficient recovery. For this experiment, we use ZCA strength $\omega^2 = 10^{-2}$ and sample normalization.

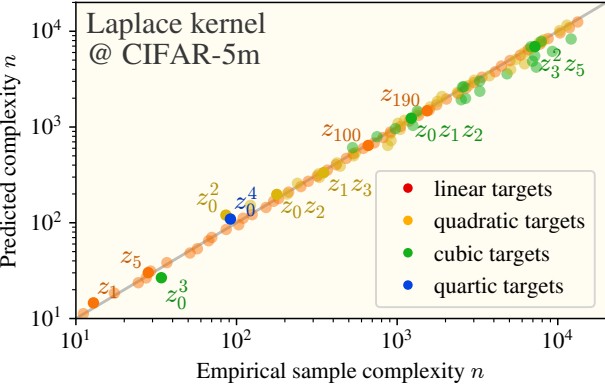

Figure 12: Additional sample complexity plot, this time for Laplace kernel on CIFAR-5m. We use sample normalization and a ZCA strength of $3 \times 10^{-3}$.

# E   WHAT BREAKS THE HEA? CASE STUDIES OF THREE CAUSES OF FAILURE

In the main text, we have primarily shown results in settings in which the HEA works quite well in predicting kernel eigenstructure, learning curves, and monomial learning rates. Here we get our hands dirty and show where it breaks.

**First cause of failure: narrow kernel width.** The discussion of Section 4.1 suggests that if we make kernel width too narrow — or, in kernels without a notion of width, change the kernel so that $\frac{c_{\ell+1}\gamma_1}{c_\ell}$ is not much less than one — we should start to see the HEA break. In particular, since for a Gaussian kernel $c_\ell \propto \sigma^{-2\ell}$, narrow width means non-small level coefficient ratios and vice versa. Figure 13 shows an empirical test: on the same synthetic data distribution, the kernel is made progressively narrower. As expected, the HEA breaks.

**Second cause of failure: low data dimension leading to non-concentration of norm.** The HEA treats all rotation-invariant kernels as if they were dot-product kernels. In fact, it throws away all information about the kernel function that cannot be obtained by its value on a sphere. For this to work in practice, we should expect our data to lie fairly close to a sphere. For Gaussian data $x \sim \Sigma$, this is assured if the data has high effective dimension $d_{\text{eff}} := \frac{\text{Tr}[\Sigma]^2}{\text{Tr}[\Sigma^2]}$. Figure 14 shows that with a Laplace kernel on a Gaussian dataset, the HEA breaks as $d_{\text{eff}}$ decreases.

Interestingly, not all rotation-invariant kernels require high effective dimension. For example, our heuristic derivation of the HEA for Gaussian data in Section 4.1 took place in just one dimension! As it turns out, this is because the Gaussian kernel is given by a polynomial feature map (of infinite dimension), but the Laplace kernel has a cusp at $x = x'$ and is not. (Equivalently, the Gaussian kernel is analytic, but the Laplace kernel is not.) For a stationary kernel $K(x, x') = k\left(\frac{1}{\sigma^2}\|x - x'\|^2\right)$ which depends only on the distance between points with scaling factor $\sigma$, the $\sigma \to 0$ limit will give the HEA on Gaussian data if the function $k(\cdot)$ is real-analytic at zero. The Gaussian kernel satisfies this real-analytic criterion, but the Laplace kernel does not. As evidence of this, Figure 15 repeats the above experiment with a Gaussian kernel and finds that low $d_{\text{eff}}$ is no problem.

We note that there is another problem conflated with this one: *as discussed in Appendix B, the Laplace kernel's level coefficients actually diverge superexponentially, and the predicted HEA eigenvalues diverge faster the closer $\frac{\gamma_1}{\sum_i \gamma_i}$ is to one.* One could disentangle this by studying a rotation-invariant kernel that (a) does not factor into a product of one-dimensional kernels like the Gaussian and exponential kernels do but (b) is still analytic, so its level coefficients will not diverge superexponentially. Such a kernel is not among those studied here, so we leave this to future work.

**Third cause of failure: the data not being "Gaussian enough."**

The HEA is theoretically derived for Gaussian data, but nonetheless works for some real datasets. This suggests that the datasets for which it works are "Gaussian enough" in some sense. Here we support this intuition with experiments. First, in Figure 16, we plot the marginal distributions of the first few normalized principal coordinates for four datasets of decreasing complexity: CIFAR-10, SVHN, MNIST, and the tabular UCI Mushrooms dataset (Dua & Graff, 2017). Plotted this way, these datasets appear increasingly non-Gaussian. Then, in Figure 17, we test the HEA on these four datasets, finding that the HEA indeed seems to work worse as the dataset becomes "less Gaussian." For good measure and out of interest, we plot the first few PCA coordinates for the three image datasets in Figure 18.

Of course, we put "Gaussian enough" in quotes because this is a nonrigorous and ill-defined notion: we have not invoked any precise method of determining which of two datasets is "closer" to Gaussian! We do not attempt to give a precise definition here. This seems like a worthwhile direction for future exploration, especially from the statistics community.

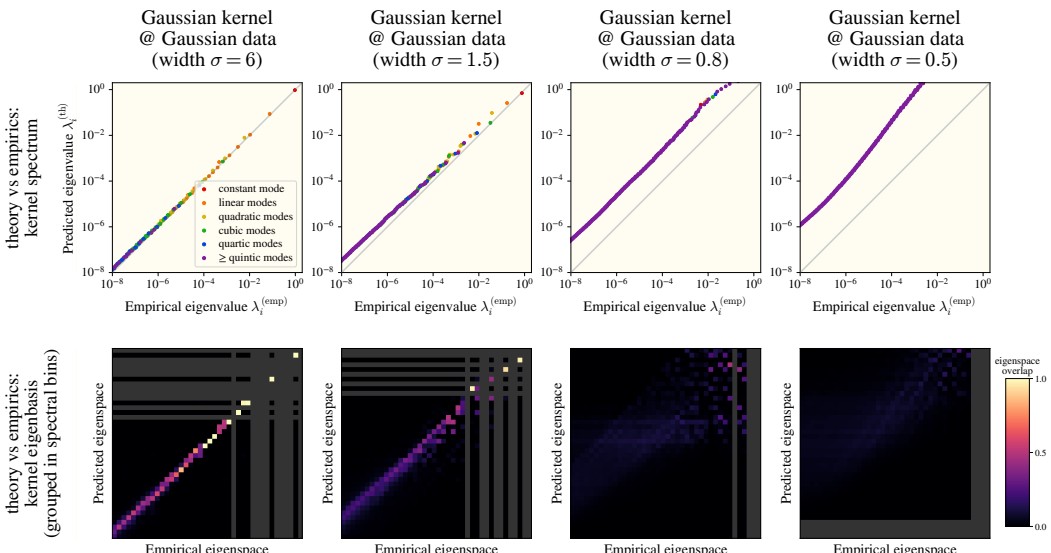

Figure 13: **The HEA breaks when kernel width is too narrow.** We repeat the experiment of Figure 2 with the Gaussian kernel function $K(\boldsymbol{x}, \boldsymbol{x}') = e^{-\frac{1}{2\sigma^2}\|\boldsymbol{x}-\boldsymbol{x}'\|^2}$ and synthetic Gaussian data. The data is generated as $\boldsymbol{x} \sim \mathcal{N}(0, \boldsymbol{\Gamma})$, where $\boldsymbol{\Gamma}$ has eigenvalues $\gamma_i \propto i^{-3}$ for $i = 1 \ldots 30$, with $\gamma_1 \approx 0.83$ after normalization. As the kernel width $\sigma$ shrinks to $\sigma \lesssim 1$, the HEA breaks: eigenvalue prediction fails with higher-order modes dominating predictions, and the clear eigenspace overlap present at large width evaporates.

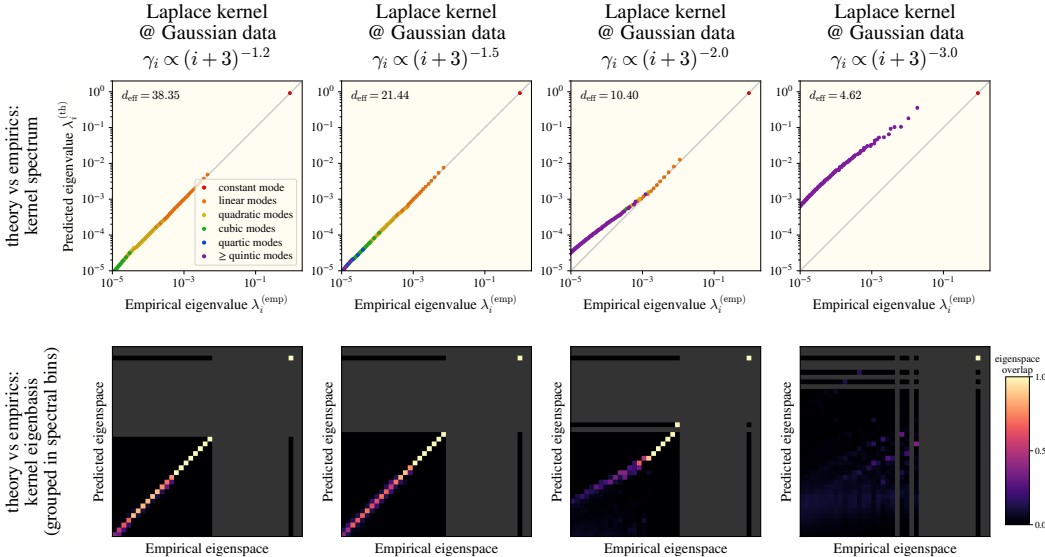

Figure 14: **With the Laplace kernel, the HEA requires high effective data dimension...** We repeat the experiment of Figure 13 with two modifications: first, the kernel function is Laplace $K(\boldsymbol{x}, \boldsymbol{x}') = e^{-\frac{1}{\sigma}\|\boldsymbol{x}-\boldsymbol{x}'\|}$ with width $\sigma = 8\sqrt{2}$, and second, the eigenvalues are $\gamma_i \propto (i+3)^{-\alpha}$ with variable exponent $\alpha$. As $\alpha$ increases (moving left to right), the effective dimension $d_{\text{eff}} = \frac{(\sum_i \gamma_i)^2}{\sum_i \gamma_i^2}$ decreases, and agreement with the HEA degrades. Empirically, we find that $d_{\text{eff}} \approx 20$ is usually high enough to see good agreement.

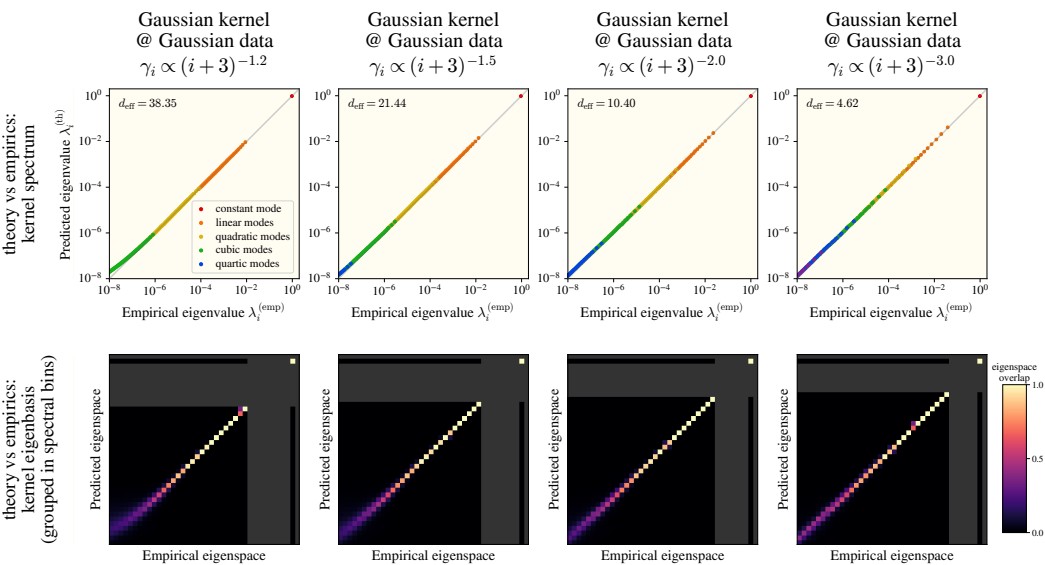

Figure 15: **...but with the Gaussian kernel, it does not need high effective dimension.** We repeat the experiment of Figure 14 exactly, but use a Gaussian kernel with width $\sigma = 3$. Thanks to the smoothness of the Gaussian kernel, we see little degradation of the theory-experiment match of the HEA.

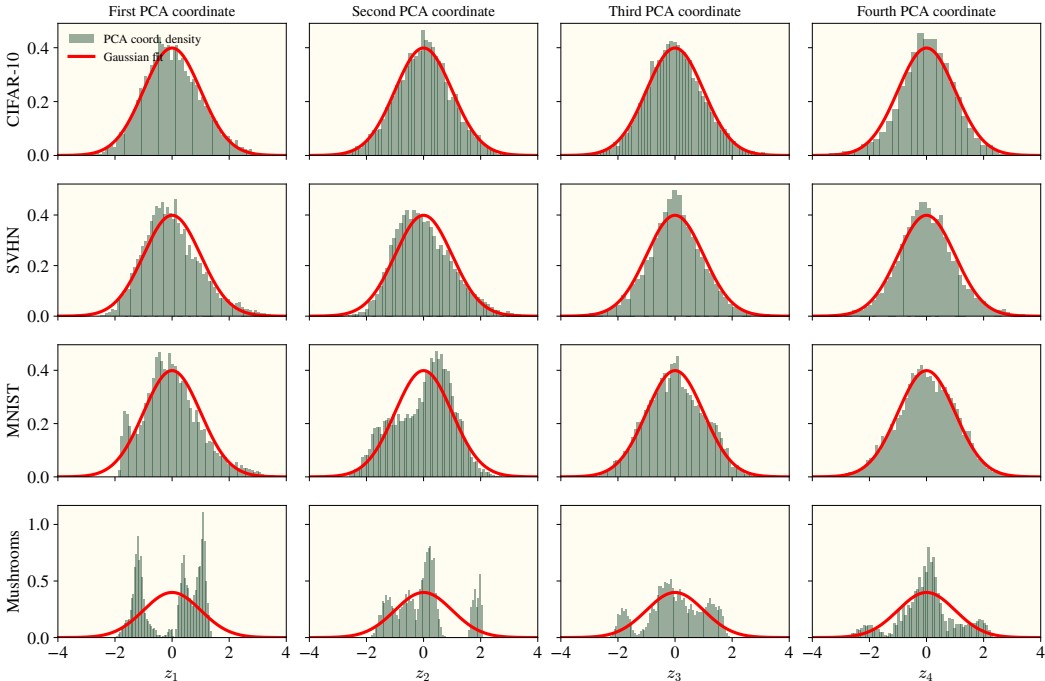

Figure 16: **Different datasets have different degrees of Gaussianity in their early principal components.** For four natural datasets — CIFAR-10, SVHN, MNIST, and the UCI Mushrooms tabular dataset — we compute the first four normalized principal coordinates $(z_i)_{i=1}^4$ and plot their distributions as histograms. We find that the first few principal coordinates for CIFAR-10 are fairly close to Gaussian, this is less true for SVHN, yet less true for MNIST, and not at all true for the Mushrooms dataset.

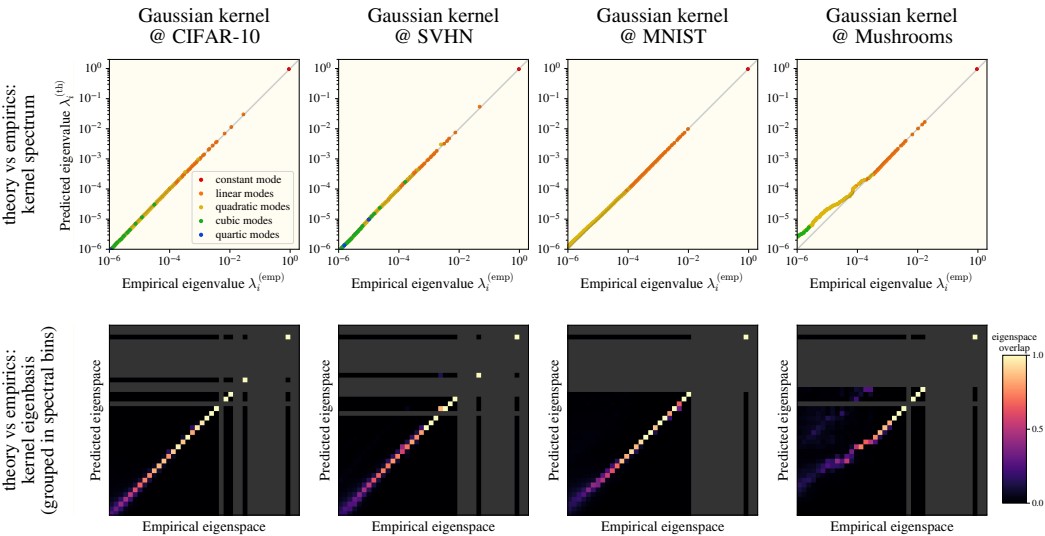

Figure 17: **The HEA works better with complex high-dimensional datasets and worse with simpler datasets.** We repeat the experiment of Figure 2 with a Gaussian kernel of width $\sigma = 3$ on 8000 samples from four datasets of increasing non-Gaussianity: CIFAR-10, SVHN, MNIST, and the UCI Mushrooms tabular dataset. As the dataset becomes simpler, the HEA works worse and worse, though the predicted trend is still roughly present, if quantitatively wrong, even for the tabular dataset.

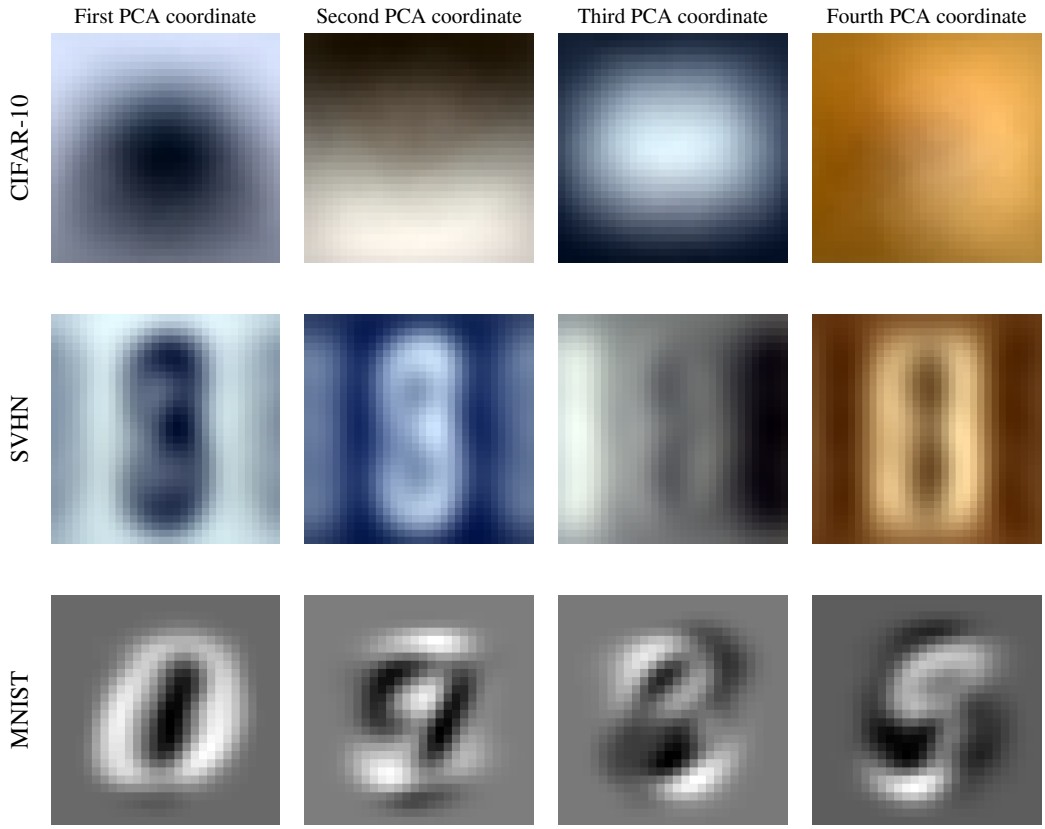

Figure 18: Visualization of the first four PCA directions for the CIFAR-10, SVHN, and MNIST datasets.

## F    List of Assumptions Made in the Prediction of Learning Curves

In Section 5, we demonstrate that using our HEA framework, we are able to predict KRR learning curves from the data covariance, a Taylor expansion of the kernel, and a polynomial decomposition of the target function. We made many assumptions and simplifications on the path to deriving this theory. We summarize and discuss them here:

1. We model the data distribution as an anisotropic Gaussian in the input space in order to get analytical eigenvalue and eigenfunctions.
2. We assume any given rotation-invariant kernel may be replaced with a dot-product kernel.
3. Even with assumptions 1 and 2, the HEA holds exactly only in limiting cases (e.g. Theorems 1 and 2). We pretend it is exact.
4. We assume our Gram-Schmidt procedure finds the correct Hermite coefficients of $f_*$.
5. In order to invoke the KRR eigenframework, we assume Gaussian universality on the kernel eigenfeatures.
6. Even with Gaussian universality, the KRR eigenframework has finite error at finite sample size $n$. We assume that $n$ is large enough that this error is small.

Steps 1-3 above get us to the point where the HEA gives the kernel eigensystem. These are the assumptions discussed and validated in Section 4.2. Step 4 treats the decomposition of $f_*$. Steps 5 and 6 are assumptions made in the use of the KRR eigenframework. Prior work has established that Gaussian universality reliably holds and the eigenframework gives accurate predictions even at modest $n$, and we encountered no additional subtleties in this work. We note that this prior Gaussian universality in kernel feature space, which allows the use of random matrix theory tools in the derivation of the KRR eigenframework, is distinct from our modeling of the data as Gaussian in the *input* space.[15] The latter is a stronger assumption, more naive and so more surprising that it often seems to work well.

Each step here will introduce some error to the final prediction of learning curves. The total error will be small only when each individual source of error is small. In this paper, we opt for a scientific approach to this problem: we use theory as guidance to suggest when each source of error will be small, and ultimately rely on empirics to validate our assumptions, checking steps individually and then putting them all together for the end-to-end prediction of learning curves. Much of the challenge in getting the empirical results for this paper lay in identifying, isolating, and fixing errors due to one of these steps. While it comes with no formal guarantees, this scientific approach has the practical advantage of efficiently surveying a wide swath of settings and cases, notably including real datasets.

It is also possible to take a mathematical approach to the problem of cascading errors: presumably, one might analytically control the error introduced in each of the above steps, leading to error bounds in the final learning curve. (See Wortsman & Loureiro (2025) for an analysis of this type.) This approach will be difficult, with the greatest challenge lying in the application to real datasets: what can we say analytically about CIFAR-10 that might enable formal proof of the empirical success of our theory? This is a worthy challenge, and this general type of question — finding useful formal conditions applicable to real datasets — is important for the advancement of the theory of machine learning. We leave this as an open problem and invite interested parties to reach out to us with ideas or results.

---

[15]Clearly it is a useful trick in the study of machine learning theory to replace distributions with moment-matched Gaussians.

# G   MLP EXPERIMENTS: DETAILS AND FURTHER DISCUSSION

As the HEA requires level coefficients to predict $\lambda_{\boldsymbol{\alpha}}$, level coefficients for MLP experiments are computed with the ReLU NTK with both bias and weight variances of 1 which match the empirical variances of our MLP networks at initialization.

Unless otherwise stated, all experiments were performed using a feature-learning ($\mu$P) MLP with the following data hyperparameters:

$$\gamma_i = (i + 6)^{-\alpha}$$
$$\alpha = 1.7$$
$$\text{input dimension } d = 200$$

and MLP hyperparameters,

$$\text{width w} = 8192$$
$$\text{depth (number of hidden layers) L} = 2$$
$$\text{learning rate } \eta = 2 \cdot 10^{-2}$$
$$\text{batch size bsz} = 1024$$
$$\zeta = 1,$$

where $\zeta$ is the richness parameter (defined by a rescaling of the network's output $f \mapsto f/\zeta$ and either learning rate $\eta_{\max} \longmapsto \eta_0 \cdot \zeta^{2/L}$ for $\zeta > 1$ or $\eta_{\max} \longmapsto \eta_0 \cdot \zeta^2$ for $\zeta < 1$, $\eta_0$ representing a richness-independent learning rate) allowing a network to scale between an ultra-rich and a lazy/NTK regime (Atanasov et al., 2024). In order to avoid network outputs exploding for $\zeta \ll 1$, we have all MLPs output the difference $f(\boldsymbol{x}; \theta_t) - f(\boldsymbol{x}; \theta_0)$ for network parameters at gradient step $t$ being $\theta_t$.

All experiments were carried out in an online setting, with $n_{\text{iter}}$ being the number of SGD steps it took to reach a train error $\text{MSE}_{\text{tr}} \leq 0.1$. This choice is largely inconsequential to the results, and was only chosen such that (1) we ensure each target function is properly fit enough, and (2) training did not run for a prolonged time. Random samplings of n=bsz Gaussian datapoints produce variance in high-order Hermite polynomials, so an exponential moving average (EMA) for $\text{MSE}_{\text{tr}}$ is used with decay constant 0.9: $\text{EMA}_{\text{tr}, t} = 0.9\text{MSE}_{\text{tr}, t} + 0.1\text{EMA}_{\text{tr}, t-1}$. This corresponds with a half-life for the train error of about 7 steps. This ensures high-variance random samplings do not cause the effective optimization time $\eta \cdot n_{\text{iter}}$ to be skewed in favor of higher order modes. We varied the EMA constant and found values less than 0.9 to underestimate high order modes' (as we had a higher probability for random samples to have a naturally low Hermite norm) optimization time and values greater than 0.9 to overestimate lower order modes' optimization time (due to all modes' optimization time being offset by a constant). One could subtract the half-life from an empirically found $n_{\text{iter}}$ with any EMA constant to get a more EMA-unbiased estimate for when the test error goes below the set cutoff, although we do not perform such a procedure. We additionally varied the loss cutoff (in the termination condition $\text{MSE}_{\text{tr}} \leq \text{cutoff}$) and found that small changes did not significantly affect our results.

This network is empirically validated to be in the feature learning regime by looking at the Gram matrix $W_1^\top W_1$, with $W_1 \in \mathbb{R}^{\text{width} \times \text{dim}}$ initialized with Gaussian entries. A lazy network has this Gram matrix constant throughout training, whereas feature learning allows for $\Theta(1)$ changes to the spectral norm of the $W_1$, producing $\Theta(1)$ changes to $W_1^\top W_1$ (Yang et al., 2023). We train $\zeta = 1$ networks on simple PCA-mode target functions and find significantly more power in the Gram matrix entries corresponding to the data indices used in the target function, confirming that the MLP is operating in the feature-learning regime (Figure 19).

We check if our hyperparameter choices were simply lucky, or if our $\lambda_{\boldsymbol{\alpha}}^{-1/2} \sim \eta \cdot n_{\text{iter}}$ scaling is real and reproducible. We take our base case, and vary all hyperparameters (depth, width, learning rate, batch size, data covariances) we do not expect to affect the scaling, finding that none do; the only change we observe is an eigenvalue-independent rise of effective optimization time when going to a more shallow network. Results are summarized in Figure 20.

The only hyperparameter we know will change the exponent of $\lambda_{\boldsymbol{\alpha}}^{\text{exp.}} \sim \eta \cdot n_{\text{iter}}$ is the richness parameter $\zeta$: when $\zeta \ll 1$, the kernel analysis in the main text tells us we expect exponent 1. What is

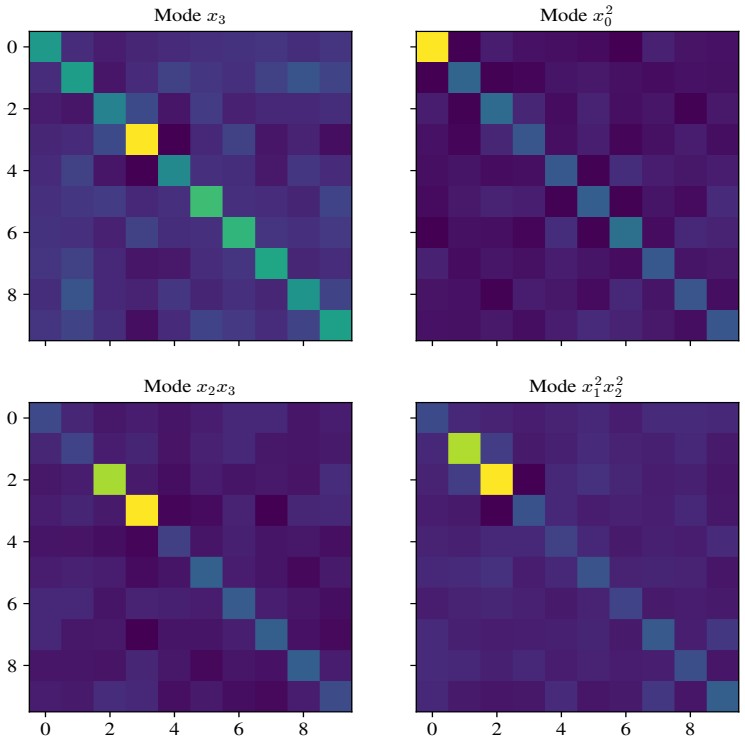

Figure 19: **Validation of MLPs being feature-learning.** We investigate the Gram matrix of the first weight matrix, $W_1^\top W_1$. We train MLPs with $\zeta = 1$ on $d = 10$ Gaussian synthetic data for 1000 GD steps. Yellow denotes higher entries, while blue denotes near-zero entries. The Gram matrices are heavily weighted to modes present in the target function, confirming the feature learning regime.

unknown is the scaling exponent as $\zeta \gg 1$, which we find to be lower than the baseline $1/2$. Lastly, our effective optimization time's $\eta$ is defined as the base $\zeta$-independent learning rate. Our findings are shown in Figure 21.

Lastly, we would like to note that the base synthetic case and the CIFAR-5m case were averaged over 3 trials, and upon finding a remarkably small standard deviation in effective optimization time, all further validation experiments were completed with only a single trial.

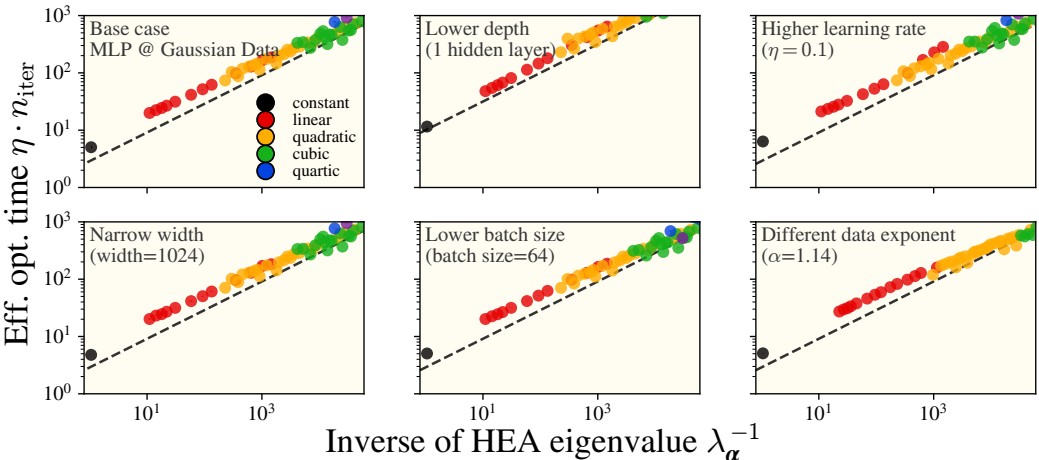

Figure 20: **Our primary MLP results are largely invariant to hyperparameter choices.** We train MLPs on synthetic Gaussian data, with one hyperparameter being motified. The base case is shown in the **(top left)** for reference. Apart from the **(top center)**, all modifications produce no notable changes to our observation of $\lambda_{\alpha}^{-1/2} \sim \eta \cdot n_{\text{iter}}$ scaling. **(Top center).** The 2 hidden layer case is only different than the rest by a change in eigenvalue-independent prefactor, the scaling law still holding.

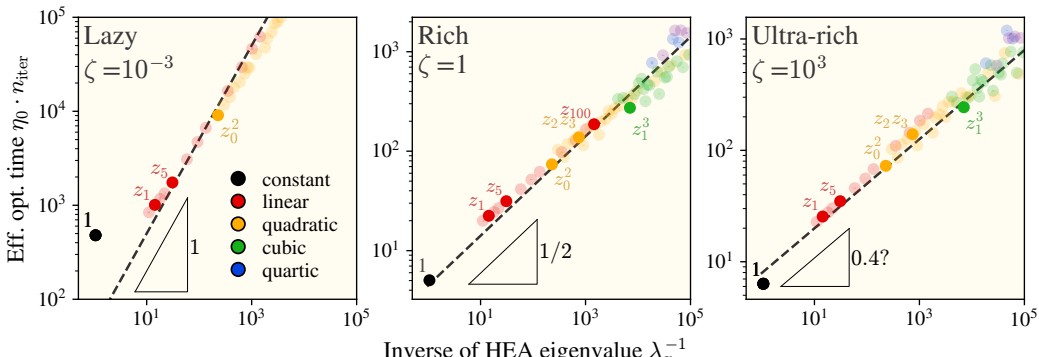

Figure 21: **Sweeping the richness parameter changes the slope. All.** The effective optimization time uses the richness-independent learning rate $\eta_0$ instead of $\eta$. **(Left)** We validate taking $\zeta \ll 1$ to bring us into the lazy/NTK regime, where we know the eigenvalue to be inversely proportional to the effective optimization time. To speed up training for the lazy runs, we set $\eta_0 = 2 \cdot 10^1$. This likely is the cause of the constant mode being notably high. **(Right)** We investigate the so-called "ultra-rich" regime $\zeta \gg 1$ and find there to be a $\lambda_{\alpha}^{-0.4} \sim \eta \cdot n_{\text{iter}}$ relationship. We suspect the $0.4$ slope in the ultra-rich regime would change with variation of hyperparameters and is thus not fundamental, unlike the slopes of $1$ and $1/2$ in the lazy and rich regimes.

# H REVIEW OF HILBERT SPACE THEORY

In this section we review the basics of Hilbert space theory, which would be very useful in proving theorems in the next few appendices.

Naïvely, a Hilbert space is just $\mathbb{R}^n$ or $\mathbb{C}^n$, where $n$ is allowed to go to infinity. In the standard notation, it is written as $(\mathcal{H}, \langle \cdot | \cdot \rangle)$, where $\mathcal{H}$ is the set of vectors, and the angular brackets $\langle \cdot | \cdot \rangle$ is the inner product on pairs of vectors in the set.

## H.1 DIRAC NOTATION

We use the standard Dirac notation.

- Vectors are written as $|v\rangle \in \mathcal{H}$.

- The duals of vectors on $\mathcal{H}$ are written as $\langle v|$, defined via the inner product: $\langle v| : |w\rangle \mapsto \langle v|w\rangle$.
- Linear operators are written as capital letters: $A$.
- Linear operators act on the right: $\langle v|A|w\rangle := \langle v|(|Aw\rangle)$.

For example, the Euclidean space $\mathbb{R}^d$ is a Hilbert space. Its inner product is the dot-product $\langle v|w\rangle = v^\top w$. Write column vectors as kets and row conjugates as bras:

$$|v\rangle = \begin{bmatrix} v_1 \\ \vdots \\ v_d \end{bmatrix}, \quad \langle v| = |v\rangle^\top = [v_1 \quad \cdots \quad v_d], \quad \langle v|w\rangle = \langle v| \, |w\rangle = v^\top w.$$

For a matrix $A \in \mathbb{R}^{d \times d}$, it acts on vectors by multiplication on the right. Rank-one operators are of the form $|v\rangle\langle w|$, with action $|x\rangle \mapsto |v\rangle\langle w|x\rangle$.

## H.2 OPERATORS

In a Euclidean space, such as $\mathbb{R}^3$, the geometry is defined by its lengths and angles. A linear operator on $\mathbb{R}^3$ preserves lengths and angles, and therefore preserves its geometry, iff it preserves the dot-product between vectors. Such geometry-preserving linear operators are called orthogonal operators.

Generalized to Hilbert spaces, geometry-preserving linear operators are called **unitary**. An operator $V : \mathcal{H} \to \mathcal{K}$ is unitary iff

$$\langle Vv|Vw\rangle = \langle v|w\rangle, \quad \|V|v\rangle\| = \||v\rangle\|.$$

For example, a unitary $V : \mathbb{R}^n \to \mathbb{R}^n$ is just an orthogonal matrix. Unitary maps preserve all geometric relationships, such as angles between vectors, lengths of vectors, and orthonormal bases.

The rank of an operator is the dimension of its range. By the analog of singular value decomposition in Hilbert space, if an operator has finite rank $r$, then it can be written in the form of $\sum_{k=1}^{r} |v_k\rangle\langle u_k|$.

In a Euclidean space, operators have transposes. The transposition of operators satisfies $(A^\top w)^\top v = w^\top(Av)$. Generalized to Hilbert space, operators have **adjoints**. The adjoint of an operator $A$ is written as $A^*$, and satisfies

$$\langle w|Av\rangle = \langle A^*w|v\rangle$$

In a Euclidean space, symmetric operators are operators such that $A = A^\top$. In a Hilbert space, **self-adjoint** operators are operators such that $A = A^*$.

Just like how symmetric operators on $\mathbb{R}^d$ are of the form $\sum_{k=1}^{d} a_k v_k v_k^\top$, self-adjoint operators are of the form $\sum_k a_k |v_k\rangle\langle v_k|$, although the summation may be infinite.

The **operator norm** is a norm defined on operators. It is defined by $\|A\|_{op} := \sup_{x \in \mathcal{H}, \|x\|=1} \|Ax\|$.

An operator is **positive definite**, written as $A \succ 0$, iff it is self-adjoint, and $\langle v|A|v\rangle > 0$ for all nonzero $|v\rangle$. Similarly, an operator is **positive semidefinite**, written as $A \succeq 0$, iff it is self-adjoint, and $\langle v|A|v\rangle geq 0$ for all $|v\rangle$.

## H.3 COMPACT OPERATORS

In infinite dimensions, some finite-dimensional intuitions break. A finite-rank operator, by virtue of being finite-rank, behaves essentially the same as a linear operator between two finite-dimensional spaces. However, they are somewhat too trivial. Compact operators are a compromise. They can have infinite rank, but still allow some finite-dimensional intuitions to apply.

An operator $K : \mathcal{H} \to \mathcal{H}$ is **compact** iff it is the operator-norm limit of a sequence of finite-rank operators.

Symmetric matrices are diagonalizable. Similarly, if an operator is compact and self-adjoint, then it is diagonalizable: There exists an orthonormal basis of eigenvectors $\{|e_j\rangle\}_{j \in J}$ and real eigenvalues $\{\lambda_j\}_{j \in J}$ with

$$K = \sum_{j \in J} \lambda_j |e_j\rangle\langle e_j|, \quad \lambda_j \in \mathbb{R}, \quad \lambda_j \to 0.$$

Nonzero eigenvalues have finite multiplicity, and the eigenvalues can only accumulate at zero.

We will only study operators of a very specific form that arises naturally from kernel regression.

Let $\{|v_n\rangle\}_{n\geq 0}$ be an orthonormal set in $\mathcal{H}$ and let $a_n \geq 0$ with $\sum_n a_n < \infty$. Define

$$K := \sum_{n=0}^{\infty} a_n |v_n\rangle\langle v_n|,$$

Then:

- $K$ is self-adjoint, positive, has finite trace, and compact.
- $K$ has eigenpairs $\{(a_n, |v_n\rangle) : n \geq 0\}$, counting multiplicities of repeated $a_n$.
- $\|K\| = \sup_n a_n$, $\text{Tr}\,[K] = \sum_n a_n$.

### H.4 FUNCTION SPACES

Generally, the vector space of functions is infinite-dimensional. Hilbert space theory shows that certain geometric intuitions honed from low-dimensional Euclidean spaces are still useful even in an infinite-dimensional function space.

As a basic example, consider $L^2(dx)$, the set of square-integrable functions on $\mathbb{R} \to \mathbb{R}$, with inner product

$$\langle f|g\rangle = \int_{\mathbb{R}} f(x)g(x)dx$$

This generalizes to $L^2(d^D x)$, the space of square-integrable functions on $\mathbb{R}^D \to \mathbb{R}^D$. It has inner product

$$\langle f|g\rangle = \int_{\mathbb{R}^D} f(x)g(x)d^D x$$

Let $\mu$ be a probability measure on $\mathbb{R}^d$. Then $L^2(\mu)$ is the space of functions $f$ such that $\int_{\mathbb{R}^d} |f(x)|^2 \mu(dx)$. That is, the space of functions $f$, such that if $X \sim \mu$, then $f(X)$ has finite second-moment.

The inner product $\langle f|g\rangle = \mathbb{E}_{X\sim\mu}[f(X)g(X)]$.

In this formalism, the expectation operator is simply taking the inner product by the constant-1 vector:

$$\mathbb{E}_{X\sim\mu}[f(X)] = \mathbb{E}_{X\sim\mu}[1\ f(X)] = \langle 1|f\rangle$$

We will mostly study $L^2(\mu)$ in the case where $\mu$ is a Gaussian distribution $\mathcal{N}(0, \Sigma)$. Those cases are called **Gaussian spaces**.

We will concern ourselves mainly with integral kernel operators over $L^2(\mu)$. An integral kernel operator is a linear operator of the type $K : L^2(\mu) \to L^2(\mu)$. It is defined using a kernel function $K : \mathbb{R}^d \times \mathbb{R}^d \to \mathbb{R}$, such that

$$K|v\rangle = |w\rangle : \quad w(x) = \int_{\mathbb{R}^d} K(x, y)v(y)\mu(dy)$$

Solving for the eigensystem of $K$ is equivalent to diagonalizing the kernel operator, such that

$$K = \sum_{n=0}^{\infty} \lambda_n |w_n\rangle\langle w_n|$$

for some orthonormal basis $|w_n\rangle$.

As an example, the dot-product kernel in $\mathbb{R}^1$ is

$$K(x, y) = \sum_{n=0}^{\infty} \frac{c_n}{n!}(xy)^n$$

In operator form, it is equivalent to

$$K = \sum_{n=0}^{\infty} \frac{c_n(2n-1)!!}{n!}|v_n\rangle\langle v_n|$$

where we define the vectors corresponding to the monomial functions:

$$|v_n\rangle, \quad v_n(x) = \frac{x^n}{\sqrt{(2n-1)!!}}$$

The factor of $\sqrt{(2n-1)!!}$ is necessary to make sure the vectors are unit-length: $\langle v_n|v_n\rangle = 1$.

The problem with the representation

$$K = \sum_{n=0}^{\infty} \frac{c_n(2n-1)!!}{n!}|v_n\rangle\langle v_n|$$

is that, while it looks like a diagonalization, it is not, because the vectors are unit-length, but not orthogonal to each other. Diagonalizing the kernel is impossible in general, but it is possible in certain limiting cases. Our main theoretical work in this paper is just working out these cases.

## H.5 1D GAUSSIAN SPACE

Let $\mu(dx) = (2\pi)^{-1/2}e^{-x^2/2}dx$ be the standard Gaussian distribution on $\mathbb{R}^1$. We have the 1D Gaussian space $L^2(\mu)$.

The differentiation operator $D_x$ maps $f(x)$ to $f'(x)$, and by integration-by-parts, has adjoint

$$D_x^* = -D_x + x$$

Therefore, we can define the Ornstein–Uhlenbeck (OU) operator

$$L := -D_x^2 + xD_x = (D_x^*D_x).$$

Then $L$ is self-adjoint and positive semidefinite on $L^2(\mu)$.

The eigenvector equation for $L$ is

$$L|v_n\rangle = \lambda_n|v_n\rangle \implies v_n''(x) - xv_n'(x) + \lambda_n v_n(x) = 0$$

This is just Hermite's differential equation, with solutions being the probabilist's Hermite polynomials $\mathrm{He}_n$. They are orthogonal in $L^2(\mu)$ with

$$\mathbb{E}_\mu[\mathrm{He}_m(X)\mathrm{He}_n(X)] = n!\delta_{mn}.$$

but not normalized. They are normalized by

$$h_n(x) := \frac{\mathrm{He}_n(x)}{\sqrt{n!}}, \quad |h_n\rangle \in L^2(\mu).$$

Thus we see that the normalized Hermite polynomials is the eigensystem for the OU operator in Gaussian space:

$$L|h_n\rangle = n|h_n\rangle, \quad L = \sum_{n=0}^{\infty} n|h_n\rangle\langle h_n|$$

Since the OU operator is the simplest way to make a self-adjoint operator out of the differentiation operator $D_x$, it is a natural object to consider.

## H.6 MULTIDIMENSIONAL GAUSSIAN SPACE

The gaussian space over $\mathbb{R}^1$ can be generalized to multiple dimensions. Let $\mu_d = \mathcal{N}(0, I_d)$ be the standard gaussian distribution on $\mathbb{R}^d$. The corresponding OU operator is

$$L = -\Delta + x \cdot \nabla = \sum_{i=1}^{d} -\partial_{x_i}^2 + x\partial_{x_i}$$

That is, we can write $L$ as a sum of 1-dimensional OU operators, as $L = \sum_{i=1}^{d} L_i$. Note that the operator operates on the dimensions $x_1, \ldots, x_d$ without interfering with each other:

$$L(f_1(x_1) \cdots f_d(x_d)) = (L_1 f_1)(x_1) f_2(x_2) \cdots f_d(x_d) + \cdots + f_1(x_1) \cdots f_{d-1}(x_{d-1})(L_d f_d)(x_d)$$

Because of this, the eigensystem of $L$ is obtained directly by taking the product of the eigensystem of $L_1, \ldots, L_d$.

For a multiindex $\boldsymbol{\alpha} = (\alpha_1, \ldots, \alpha_d) \in \mathbb{N}^d$ set

$$\mathrm{He}_{\boldsymbol{\alpha}}(x) := \prod_{i=1}^{d} \mathrm{He}_{\boldsymbol{\alpha}_i}(x_i), \quad h_{\boldsymbol{\alpha}}(x) := \prod_{i=1}^{d} \frac{\mathrm{He}_{\alpha_i}(x_i)}{\sqrt{\alpha_i!}}, \quad |\boldsymbol{\alpha}| := \sum_i \alpha_i.$$

Then, $L$ is diagonalized as

$$L = \sum_{\boldsymbol{\alpha} \in \mathbb{N}^d} |\boldsymbol{\alpha}| |h_{\boldsymbol{\alpha}}\rangle\langle h_{\boldsymbol{\alpha}}|$$

For each $n = 0, 1, 2, \ldots$, the eigenspace

$$\mathcal{E}_n := \mathrm{span}\{|h_{\boldsymbol{\alpha}}\rangle : |\boldsymbol{\alpha}| = n\}$$

has multiplicity

$$\dim \mathcal{E}_n = \#\{\boldsymbol{\alpha} \in \mathcal{N}^d : |\boldsymbol{\alpha}| = n\} = \binom{n+d-1}{d-1}.$$

This degeneracy is due to symmetry. For any orthogonal transformation $M : \mathbb{R}^d \to \mathbb{R}^d$ and any smooth $f : \mathbb{R}^d \to \mathbb{R}$, we have $L(f \circ M) = (Lf) \circ M$. Consequently, if $|v_n\rangle$ is a solution to the eigenvector equation $L|v_n\rangle = \lambda_n |v_n\rangle$, so would its spherical rotations, and any vector sum of them would have the same eigenvalue $\lambda_n$.

This is similar to the case of solid harmonics in $\mathbb{R}^d$. In that case, $\Delta$ is also spherically symmetric, and its eigenspaces accordingly are degenerate.

## H.7    UNITARY TRANSFORMATIONS OF GAUSSIAN SPACES

Consider two Gaussian spaces $L^2(\mathcal{N}(0, I_d))$, $L^2(\mathcal{N}(0, \Sigma))$. One of them is standardized, the other is not. Since many properties of the Hermite polynomials are derived over the standard Gaussian space, we would like to translate statements in $L^2(\mathcal{N}(0, I_d))$ to statements in $L^2(\mathcal{N}(0, \Sigma))$.

By the geometric view point, as long as a statement is cast in the language of Hilbert space geometry, the statement will continue to hold true after any unitary transformation.

Now, suppose we have a matrix $M$ such that $MM^\top = \Sigma$, then

$$\mathbb{E}_{X \sim \mathcal{N}(0, \Sigma))}[f(X)g(X)] = \mathbb{E}_{Z \sim \mathcal{N}(0, I_d))}[f(LZ)g(LZ)]$$

Thus, we have a unitary transformation $V : L^2(\mathcal{N}(0, \Sigma)) \to L^2(\mathcal{N}(0, I_d))$, defined by

$$(Vf)(x) = f(Mx), \quad (V^* f)(x) = f(M^{-1}x)$$

Note that the matrix $M$ has an ambiguity: If $MM^\top = \Sigma$, then given any orthogonal matrix $O$, we also have $(OM)(OM)^\top = \Sigma$. Thus, we can obtain an entire family of unitary transformations, one per orthogonal matrix $O$. We will exploit this degree of ambiguity later.

Vectors can be transformed. So can operators. An operator $A$ is transformed to $VAV^*$, so that $(VAV^*)(V|v\rangle) = V(A|v\rangle)$.

$$
\begin{array}{ccc}
L^2(\mathcal{N}(0, \Sigma)) & \xrightarrow{\ A\ } & L^2(\mathcal{N}(0, \Sigma)) \\
V \downarrow & & \downarrow V \\
L^2(\mathcal{N}(0, I_d)) & \xrightarrow[VAV^*]{} & L^2(\mathcal{N}(0, I_d))
\end{array}
$$

If an operator is represented as $\sum_n a_n |w_n\rangle\langle v_n|$, then its transformed operator is represented as $\sum_n a_n V|w_n\rangle\langle v_n|V^*$. In particular, diagonalized operators stay diagonalized:

$$A = \sum_n \lambda_n |v_n\rangle\langle v_n| \implies VAV^* = \sum_n \lambda_n V|v_n\rangle\langle v_n|V^*$$

Kernels are transformed in the same way as functions, since

$$\mathbb{E}_{X \sim \mathcal{N}(0, \Sigma))}\big[\mathbb{E}_{Y \sim \mathcal{N}(0, \Sigma))}[f(X)K(X, Y)g(Y)]\big]$$
$$= \mathbb{E}_{X \sim \mathcal{N}(0, I_d))}\big[\mathbb{E}_{Y \sim \mathcal{N}(0, I_d))}[f(MX)K(MX, MY)g(MY)]\big]$$

### H.8 SOME USEFUL PROPERTIES OF HERMITE POLYNOMIALS

Let $\mu$ be the standard normal distribution on $\mathbb{R}^1$. For each $n \in \mathbb{N}$, define the normalized monomial vector $|v_n\rangle$ by the function $\frac{x^n}{\sqrt{(2n-1)!!}}$, and define the normalized probabilist's Hermite polynomial vector $|h_n\rangle = \frac{1}{\sqrt{n!}}\operatorname{He}_n(x)$.

They are both sequences of unit vectors in $L^2(\mu)$. However, the sequence $|h_0\rangle, |h_1\rangle, \ldots$ is orthonormal, but the sequence $|v_0\rangle, |v_1\rangle, \ldots$ is not orthonormal, and in fact, is increasingly ill-conditioned.

By looking up a standard reference table, we have the following basic properties:

1. $|h_0\rangle, |h_2\rangle, |h_4\rangle, \ldots$ are obtained by Gram–Schmidt orthonormalization of $|v_0\rangle, |v_2\rangle, |v_4\rangle, \ldots$.

2. $|h_1\rangle, |h_3\rangle, |h_5\rangle, \ldots$ are obtained by Gram–Schmidt orthonormalization of $|v_1\rangle, |v_3\rangle, |v_5\rangle, \ldots$.

3.
$$|h_n\rangle = \sqrt{n!} \sum_{m=0}^{\lfloor \frac{n}{2} \rfloor} \frac{(-1)^m \sqrt{(2n-4m-1)!!}}{2^m m!(n-2m)!} |v_{n-2m}\rangle$$

4. $|v_n\rangle = \sum_{m=0}^{\infty} M_{nm}|h_m\rangle$ where $M$ is an invertible lower-triangular matrix satisfying

$$M_{n,n-2m} = \frac{n!}{\sqrt{(2n-1)!!}} \frac{1}{2^m m!\sqrt{(n-2m)!}}$$

$$M_{n,n-2m}^{-1} = \sqrt{n!}\frac{(-1)^m \sqrt{(2n-4m-1)!!}}{2^m m!(n-2m)!}$$

for $0 \leq 2m \leq n$.

5.
$$\langle v_n|v_m\rangle = \begin{cases} \frac{(n+m-1)!!}{\sqrt{(2n-1)!!(2m-1)!!}} & n \equiv m \mod 2 \\ 0 & \text{else} \end{cases}$$

In particular,

$$\vdots$$

$$\langle v_n|v_{n-4}\rangle = \frac{\sqrt{(2n-1)(2n-3)(2n-5)(2n-7)}}{(2n-1)(2n-3)},$$

$$\langle v_n|v_{n-2}\rangle = \frac{\sqrt{(2n-1)(2n-3)}}{2n-1},$$

$$\langle v_n|v_n\rangle = 1,$$

$$\langle v_n|v_{n+2}\rangle = \frac{2n+1}{\sqrt{(2n+1)(2n+3)}},$$

$$\langle v_n|v_{n+4}\rangle = \frac{(2n+1)(2n+3)}{\sqrt{(2n+1)(2n+3)(2n+5)(2n+7)}}$$

$$\vdots$$

A useful operator is $M_x$, the multiply-by-$x$ operator, which, using the multiplication formula

$$x\operatorname{He}_n = \operatorname{He}_{n+1} + n\operatorname{He}_{n-1}$$

gives

$$M_x = \sum_n \sqrt{n+1}|h_{n+1}\rangle\langle h_n| + \sqrt{n+1}|h_n\rangle\langle h_{n+1}|$$

It is equivalently expressed as $M_x = a + a^*$, where $a = \sum_n \sqrt{n+1}|h_n\rangle\langle h_{n+1}|$ is the lowering ladder operator. Furthermore, because $\operatorname{He}_n' = n\operatorname{He}_{n-1}$, the lowering ladder operator is the differentiation operator $D_x$. Therefore, $a^* = D_x^* = M_x - a = M_x - D_x$.

# I PROOF OF THEOREM 1

In this appendix, we prove Theorem 1, which states that the eigensystem of a Gaussian kernel $K_\sigma(\boldsymbol{x}, \boldsymbol{x}') = e^{-\frac{1}{2\sigma^2}\|\boldsymbol{x}-\boldsymbol{x}'\|^2}$ on a multidimensional anisotropic Gaussian measure $\mu = \mathcal{N}(0, \boldsymbol{\Sigma})$ approaches the Hermite eigenstructure given in Definition 4 as $\sigma \to \infty$. To show this, we will obtain exact expressions for the kernel eigensystem, then simply show that the eigenvalues and eigenvectors are those predicted by the HEA up to terms that vanish as $\sigma$ grows.

We will proceed in three stages of successive generality:

1. We solve the problem for $L^2(\mu)$, where $\mu = \mathcal{N}(0, 1)$ is the standard Gaussian. This is the hardest stage.

2. We take an outer product of measures to solve the problem for $L^2(\mu)$, where $\mu = \mathcal{N}(0, \boldsymbol{I}_d)$.

3. We solve the problem for $L^2(\mu)$, where $\mu = \mathcal{N}(0, \boldsymbol{\Sigma})$, by taking an opportune unitary transformation from $L^2(\mathcal{N}(0, \gamma))$ to $L^2(\mathcal{N}(0, 1))$.

## I.1 PART 1: THE 1D UNIT GAUSSIAN

We solve the problem for $L^2(\mu)$, where $\mu = \mathcal{N}(0, 1)$ is the standard Gaussian. Our approach here is more conceptual and geometric compared to the one used by (Zhu et al., 1997).

We first quote the Mehler formula (Mehler, 1866):

$$\frac{1}{\sqrt{1-\rho^2}} \exp\left(-\frac{\rho^2(x^2+y^2)-2\rho xy}{2(1-\rho^2)}\right) = \sum_{n=0}^{\infty} \rho^n h_n(x) h_n(y)$$

for any $\rho \in [0, +1)$. where $h_k$ is the normalized Hermite polynomial.

We multiply by $(1-\rho)$, and substitute with $\rho = e^{-t}$, to make the form more beautiful:

$$\sqrt{\tanh(t/2)} \exp\left(-\frac{e^{-t}(x^2+y^2)-2xy}{4\sinh t}\right) = \sum_{n=0}^{\infty} (1-e^{-t}) e^{-nt} h_n(x) h_n(y)$$

It is more beautiful, because $\sum_{n=0}^{\infty}(1-e^{-t})e^{-nt} = 1$, meaning that the right side could be read as a discrete exponential distribution over the Hermitian modes.

Define an integral kernel operator $K_t : L^2(\mu) \to L^2(\mu)$ using the expression on the left:

$$(K_t f)(x) = \int_{\mathbb{R}^1} \sqrt{\tanh(t/2)} \exp\left(-\frac{e^{-t}(x^2+y^2)-2xy}{4\sinh t}\right) f(y) \mu(dy)$$

The Mehler formula then states that $K_t$ is diagonalized as:[16]

$$K_t = \sum_{n=0}^{\infty} (1-e^{-t}) e^{-nt} |h_n\rangle\langle h_n|.$$

Thus, we have successfully obtained a one-parameter family of integral kernel operators, all diagonalized in the Hermite basis. Each operator has a Gaussian kernel. However, none of them matches the form that we want:

$$K(x, y) = e^{-\frac{1}{2\sigma^2}(x-y)^2}$$

In order to reach such a form, we need to construct $\{T_\tau : \tau \in \mathbb{R}\}$, a one-parameter family of unitary transformations $L^2(\mu)$. Then we will solve for $\tau, t$, such that $T_\tau K_t V_\tau^*$ has the desired kernel form.

For any $\tau \in \mathbb{R}$, define the **squeezing operator** $T_\tau : L^2(\mu) \to L^2(\mu)$ by

$$(T_\tau f)(x) = e^{\tau/2} e^{-\frac{e^{2\tau}-1}{4}x^2} f(e^\tau x)$$

This is unitary by direct integration and change of variables. It is a one-parameter family, and it satisfies

$$T_\tau^* = T_{-\tau}, \quad T_{\tau_1} \circ T_{\tau_2} = T_{\tau_1+\tau_2}$$

---

[16]In the language of quantum mechanics, this is a Boltzmann distribution of the quantum harmonic oscillator, i.e. a thermalized state.

We can interpret this one-parameter family as a continuous "rotation" in $L^2(\mu)$. Except that, because $L^2(\mu)$ has infinitely many dimensions, the rotation need not return to the starting point. Concretely, consider what happens when the 0th Hermite vector $|h_0\rangle$ is rotated. The family of vectors $T_\tau|h_0\rangle$ would not turn back towards $|h_0\rangle$ again. Instead, it continues to rotate further and further away, with

$$\langle h_0|T_\tau|h_0\rangle = \frac{1}{\sqrt{\cosh\tau}} \to 0$$

as $\tau \to \infty$. Such behavior is possible, because there are infinitely many dimensions to rotate to.

The one-parameter family $T_\tau$ can be written as $T_\tau = e^{\tau A}$, where $A$ is the **infinitesimal generator** of the family:

$$A = \partial_\tau|_{\tau=0}T_\tau = xD_x + \frac{1}{2}(1 - x^2) = \mathrm{He}_1 D_x + \frac{1}{2}\mathrm{He}_2$$

Using the multiplication and differentiation formulas of the Hermite polynomials, we have

$$A\mathrm{He}_n = -\frac{1}{2}(\mathrm{He}_{n+2} - n(n-1)\mathrm{He}_{n-2}),$$

$$A^2\mathrm{He}_n = \frac{1}{4}(\mathrm{He}_{n+4} - 2(n^2 + n + 1)\mathrm{He}_n + n(n-1)(n-2)(n-3)\mathrm{He}_{n-4})$$

Thus,

$$\langle h_n|A|h_n\rangle = 0, \quad \langle h_n|A^2|h_n\rangle = -\frac{n^2 + n + 1}{2}$$

Therefore, the angle between $|h_n\rangle$ and $T_\tau|h_n\rangle$, at the limit of small $\tau$, is

$$\arccos\langle h_n|e^{\tau A}|h_n\rangle = \sqrt{\frac{n^2 + n + 1}{2}}\tau + O(\tau^2)$$

Equivalently, since

$$A|h_n\rangle = -\frac{1}{2}\sqrt{(n+1)(n+2)}|h_{n+2}\rangle + \frac{1}{2}\sqrt{n(n-1)}|h_{n-2}\rangle$$

we see that the operator rotates $|h_n\rangle$ in the direction of $|h_{n+2}\rangle$ and $|h_{n-2}\rangle$ simultaneously. This explains our previous statement that $e^{\tau A}$ rotates $|h_n\rangle$ further and further away, towards $|h_\infty\rangle$.

Now, for any two function $f, g$, we have

$$\langle f|T_\tau K T_\tau^*|g\rangle = \iint (T_{-\tau}f)(x)K(x, y)(T_{-\tau}g)(y)\mu(dx)\mu(dy)$$

$$= \iint f(u)K_\tau(u, v)g(v)\mu(du)\mu(dv)$$

where the transformed kernel is

$$K_\tau(x, y) = e^\tau e^{-\frac{e^{2\tau}-1}{4}(x^2+y^2)}K(e^\tau x, e^\tau y)$$

Consider the previously solved case of $K = K_t$. Plugging it in, we find that $T_\tau K T_\tau^*$ is an integral kernel operator with kernel function

$$K_{t,\tau}(x, y) = e^\tau \sqrt{\tanh(t/2)} \exp\left(-\left(\frac{e^{2\tau}-1}{4} + \frac{e^{-t}e^{2\tau}}{4\sinh t}\right)(x^2 + y^2) + \frac{e^{2\tau}}{2\sinh t}xy\right).$$

To match the target form $K(x, y) = \exp(-(x - y)^2/(2\sigma^2))$,

$$\frac{e^{2\tau}}{2\sinh t} = \frac{1}{\sigma^2}, \qquad \frac{e^{2\tau}-1}{4} + \frac{e^{-t}e^{2\tau}}{4\sinh t} = \frac{1}{2\sigma^2}.$$

Eliminate $e^{2\tau}$ and solve for $t$, we have $\sigma^2 = e^t + e^{-t} - 1$, that is,

$$\sigma^2 = 4\sinh^2(t/2), \quad t = 2\,\mathrm{arsinh}\,(\sigma/2)$$

Plug it back,

$$e^{2\tau} = \frac{2\sinh t}{\sigma^2} = \tanh(t/2)^{-1} = \sqrt{1 + \frac{4}{\sigma^2}}$$

So the prefactor simplifies to $e^\tau \sqrt{\tanh(t/2)} = 1$, hence

$$(T_\tau K_t T_\tau^*)(x, y) = \exp\left(-\frac{(x-y)^2}{2\sigma^2}\right), \quad t = 2\operatorname{arsinh}(\sigma/2), \quad \tau = \frac{1}{4}\ln\left(1 + \frac{4}{\sigma^2}\right)$$

The diagonalization of $T_\tau K_t T_\tau^*$ is

$$T_\tau K_t T_\tau^* = \sum_{n=0}^{\infty}(1 - e^{-t})e^{-nt}T_\tau|h_n\rangle\langle h_n|T_\tau^*$$

Thus, the eigensystem is

$$\text{eigensystem}(\mu, K) = \{(1 - e^{-t})e^{-nt}, \; T_\tau|h_n\rangle : n = 0, 1, 2, \dots\}$$

At the $\sigma \to \infty$ limit, we have

$$t = 2\ln\sigma + 2/\sigma^2 + O(\sigma^{-4}), \quad \tau = \sigma^{-2} - 2\sigma^{-4} + O(\sigma^{-6})$$

Now, the Hermite eigensystem corresponding to $e^{-\frac{(x-y)^2}{2\sigma^2}}$ is $\mathcal{HE}(1, (c_n))$ with $c_n = \sigma^{-2n}$, so we just need to check that $\text{eigensystem}(\mu, K) \to \mathcal{HE}(1, (c_n))$, which is true:

$$\frac{(1 - e^{-t})e^{-nt}}{c_n} = 1 - (2n + 2)\sigma^{-2} + O(\sigma^{-4}) \to 1,$$

$$\arccos\langle h_n|T_\tau|h_n\rangle = \sqrt{\frac{n^2 + n + 1}{2}}\sigma^{-2} + O(\sigma^{-4}) \to 0$$

Not only do we see the convergence, we also see that, for any fixed $n$, the rate of convergence is on the order of $n\sigma^{-2} = nc_{\ell+1}/c_\ell$. We will show in the next few sections that this is a generic phenomenon.

In general, if a kernel has coefficients $c_\ell$ decaying at a rate of $\epsilon$, then the $n$-th entry in the kernel's eigensystem converges to the corresponding $n$-th entry in the Hermite eigensystem, at a rate of $O(\epsilon)$. The constant in $O(\epsilon)$ increases with $n$, so that the convergence is not uniform. The higher-order entries converge slower.

## I.2 PART 1 BONUS

We can diagonalize $K(x, y) = e^{xy/\sigma^2}$ in the same way:

$$K(x, y) = (1 - 2/\sigma^2)^{-1/2}K_{t,\tau}(x, y), \quad t = \operatorname{arcosh}(\sigma^2/2), \quad \tau = \frac{1}{4}\ln\left(1 - \frac{4}{\sigma^4}\right)$$

Thus, its eigensystem is

$$\text{eigensystem}(\mu, K) = \{(1 - 2/\sigma^2)^{-1/2}(1 - e^{-t})e^{-nt}, \; T_\tau|h_n\rangle : n = 0, 1, 2, \dots\}$$

At the $\sigma \to \infty$ limit,

$$t = 2\ln\sigma - \sigma^{-4} + O(\sigma^{-8}), \quad \tau = -\sigma^{-4} + O(\sigma^{-8})$$

yielding

$$\frac{(1 - e^{-t})e^{-nt}}{c_n} = 1 + (n + \tfrac{1}{2})\sigma^{-4} + O(\sigma^{-8}),$$

$$\arccos\langle h_n|T_\tau|h_n\rangle = -\sqrt{\frac{n^2 + n + 1}{2}}\sigma^{-4} + O(\sigma^{-8})$$

This case is special, in that the convergence is on the order of $n\sigma^{-4} = n(c_{\ell+1}/c_\ell)^2$, which is faster by one order of magnitude than the generic case.

To understand this, we directly expand the operators. The operator for $\exp(-(x - y)^2/2\sigma^2)$ has expansion

$$K_{\exp(-(x-y)^2/2\sigma^2)} = I + \sigma^{-2}(|h_1\rangle\langle h_1| - M_x^2) + O(\sigma^{-4})$$

where we note that the multiplication operator $M_x$ is self-adjoint, with expression

$$M_x = \sum_n \sqrt{n+1}|h_{n+1}\rangle\langle h_n| + \sqrt{n+1}|h_n\rangle\langle h_{n+1}|$$

Because $M_x^2$ is not diagonal in the Hermite basis, the operator $K_{\exp(-(x-y)^2/2\sigma^2)}$ is not diagonal at the $\sigma^{-2}$ order, and perturbation occurs at that order.

In contrast, the operator for $\exp(xy/\sigma^2)$ has expansion

$$K_{\exp(xy/\sigma^2)} = I + \sigma^{-2}|h_1\rangle\langle h_1| + \sigma^{-4}(|h_2\rangle + 2^{-1/2}|h_0\rangle)(\langle h_2| + 2^{-1/2}\langle h_0|) + O(\sigma^{-6})$$

Therefore, it is not diagonal at the $\sigma^{-4}$ order, and perturbation occurs at that order.

### I.3 PART 2

We take its product, to solve the problem for $L^2(\mu)$, where $\mu = \mathcal{N}(0, I_d)$.

The kernel $K(\boldsymbol{x}, \boldsymbol{y}) = e^{-\frac{\|\boldsymbol{x}-\boldsymbol{y}\|^2}{2\sigma^2}}$ decomposes into a product of kernels:

$$K(\boldsymbol{x}, \boldsymbol{y}) = \prod_{i=1}^{d} e^{-\frac{(x_i - y_i)^2}{2\sigma^2}}$$

Therefore, its kernel operator decomposes into a tensor product of kernel operators:

$$K = \bigotimes_{i=1}^{d} K_i$$

$$= \sum_{\boldsymbol{\alpha} \in \mathbb{N}^d} \bigotimes_{i=1}^{d} (1 - e^{-t}) e^{-\alpha_i t} T_\tau |h_{\alpha_i}\rangle\langle h_{\alpha_i}| T_\tau^*$$

$$= \sum_{\boldsymbol{\alpha} \in \mathbb{N}^d} (1 - e^{-t})^d e^{-|\boldsymbol{\alpha}|t} T_\tau^{\otimes d} |h_{\boldsymbol{\alpha}}\rangle\langle h_{\boldsymbol{\alpha}}| (T_\tau^{\otimes d})^*$$

So, its eigensystem is

$$\{(1 - e^{-t})^d e^{-|\boldsymbol{\alpha}|t}, \ T_\tau^{\otimes d} |h_{\boldsymbol{\alpha}}\rangle : \boldsymbol{\alpha} \in \mathbb{N}^d\}$$

At the $\sigma \to \infty$ limit, using the previous result, each eigenvalue converges as

$$\frac{(1 - e^{-t})^d e^{-|\boldsymbol{\alpha}|t}}{c_{|\boldsymbol{\alpha}|}} = 1 - (2|\boldsymbol{\alpha}| + 2d)\sigma^{-2} + O(\sigma^{-4})$$

and each eigenvector converges as

$$\arccos\langle h_{\boldsymbol{\alpha}} | T_\tau^{\otimes d} | h_{\boldsymbol{\alpha}}\rangle = \sqrt{\frac{\sum_{i=1}^{d}(\alpha_i^2 + \alpha_i + 1)}{2}} \sigma^{-2} + O(\sigma^{-4})$$

### I.4 PART 3

We solve the problem for $L^2(\mu)$, where $\mu = \mathcal{N}(0, \boldsymbol{\Sigma})$, by taking an opportune unitary transformation $V : L^2(\mathcal{N}(0, \boldsymbol{\Sigma})) \to L^2(\mathcal{N}(0, I_d))$.

As previously stated, if $MM^\top = \boldsymbol{\Sigma}$, then $V : L^2(\mathcal{N}(0, \boldsymbol{\Sigma})) \to L^2(\mathcal{N}(0, I_d))$ defined by

$$(Vf)(x) = f(Mx), (V^*f)(x) = f(M^{-1}x)$$

is unitary.

Now, define the operator using the kernel function $K(\boldsymbol{x}, \boldsymbol{y}) = e^{-\frac{\|\boldsymbol{x}-\boldsymbol{y}\|^2}{2\sigma^2}}$. With the unitary transformation, its kernel becomes

$$(VKV^*)(\boldsymbol{x}, \boldsymbol{y}) = K(M\boldsymbol{x}, M\boldsymbol{y}) = e^{-\frac{(\boldsymbol{x}-\boldsymbol{y})^\top M^\top M (\boldsymbol{x}-\boldsymbol{y})}{2\sigma^2}}$$

Therefore, the kernel decomposes if $M^\top M$ is diagonalized. This can be solved by taking the SVD of the covariance matrix as $\boldsymbol{\Sigma} = U\Gamma U^\top$, where $\Gamma = \mathrm{diag}(\gamma_1, \ldots, \gamma_d)$, then we can set

$$M = U\Gamma^{1/2}$$

so that $VKV^*$ has kernel

$$\prod_{i=1}^{d} e^{-\frac{(x_i - y_i)^2}{2(\sigma/\sqrt{\gamma_i})^2}}$$

Thus, $VKV^*$ diagonalizes as

$$VKV^* = \sum_{\boldsymbol{\alpha} \in \mathbb{N}^d} \prod_{i=1}^{d} (1 - e^{-t_i}) e^{-\alpha_i t_i} \otimes_{i=1}^{d} T_{\tau_i} |h_{\alpha_i}\rangle (\otimes_{i=1}^{d} T_{\tau_i} |h_{\alpha_i}\rangle)^*$$

where $t_i, \tau_i$ are defined by

$$t_i = 2\operatorname{arsinh}\left(\sigma/2\sqrt{\gamma_i}\right), \; \tau_i = \frac{1}{4}\ln\left(1 + \frac{4\gamma_i}{\sigma^2}\right)$$

Now, we convert this back to $K$ to obtain

$$K = \sum_{\boldsymbol{\alpha}\in\mathbb{N}^d}\prod_{i=1}^{d}(1 - e^{-t_i})e^{-\alpha_i t_i}V^* \otimes_{i=1}^{d} T_{\tau_i}|h_{\alpha_i}\rangle(V^* \otimes_{i=1}^{d} T_{\tau_i}|h_{\alpha_i}\rangle)^*$$

At the limit of $\sigma \to \infty$, the eigenvalues converge to

$$\prod_{i=1}^{d}(1 - e^{-t_i})e^{-\alpha_i t_i} = \left(\sigma^{-2|\boldsymbol{\alpha}|}\prod_i \gamma_i^{\alpha_i}\right)(1 - (2|\boldsymbol{\alpha}| + 2d)\sigma^{-2} + O(\sigma^{-4}))$$

at a rate of $\sigma^{-2}$.

As before, the eigenvectors converge to $V^*|h_{\boldsymbol{\alpha}}\rangle$ at a rate of $\sqrt{\frac{\sum_{i=1}^{d}(\alpha_i^2+\alpha_i+1)}{2}}\sigma^{-2}$. Since

$$(V^* h_{\boldsymbol{\alpha}})(\boldsymbol{x}) = h_{\boldsymbol{\alpha}}(\Gamma^{-1/2}U^\top \boldsymbol{x}) = h_{\boldsymbol{\alpha}}^{(\boldsymbol{\Sigma})}(\boldsymbol{x})$$

the theorem is proven fully.

Similarly to the case in Appendix I.2, the kernel $K(\boldsymbol{x}, \boldsymbol{y}) = e^{\boldsymbol{x}^\top \boldsymbol{y}/\sigma^2}$ converges to the Hermite eigensystem at a rate of $O(\sigma^{-4})$, one order of magnitude faster.

# J  PROOF OF THEOREM 2 IN ONE DIMENSION

In this appendix, we prove Theorem 2 in the special case of one dimension. The general case is proven in Appendix K. The proof of this case is easier than the general case since there is no multiplicity. The ideas in the proof would be reused in the general case.

Appendix J.1 is a reference sheet quoting several theorems we need for the proofs. The reader can skip it and refer back when needed. Appendix J.2 states Theorem 2 rigorously as Theorem 5, then shows that it is a special case of a more general theorem (Theorem 6). The next section proves the general case. Appendix J.3.1 shows that the eigenvalues of the kernel converge to the desired form with relative error decaying at a rate of $O(\epsilon)$. Appendix J.3.2 leverages this to show that the eigenvectors also converge to the desired form at a rate of $O(\epsilon)$.

## J.1  SETUP

We need to quote some big-name theorems for later use.

We will need to cut up the spectrum of an operator into segments, each falling within an interval. The following theorem allows us to construct tight enough bounds on the intervals. It is a special case of the general Courant–Fischer–Weyl min-max principle, strong enough for our purpose. Our special case avoids the part about essential spectrum, which makes the general statement inconvenient to use.

---

**Theorem 3** (Courant–Fischer–Weyl min-max principle). *Let $A$ be a compact, positive semidefinite operator over a Hilbert space $\mathcal{H}$. Let its eigenvalues be enumerated as $\lambda_1 \geq \lambda_2 \geq \cdots \geq 0$. They can be finite or infinite. Then,*

$$\lambda_k = \max_{\substack{\mathcal{M} \subset \mathcal{H} \\ \dim \mathcal{M} = k}} \min_{\substack{x \in \mathcal{M} \\ \|x\| = 1}} \langle x | A | x \rangle \tag{51}$$

$$= \min_{\substack{\mathcal{M} \subset \mathcal{H} \\ \mathcal{M} = k-1}} \max_{\substack{x \in \mathcal{M}^\perp \\ \|x\| = 1}} \langle x | A | x \rangle \tag{52}$$

---

*Proof.* The second equation is (Teschl, 2009, Theorem 4.10). The first equation is proven by essentially the same technique as the finite-dimensional case.

Let $v_1, v_2, \ldots$ be its eigenvectors. If we set $\mathcal{M}$ to be the span of $\{v_1 | \ldots, v_k\}$, then we have

$$\lambda_k = \min_{\substack{x \in \mathcal{M} \\ \|x\| = 1}} \langle x, Ax \rangle$$

So it remains to prove the other half.

For any subspace with $\dim \mathcal{M} = k$, it must have a nontrivial intersection with $\{v_k, v_{k+1}, \ldots\}$, therefore, there exists some unit vector $x \in \mathcal{M}$, such that it has decomposition $x = \sum_{i \geq k} a_i v_i$. With that, we have

$$\langle x | Ax \rangle = \sum_{i \geq k} |a_i|^2 \lambda_i \leq \sum_{i \geq k} |a_i|^2 \lambda_k = \lambda_k$$

$\square$

The min-max principle has a corollary that we will use, more convenient than quoting the full min-max principle.

---

**Corollary 1** (Cauchy interlacing law). *For any $n \times n$ Hermitian matrix $A_n$ with top left $n-1 \times n-1$ minor $A_{n-1}$, then*

$$\lambda_{i+1}(A_n) \leq \lambda_i(A_{n-1}) \leq \lambda_i(A_n)$$

*for all $1 \leq i < n$. (Tao, 2012, Eq. 1.75)*

---

After bounding the eigenvalues, we will use the second theorem to bound the eigenvectors. However, we need something more, because we need to bound the eigenvector rotations of *segments* of the

spectrum, and the segment may be badly separated within, even though it is well-separated from other segments.

Therefore, we need to handle the rotations of *reducing subspaces*, not eigenspaces. A reducing subspace for an operator $A$ is a closed subspace $V$, such that $A(V) \subset V$. For self-adjoint operators, the reducing subspaces are direct sums of eigenspaces. In particular, the span of an eigenvector is a reducing subspace.

Given a segment of the spectrum, $\Lambda \subset \sigma(A)$, we define $V_\Lambda$ as the reducing space of $\Lambda$. For example, if $\Lambda = \{E\}$, then $V_\Lambda$ is the closed span of all eigenvectors with eigenvalue $E$. We also define $P_\Lambda$ as the orthogonal projector to $V_\Lambda$.

The following theorem shows that, if the operator spectrum split into two parts mutually well-separated by $\Omega(1)$, then their corresponding two reducing subspaces would each only rotate by $O(\epsilon)$ under an operator perturbation by $O(\epsilon)$.

---

**Theorem 4** (Davis–Kahan $\sin\Theta$ theorem (Davis & Kahan, 1970)). *Let $A, B$ be self-adjoint operators such that*

- *$\sigma(A)$ is partitioned into $\Lambda_0, \Lambda_1$*
- *$\sigma(B)$ is partitioned into $\Gamma_0, \Gamma_1$*
- *The spaces $V_{\Lambda_0}, V_{\Gamma_0}$ are of the same dimension. Similarly for $V_{\Lambda_1}, V_{\Gamma_1}$.*
- *$\Lambda_0$ is contained in an interval $[x, y]$.*
- *$\Gamma_1$ is disjoint from the enlarged interval $(x - \delta, y + \delta)$.*

*Then there exists a self-adjoint "angle" operator $\Theta$, such that the rotation operator*
$\begin{bmatrix} \cos\Theta & \sin\Theta \\ -\sin\Theta & \cos\Theta \end{bmatrix}$ *rotates $V_{\Lambda_i}$ to $V_{\Gamma_i}$ for $i = 0, 1$. Furthermore, the angle operator satisfies*

$$\|\sin\Theta\| \leq \tfrac{1}{\delta} \|A - B\| \tag{53}$$

*for any unitarily invariant norm $\|\cdot\|$.*

---

Intuitively, the operator $\Theta$ is just a diagonal matrix of angles, in a suitable orthonormal basis. The operator $\begin{bmatrix} \cos\Theta & \sin\Theta \\ -\sin\Theta & \cos\Theta \end{bmatrix}$ performs simultaneous rotation in many (potentially infinitely many) 2-dimensional planes, such that the two reducing subspaces of $A$ are rotated to two reducing subspaces of $B$.

## J.2 REDUCTION TO A GENERAL CASE

We clean up the form of Theorem 2 by performing a few WLOGs and reductions into Theorem 5, so that we can deduce the theorem as a corollary of a more general Theorem 6. Alternatively, the theorem can be proven by proving the more general, multidimensional case. This is done in Appendix K. However, it may be worthwhile to study the following special case before reading the more general case, since most of the proof ideas are present.

We begin with a convenient definition.

**Definition 5** (fast-decay). A real-valued sequence $c_0, c_1, \dots$ is **fast-decaying** iff there exists a number $\epsilon \in [0, 1)$, such that $c_{n+1} \leq \epsilon c_n$ for all $n \in \mathbb{N}$.

As in the proof of Theorem 1 in Appendix I, we can perform a unitary transform of $L^2(\mathcal{N}(0, \gamma))$ to $L^2(\mathcal{N}(0, 1))$.

Next, we define $a_n = \frac{(2n-1)!!}{n!} c_n$, so that

$$K = \sum_n a_n |v_n\rangle\langle v_n|$$

By Stirling's approximation,

$$\frac{(2n-1)!!}{n!} = \frac{1}{2^n}\binom{2n}{n} = \frac{2^n}{\sqrt{\pi n}}(1 + O(1/n))$$

Therefore, if $c_n$ is fast-decaying with parameter $\epsilon$, then $a_n$ is fast-decaying with parameters $2\epsilon, \delta_{low,n}/\sqrt{n}$. So, we can study the kernel $\sum_n a_n|v_n\rangle\langle v_n|$, with the fast-decaying condition directly imposed on $a_n$. This makes the notation cleaner.

Because $\langle v_n|h_n\rangle = \sqrt{\frac{n!}{(2n-1)!!}}$ (see Appendix H.8), convergence to $c_n$ is equivalent to convergence to $a_n|\langle v_n|h_n\rangle|^2$.

With that, we can we restate the theorem we wish to prove, more rigorously:

---

**Theorem 5** (The HEA holds for a fast-decaying kernel on 1D Gaussian measure). *Let $\mu = \mathcal{N}(0,1)$ be the standard Gaussian measure, and let*

$$K = \sum_{n=0}^{\infty} a_n|v_n\rangle\langle v_n|$$

*be a dot-product kernel with fast-decaying coefficients $a_n$ with parameters $\epsilon$, then for any $n \in \mathbb{N}$, there exists an eigensystem of $K$, written as $(\lambda_0(K), |v_0(K)\rangle), (\lambda_1(K), |v_1(K)\rangle), \ldots$, such that*

$$\frac{\lambda_n(K)}{a_n|\langle v_n|h_n\rangle|^2} = 1 + O(\epsilon)$$
$$|\angle(|v_n(K)\rangle, |h_n\rangle)| = O(\epsilon)$$

*as $\epsilon \to 0$. Furthermore, the scaling factor in $O(\epsilon)$ depends only on the Gram matrix of the vectors $|v_0\rangle, \ldots, |v_n\rangle$.*

---

The above statement may seem oddly convoluted with "for any $n \in \mathbb{N}$, there exists...", but this is necessary, because the rate of convergence is not uniform over the sequence. In general, higher-order eigenvectors converge slower than lower-order eigenvectors, which means the constant in the $O(\epsilon)$ terms is larger for larger $n$, and no uniform-convergence rate exists.

The scaling factor in $O(\epsilon)$ measures the speed of convergence for the $n$-th eigenpair. It is slower if the Gram matrix of the vectors $|v_0\rangle, \ldots, |v_n\rangle$ is ill-conditioned, because in this case, the vectors are almost linearly dependent.

Indeed, the Hankel moment matrices of most commonly used probability distributions, including the uniform distribution, the gaussian distribution, and the exponential distribution, are exponentially ill-conditioned (Chen & Lawrence, 1999). Therefore, we should expect the constant in $O(\epsilon)$ to grow exponentially with $n$.

Also, take note of the phrasing "there exists an eigensystem of $K$ denoted as $(\lambda_k(K), |v_k(K)\rangle)_k$", because we allow $K$ to suffer multiplicity. In these cases, the corresponding eigenspace would have more than 1 dimension, and therefore there is freedom in choosing any orthonormal basis of the eigenspace as "the eigenvectors". The theorem states that, despite the multiplicity, there exists a good choice of eigenvectors, such that they make a small angle with the canonical eigenvectors $|\hat{v}_n\rangle$. This will become especially relevant in the proof of the multidimensional generalization in Appendix K.

Because the orthonormal basis $|h_0\rangle, |h_1\rangle, \ldots$ is obtained by Gram–Schmidt orthonormalization on $|v_0\rangle, |v_1\rangle, \ldots$, we can generalize it, this time with all epsilons and deltas in place for maximal rigor:

---

**Theorem 6.** *Let:*

1. *$|v_0\rangle, |v_1\rangle, \ldots$ be a sequence of linearly independent unit vectors in a Hilbert space, such that their closed span is the whole space;*
2. *$|\hat{v}_0\rangle, |\hat{v}_1\rangle, \ldots$ be the sequence obtained by performing Gram–Schmidt process on the sequence;*

*Then, there exists a sequence of constants $C_0, C_1, \ldots, > 0$ and $\epsilon_0, \epsilon_1, \cdots > 0$, such that for all $n \in \mathbb{N}$, $\epsilon \in [0, \epsilon_n]$, and all $a_0, a_1, \ldots$ fast-decaying sequences with parameters $\epsilon$, there exists an eigensystem of the kernel $K := \sum_{n=0}^{\infty} a_n |v_n\rangle\langle v_n|$, denoted as $(\lambda_k(K), |v_k(K)\rangle)_k$, such that*

$$\lambda_n(K) \in a_n |\langle \hat{v}_n | v_n \rangle|^2 (1 \pm C_n \epsilon)$$

*and*

$$|\angle(|v_n(K)\rangle, |\hat{v}_n\rangle)| \leq C_n \epsilon$$

*Furthermore, the scaling factor $C_n$ and the bound $\epsilon_n$ depend on only on the Gram matrix of the vectors $|v_0\rangle, \ldots, |v_n\rangle$.*

---

Intuitively restated, the theorem says that as $\epsilon \to 0$, the eigensystem of the fast-decaying kernel $\sum_{n=0}^{\infty} a_n |v_n\rangle\langle v_n|$ rotates towards the canonical eigensystem at a rate of $\epsilon$.

The proof has two parts. The first part uses the min-max principle and lowest-order operator perturbation (commonly used in quantum mechanics), to segment the spectrum of $K$ into small intervals of the form

$$\lambda_n(K) \in a_n |\langle \hat{v}_n | v_n \rangle|^2 (1 + O(\epsilon))$$

In particular, since $a_n$ are fast-decaying, it shows that the eigenvalues are exponentially separated.

The second part applies Davis–Kahan twice, using this exponential separation of eigenvalues, to show that $|\sin \angle(|v_n(K)\rangle, |\tilde{K}\rangle)| = O(\epsilon)$ and $|\sin \angle(|v_n(\tilde{K})\rangle, |\hat{v}_n\rangle)| = O(\epsilon)$, for a cleverly-chosen operator $\tilde{K}$.

Before we launch into the proof, we should look at a simple case that explains why this should be true.

Consider the case of two dimensions. We only have $|v_0\rangle, |v_1\rangle$. In this case, the kernel $K = a_0 |v_0\rangle\langle v_0| + a_1 |v_1\rangle\langle v_1|$. Diagonalizing the kernel is equivalent to finding the major and minor axes of the contour ellipse defined by $\{|x\rangle : \langle x|K|x\rangle = 1\}$. This ellipse is the unique ellipse tangent to the 4 lines defined by $a_0 |\langle v_0|x\rangle|^2 = 1$, $a_1 |\langle v_1|x\rangle|^2 = 1$.

Suppose we fix $a_0$, and let $a_1 \to 0$. Then, the lines of $a_0 |\langle v_0|x\rangle|^2 = 1$ remain constant, but the lines of $a_1 |\langle v_1|x\rangle|^2 = 1$ diverge to infinity. The ellipse degenerates to two parallel lines. Its minor semiaxis rotates to become perpendicular to the two parallel lines, i.e. parallel to $|v_0\rangle$. Therefore, the eigenpair converges to $(a_0 |\langle \hat{v}_0|v_0\rangle|^2, |\hat{v}_0\rangle)$.

Suppose we fix $a_1$ and let $a_0 \to \infty$. Then, the lines of $a_1 |\langle v_1|x\rangle|^2 = 1$ remain constant, but the lines of $a_0 |\langle v_0|x\rangle|^2 = 1$ converge to the origin. The ellipse degenerates to two line segments. Its major semiaxis rotates to become the same as that line segment, i.e. parallel to $a_0 |\langle v_0|x\rangle|^2 = 1$, i.e. perpendicular to $|v_0\rangle$. Therefore, the eigenpair converges to $(a_1 |\langle \hat{v}_1|v_1\rangle|^2, |\hat{v}_1\rangle)$.

Intuitively, we see that for a given $n$, the effect of all $a_{n+1}, a_{n+2}, \ldots$ terms in the kernel is a small perturbation on the $n$-th eigenspace, and negligible because the parallel planes diverge to infinity. The effect of all $a_{n-1}, a_{n-2}, \ldots, a_0$ is a large but *fixed* perturbation, forcing the $n$-th eigenspace to be perpendicular to all of $|v_{n-1}\rangle, \ldots, |v_0\rangle$, but once that is done, their effects are also negligible because the parallel planes converge to the origin.

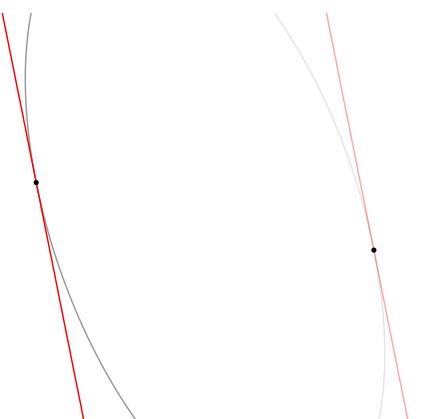
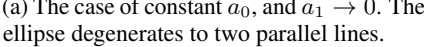
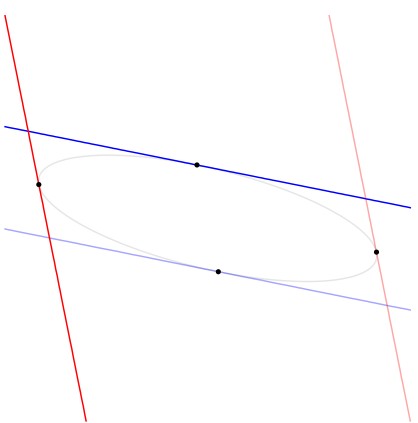

(a) The case of constant $a_0$, and $a_1 \to 0$. The ellipse degenerates to two parallel lines.

(b) The case of constant $a_1$, and $a_0 \to \infty$. The ellipse degenerates to two repeated line segments.

Figure 22: Diagonalizing the kernel in 2 dimensions at the $a_1/a_0 \to 0$ limit.

### J.3 PROOF OF THEOREM 6

#### J.3.1 PART 1

To show: $\lambda_n(K) = a_n|\langle \hat{v}_n|v_n\rangle|^2(1 + O(\epsilon))$.

*Proof.* If $a_n = 0$, then all $\lambda_n, \lambda_{n+1}, \cdots = 0$, and the kernel $K$ becomes finite-ranked, with range $\mathrm{Span}\left(|v_{0:n-1}\rangle\right)$. So the theorem becomes is trivial. Otherwise, we assume $a_n > 0$, which means that all $a_0, \ldots, a_n > 0$.

We apply the min-max principle to obtain an upper bound of the form $\lambda_n(K) \le a_n|\langle \hat{v}_n|v_n\rangle|^2(1 + O(\epsilon))$ and a lower bound of the form $\lambda_n(K) \ge a_n|\langle \hat{v}_n|v_n\rangle|^2(1 + O(\epsilon))$, thus completing the estimate.

For the upper bound, we use $V = \mathrm{Span}(|v_{0:n-1}\rangle)$, then

$$\lambda_n \le \sup_{\substack{x\in\mathrm{Span}(|v_{0:n-1}\rangle)^\perp \\ \|x\|=1}} \langle x|K|x\rangle$$

$$= \sup_{\substack{x\in\mathrm{Span}(|v_{0:n-1}\rangle)^\perp \\ \|x\|=1}} \sum_{k=0}^{\infty} a_k|\langle x|v_k\rangle|^2$$

$$= \sup_{\substack{x\in\mathrm{Span}(|\hat{v}_{n:\infty}\rangle) \\ \|x\|=1}} \sum_{k=n}^{\infty} a_k|\langle x|v_k\rangle|^2$$

$$\le \sup_{\substack{x\in\mathrm{Span}(|\hat{v}_{n:\infty}\rangle) \\ \|x\|=1}} a_n|\langle x|v_n\rangle|^2 + \sup_{\substack{x\in\mathrm{Span}(|\hat{v}_{n:\infty}\rangle) \\ \|x\|=1}} \sum_{k=n+1}^{\infty} a_k|\langle x|v_k\rangle|^2$$

$$= a_n|\langle \hat{v}_n|v_n\rangle|^2 + \lambda_{\max}\left(\sum_{k=n+1}^{\infty} a_k|v_k\rangle\langle v_k|\right)$$

$$\le a_n|\langle \hat{v}_n|v_n\rangle|^2 + \mathrm{Tr}\left(\sum_{k=n+1}^{\infty} a_k|v_k\rangle\langle v_k|\right)$$

$$= a_n|\langle \hat{v}_n|v_n\rangle|^2 + \sum_{k=n+1}^{\infty} a_k$$

$$= a_n|\langle \hat{v}_n|v_n\rangle|^2 + a_n O(\epsilon)$$

$$= a_n|\langle \hat{v}_n|v_n\rangle|^2(1 + O(\epsilon))$$

where the step $\lambda_{\max} \leq \text{Tr}$ is because all $a_k \geq 0$.

Though unnecessary, we can concretely write down the upper bound as

$$\lambda_n \leq a_n |\langle \hat{v}_n | v_n \rangle|^2 (1 + C\epsilon)$$

where

$$C = \frac{1}{|\langle \hat{v}_n | v_n \rangle|^2} \sum_{k=n+1}^{\infty} \frac{a_k}{a_n \epsilon} \leq \frac{1}{|\langle \hat{v}_n | v_n \rangle|^2} \frac{1}{1 - \epsilon}$$

For the lower bound, we use $V = \text{Span}(|v_{0:n}\rangle)$, then

$$\lambda_n \geq \inf_{\substack{x \in \text{Span}(|v_{0:n}\rangle) \\ \|x\| = 1}} \langle x | K | x \rangle$$

$$= \inf_{\substack{x \in \text{Span}(|v_{0:n}\rangle) \\ \|x\| = 1}} \sum_{k=0}^{\infty} a_k |\langle x | v_k \rangle|^2$$

$$\geq \inf_{\substack{x \in \text{Span}(|v_{0:n}\rangle) \\ \|x\| = 1}} \sum_{k=0}^{n} a_k |\langle x | v_k \rangle|^2$$

The quantity is a standard problem in quadratic programming, with exact solution $\lambda_{\min}(G^{1/2} A G^{1/2})$, where $A = \text{diag}(a_0, \ldots, a_n)$ and $G = (\langle v_i | v_j \rangle)_{i,j=0}^n$ is the Gram matrix.

To see this, write $x = \sum_{j \in 0:n} c_j v_j$, then

$$\|x\|^2 = c^T G c, \qquad \sum_{k=0}^{n} a_k |\langle x | v_k \rangle|^2 = (Gc)^T A (Gc) = c^T G A G c.$$

So the constrained minimum of $\frac{c^T G A G c}{c^T G c}$ equals the smallest eigenvalue of $G^{1/2} A G^{1/2}$ by the Rayleigh–Ritz principle.

Let $M = G^{1/2} A G^{1/2}$. We invert the matrix to use standard operator perturbation theory:

$$M^{-1} = \frac{1}{a_n} u_n u_n^T + \sum_{k=0}^{n-1} \frac{1}{a_k} u_k u_k^T$$

where $u_k = G^{-1/2} e_k$ is the $k$-th column of $G^{-1/2}$. The perturbation $\sum_{k=0}^{n-1} \frac{1}{a_k} u_k u_k^T$ is order $O(\epsilon)$ compared to the unperturbed part $\frac{1}{a_n} u_n u_n^T$.

The unperturbed operator has maximal eigenvalue $\frac{\|u_n\|^2}{a_n}$ with eigenvector $\hat{u}_n := u_n / \|u_n\|$. The perturbed operator has maximal eigenvalue:

$$\frac{\|u_n\|^2}{a_n} + \hat{u}_n^T \left( \sum_{k=0}^{n-1} \frac{1}{a_k} u_k u_k^T \right) \hat{u}_n + O(\epsilon^2)$$

Inverting the eigenvalue,

$$\lambda_{\min}(G^{1/2} A G^{1/2}) = \frac{a_n}{\|u_n\|^2} \left( 1 - \frac{a_n / a_{n-1}}{\|u_n\|^2} |u_{n-1}^T \hat{u}_n|^2 + O(\epsilon^2) \right) = \frac{a_n}{\|u_n\|^2} (1 + O(\epsilon))$$

Now, $\|u_n\|^2 = e_n^T G^{-1} e_n$, and $G^{-1}$ is the Gram matrix of the dual basis of $|v_{0:n}\rangle$ in $\text{Span}(|v_{0:n}\rangle)$. In particular, because $|\hat{v}_n\rangle$ is perpendicular to all of $|v_0\rangle, \ldots, |v_{n-1}\rangle$, the $n$-th dual vector is just $\frac{|\hat{v}_n\rangle}{\langle v_n | \hat{v}_n \rangle}$.

Therefore $\|u_n\|^2 = \left\| \frac{|\hat{v}_n\rangle}{\langle v_n | \hat{v}_n \rangle} \right\|^2 = \frac{1}{|\langle v_n | \hat{v}_n \rangle|^2}$, and we obtain the desired lower bound

$$\lambda_n \geq a_n |\langle \hat{v}_n | v_n \rangle|^2 (1 + O(\epsilon))$$

$\square$

Some comments on the constants in $O(\epsilon)$.

In the above proof, we constructed an upper bound $a_n|\langle \hat{v}_n|v_n\rangle|^2(1 + O(\epsilon))$ and a lower bound $a_n|\langle \hat{v}_n|v_n\rangle|^2(1 - O(\epsilon))$.

The constant in the upper bound is

$$\frac{1}{|\langle \hat{v}_n|v_n\rangle|^2}\frac{1}{1 - \epsilon}$$

which depends on $|\langle \hat{v}_n|v_n\rangle|^2 = \sin^2\theta$, where $\theta$ is the angle between $|v_n\rangle$ and $\mathrm{Span}(|v_{0:n-1}\rangle)$. We see that the bound is pushed wider when either the coefficients become less strictly exponentially-decaying, or the vector $|v_n\rangle$ leans into $\mathrm{Span}(|v_{0:n-1}\rangle)$, and thus becomes less orthonormal.

The constant in the lower bound is

$$\frac{1}{\|u_n\|^2}|u_{n-1}^T\hat{u}_n|^2 = \frac{1}{\|u_n\|^4}|u_{n-1}^T u_n|^2$$
$$= \langle v_n|\hat{v}_n\rangle|^4|u_{n-1}^T u_n|^2$$
$$= |\langle v_n|\hat{v}_n\rangle|^4(G_{n-1,n}^{-1})^2$$

Similarly to the previous case, $G_{n-1,n}^{-1}$ gets larger as the vectors $|v_0\rangle, \ldots, |v_n\rangle$ get less orthonormal, which worsens eigenvalue convergence.

### J.3.2 PART 2

*Proof.* If $a_n = 0$, then we can trivially select the $n$-th eigenvector to be $|\hat{v}_n\rangle$. Otherwise, we have $a_0, \ldots, a_n > 0$.

By the eigenvalue bound (that is, Part 1 of the theorem), the spectral gap around $\lambda_n(K)$ is

$$\min_{j \mathbb{N}eqn}|\lambda_n(K) - \lambda_j(K)| = a_n|\langle \hat{v}_n|v_n\rangle|^2(1 + O(\epsilon))$$

Define the truncated operator $\tilde{K} = \sum_{k=0}^n a_k|v_k\rangle\langle v_k|$. By Davis–Kahan,

$$|\sin\angle(v_n(K), v_n(\tilde{K}))| \le \frac{2}{a_n|\langle \hat{v}_n|v_n\rangle|^2(1 + O(\epsilon))}\|K - \tilde{K}\|_{op}$$
$$\le \frac{2}{a_n|\langle \hat{v}_n|v_n\rangle|^2(1 + O(\epsilon))}\mathrm{Tr}\left[K - \tilde{K}\right]$$
$$= \frac{2}{a_n|\langle \hat{v}_n|v_n\rangle|^2(1 + O(\epsilon))}\sum_{k=n+1}^{\infty} a_k$$
$$= O(\epsilon)$$

Thus, we need only bound $|\sin\angle(v_n(\tilde{K}), \hat{v}_n)|$.

Because $\tilde{K}$ lives inside $\mathrm{Span}(|v_{0:n}\rangle)$, we thenceforth restrict the Hilbert space to just $\mathrm{Span}(|v_{0:n}\rangle)$.

Define the twice-truncated operator $\bar{K} = \sum_{k=0}^{n-1} a_k|v_k\rangle\langle v_k|$. The eigenvalue bound applies to its first $n$ eigenvalues, and its $(n + 1)$-th eigenstate is the ground state, with eigenvalue 0 and eigenvector $|\hat{v}_n\rangle$.

Thus, the spectral gap around its ground state eigenvalue is

$$\min_{j \mathbb{N}eqn}|\lambda_n(\bar{K}) - \lambda_j(\bar{K})| = \lambda_{n-1}(\bar{K}) = a_{n-1}|\langle \hat{v}_{n-1}|v_{n-1}\rangle|^2(1 + O(\epsilon))$$

By Davis–Kahan again,

$$|\sin\angle(v_n(\tilde{K}), \hat{v}_n)| = |\sin\angle(v_n(\tilde{K}), v_n(\bar{K}))|$$
$$\le \frac{2}{a_{n-1}|\langle \hat{v}_{n-1}|v_{n-1}\rangle|^2(1 + O(\epsilon))}\|\tilde{K} - \bar{K}\|_{op}$$
$$= \frac{2}{a_{n-1}|\langle \hat{v}_{n-1}|v_{n-1}\rangle|^2(1 + O(\epsilon))}a_n$$
$$= O(\epsilon)$$

$\square$

There are two occurrences of $O(\epsilon)$ in the proof. Both can be bounded explicitly, to show that they only depend on the Gram matrix of $|v_0\rangle, \ldots, |v_n\rangle$, as in Part 1.

The first $O(\epsilon)$ has explicit upper bound constant

$$\frac{2}{a_n \epsilon |\langle \hat{v}_n | v_n \rangle|^2 (1 + O(\epsilon))} \sum_{k=n+1}^{\infty} a_k \leq \frac{2}{|\langle \hat{v}_n | v_n \rangle|^2} \frac{1}{1 - \epsilon} \frac{1}{1 + O(\epsilon)}$$

where the remaining $O(\epsilon)$ in $\frac{1}{1 + O(\epsilon)}$ came from Part 1, which as we showed in Part 1, only depends on the Gram matrix of $|v_0\rangle, \ldots, |v_n\rangle$.

The second $O(\epsilon)$ has explicit upper bound constant

$$\frac{2}{|\langle \hat{v}_{n-1} | v_{n-1} \rangle|^2 (1 + O(\epsilon))}$$

where the remaining $O(\epsilon)$ only depends on the Gram matrix of $|v_0\rangle, \ldots, |v_{n-1}\rangle$.

## K    PROOF OF THEOREM 2 IN THE GENERAL CASE

In this appendix, we prove Theorem 2 completely. The general idea is to modify the proof given in Appendix J to account for multiplicity.

Appendix K.1 presents the overall plan of the proof. Appendix K.2 sets up all the machinery needed to handle multiplicity in eigenvalues and eigenvectors, which is a new occurrence in multiple dimensions. Appendix K.3 states the theorem in full rigor as Theorem 7. The next two subsections prove two lemmas that apply to generic operators, not just an integral kernel operator: Appendix K.4 shows that the eigenvalues of a generic fast-decaying kernel splits into segments that are exponentially separated, and Appendix K.5 sharpens this separation into proving that the eigenvalues in the $N$-th segment are only slightly perturbed by all the other segments. Appendix K.6 specializes to the case of a dot-product kernel, showing convergence of the eigenvalues, and then leverages that convergence into the convergence of eigenspaces.

### K.1    PLAN OF THE PROOF

Let $K(\boldsymbol{x}, \boldsymbol{y}) = \sum_{n=0}^{\infty} \frac{c_n}{n!} (\boldsymbol{x}^\top \boldsymbol{y})^n$ be a dot-product kernel with fast-decaying coefficients $c_n$.

As in Appendix I.4, to study a spherically symmetric dot-product kernel over a nonstandard Gaussian distribution $\mathcal{N}(0, \boldsymbol{\Sigma})$, we construct a whitening unitary transform $V : L^2(\mathcal{N}(0, \boldsymbol{\Sigma})) \to L^2(\mathcal{N}(0, I_d))$, thus converting the problem to solving for the eigenstructure of a spherically asymmetric kernel over the standard Gaussian distribution.

Let the SVD of $\boldsymbol{\Sigma}$ be $U\Gamma U^\top$, where $\Gamma = \operatorname{diag}(\gamma_1, \ldots, \gamma_d)$ with $\gamma_1, \ldots, \gamma_d \geq 0$. Define $V$ by $(Vf)(x) = f(Mx)$. This then converts the operator $K$ to $VKV^*$. The operator $VKV^*$ is a kernel operator with kernel function satisfying

$$(VKV^*)(\boldsymbol{x}, \boldsymbol{y}) = \sum_{n=0}^{\infty} \frac{c_n}{n!} (\gamma_1 x_1 y_1 + \cdots + \gamma_d x_d y_d)^n$$

We will prove that the eigensystem of $VKV^*$ converges to

$$\left\{ \left( c_{|\boldsymbol{\alpha}|} \prod_{i=1}^{d} \gamma_i^{\alpha_i}, |h_{\boldsymbol{\alpha}}\rangle \right) \text{ for all } \boldsymbol{\alpha} \in \mathbb{N}_0^d \right\}$$

as $\epsilon \to 0$. Then, by reversing the $V$ transform, we find that the eigensystem of $K$ converges to

$$\left\{ \left( c_{|\boldsymbol{\alpha}|} \prod_{i=1}^{d} \gamma_i^{\alpha_i}, |h_{\boldsymbol{\alpha}}^{(\boldsymbol{\Sigma})}\rangle \right) \text{ for all } \boldsymbol{\alpha} \in \mathbb{N}_0^d \right\}$$

as desired.

We eliminate a pesky special case: one or more of $\gamma_1, \ldots, \gamma_d$ may be equal to zero. In this case, $VKV^*$ may not be positive definite, but merely positive semidefinite, which is annoying. For example, what if $\gamma_2 = \gamma_4 = 0$? Then the operator $VKV^*$ splits to two halves. It is the zero operator on $\operatorname{Span}(e_2, e_4)$, and it is positive definite on $\operatorname{Span}(e_1, e_3, e_5, \ldots, e_d)$. Then we can separately prove the eigensystem convergence on the two halves, and take their tensor product. The case of zero operator is obviously trivial, since its eigensystem is just

$$\left\{ \left( 0, |h_{(\alpha_2, \alpha_4))}\rangle \right) \text{ for all } (\alpha_2, \alpha_4) \in \mathbb{N}_0^2 \right\}$$

Thus, WLOG, we need only consider the case where $\gamma_1, \ldots, \gamma_d > 0$.

We note that, at least in *one* case, the theorem has been proven: If $c_n = \sigma^{-2n}$ for some $\sigma$, then it is just a minor variant of Theorem 1, which has been proven in Appendix I.2.

Thus, if we prove that the difference between the eigensystem of $VKV^*$ and the eigensystem of $ce^{\boldsymbol{x}^\top \boldsymbol{y}/\sigma^2}$ vanishes at $\epsilon \to 0$, for some well-chosen values of $\sigma, c$, we are done.

This cannot be done directly, once again due to nonuniform convergence: The higher-order parts of the eigensystem is wilder, and harder to control. To bypass this difficulty, we divide and conquer.

We prove that the eigensystem of $VKV^*$ is "segmented" into exponentially separated intervals, such that each segment is $\epsilon$-insensitive to perturbations in all other segments. This allows us to show

that, for any fixed $n \in \mathbb{N}_0$, the $n$-th segment of eigensystem$(VKV^*)$ – corresponding to the term $c_n(\boldsymbol{x}^\top \Gamma \boldsymbol{y})^n$ – converges to the $n$-th segment of eigensystem$(K_{ce^{\epsilon(\boldsymbol{x}^\top \Gamma \boldsymbol{y})}})$, where $c = c_n \epsilon^{-n}$. Since the $n$-th segment of eigensystem$(K_{ce^{\epsilon(\boldsymbol{x}^\top \Gamma \boldsymbol{y})}})$ converges to

$$\left\{ \left( \frac{c_n}{\epsilon^n} e^{|\boldsymbol{\alpha}|} \prod_{i=1}^d \gamma_i^{\alpha_i}, |h_{\boldsymbol{\alpha}}\rangle \right) \text{ for all } \boldsymbol{\alpha} \in \mathbb{N}_0^d, \ |\boldsymbol{\alpha}| = n \right\}$$

the theorem is proven.

### K.2 MACHINERY FOR MULTIPLICITY

The main difficulty, compared to the one-dimensional case, is that we must directly handle the multiplicity of eigensystems. By this, we mean that in $\mathbb{R}^d$, the Hermite basis is no longer of form $\{|h_n\rangle\}_{n \in \mathbb{N}}$, but rather, $\{|h_{\boldsymbol{\alpha}}\rangle\}_{\boldsymbol{\alpha} \in \mathbb{N}^d}$. This creates degeneracy, that is, if $\sum_i \alpha_i = \sum_i \alpha_i'$, then $|h_{\boldsymbol{\alpha}}\rangle, |h_{\boldsymbol{\alpha}'}\rangle$ belong to the same eigenspace.

Concretely, define the Ornstein–Uhlenbeck operator $\nabla^2 - x \cdot \nabla$ on $L^2(\mu_d)$. In the $d = 1$ case, its eigenvalues have no multiplicity, and its eigenvectors are precisely the normalized Hermite polynomials $\{|h_n\rangle\}_{n \in 0:\infty}$. In the $d > 1$ case, its eigenvalues suffer multiplicity. Its $n$-th eigenspace is $\{|h_{\boldsymbol{\alpha}}\rangle : |\boldsymbol{\alpha}| = n\}_{\boldsymbol{\alpha} \in \mathbb{N}^d}$, with $\binom{n+d-1}{d-1}$ dimensions.

This multiplicity is inescapable, because the Ornstein–Uhlenbeck operator is spherically symmetric, and spherical symmetry inevitably leads to multiplicity. In our case, the dot-product kernel $\sum_n \frac{c_n}{n!}(\boldsymbol{x}^\top \Gamma \boldsymbol{y})^n$ may have some entries of $\Gamma$ equal, which leads to (partial) spherical symmetry, and thus multiplicity.

As a more famous example, the spherical harmonics in $L^2(\mathbb{R}^d)$ are the eigenvectors of the Laplacian operator $\nabla^2$. Due to spherical symmetry of the operator, its eigenvalues have multiplicity, thus it splits $L^2(\mathbb{R}^d)$ into eigenspaces. The $n$-th eigenspace is spanned by $\frac{(2n+d-2)(n+d-3)!}{n!\,(d-2)!}$ degree-$n$ homogeneous polynomials.

Let's consider the prototypical case that we want to study: the convergence of dot-product kernels to the Hermite eigensystem. In $\mathbb{R}^d$, there are $\binom{n+d-1}{d-1}$ degree-$n$ monomials, and $\binom{n+d-1}{d-1}$ degree-$n$ Hermite polynomials. To obtain the Hermite polynomials, we cannot simply apply the Gram–Schmidt process on the monomials individually. Instead, we need to apply the Gram–Schmidt process simultaneously on each segment of $\binom{n+d-1}{d-1}$ monomials, to obtain the dimension-$\binom{n+d-1}{d-1}$ subspace spanned by the degree-$n$ Hermite polynomials.

To count the multiplicity, we use a function $m : \mathbb{N} \to \mathbb{N} \cup \{\infty\}$. It should be interpreted as saying that the $n$-th eigenspace has dimension $m(n)$. For example, the multiplicity counting function for the Ornstein–Uhlenbeck operator over $L^2(\mu_d)$ is $m(n) = \binom{n+d-1}{d-1}$.

Note that $m$ does not need to be strictly positive. That is, we allow $m(k) = 0$ for some $k$. We even allow $\sum_k m(k)$ to be finite, in the case that the Hilbert space under consideration is finite-dimensional, although we require $\sum_k m(k) > 0$, for otherwise it would be completely trivial.

For convenience, we will from now on assume that $m(k) > 0$ for all $k$, since we do not need more generality. The reader who needs this generality can read the next few sections and mentally generalize the constructions.

The most important condition on the multiplicity counting function is:

**Definition 6** (polynomially bounded multiplicity). A multiplicity counting function is **polynomially bounded** iff there exists some $A, d > 0$ such that $m(n) < An^d$ for all $n \in \mathbb{N}$.

This is satisfied by the Hermite basis in any dimension, since its $m(n) = O(n^{d-1})$.

Given a multiplicity counting function, we define a system of vectors that it counts:

**Definition 7** (vector systems with multiplicity). Given a multiplicity counting function $m$, a vector system with multiplicity $m$ is an indexed set of vectors of form $\{v_{k,l} : k \in \mathbb{N}, l \in [1 : m(k)]\}$.

As in the last few sections, we will only consider vector systems that consists of linearly independent unit vectors, and such that the closure of their span is the entire Hilbert space.

Similarly, we define a system of coefficient it counts:

**Definition 8** (coefficient systems with multiplicity). Given a multiplicity counting function $m$, a coefficient system with multiplicity $m$ is an indexed set of real numbers of form $\{v_{k,l} : k \in \mathbb{N}, l \in [1 : m(k)]\}$.

We next generalize the Gram–Schmidt process to handle multiplicity.

**Definition 9** (Gram–Schmidt process on vector systems). Given a multiplicity counting function $m$, and a linearly independent vector system $\{v_{k,l} : k \in \mathbb{N}, l \in [1 : m(k)]\}$, we define the Gram–Schmidt process on the vector system by the following algorithm:

1. Construct an arbitrary orthonormal basis of $\mathrm{Span}(|v_{0,1}\rangle, \ldots, |v_{0,m(0)}\rangle)$. Call them $|\hat{v}_{0,1}\rangle, \ldots, |\hat{v}_{0,m(0)}\rangle$.

2. Select the next smallest $k$ such that $m(k) > 0$, and project each of $|v_{k,1}\rangle, \ldots, |v_{k,m(k)}\rangle$ to $\mathrm{Span}\left(|\hat{v}_{0,1}\rangle, \ldots, |\hat{v}_{0,m(0)}\rangle\right)^{\perp}$, to obtain $|v'_{k,1}\rangle, \ldots, |v'_{k,m(k)}\rangle$ then construct an arbitrary orthonormal basis of their span. Call them $|\hat{v}_{k,1}\rangle, \ldots, |\hat{v}_{k,m(k)}\rangle$.

3. Continue this way inductively.

Note that the Gram–Schmidt process is not uniquely defined, due to the steps where arbitrary orthonormal bases are chosen. However, it constructs a sequence of subspaces $\{\mathrm{Span}\left(|\hat{v}_{k,1}\rangle, \ldots, |\hat{v}_{k,m(k)}\rangle\right)\}_{k \in 0:\infty}$, which *are* uniquely defined.

Also note that even the traditional Gram–Schmidt process, without multiplicity, still is not uniquely defined, because each $\hat{v}_k$ could have been $-\hat{v}_k$ instead. That is, there is a $\{-1, +1\}$ ambiguity per step. Now, note that $\{-1, +1\}$ is just $O(1)$, the orthogonal group on $\mathbb{R}^1$, and we see that it is a general phenomenon: In general, the Gram–Schmidt process with multiplicity $m$ creates an $O(m(k))$ amount of ambiguity at step $k$.

Each vector system defines a positive semi-definite kernel

$$K = \sum_{k=0}^{\infty} \sum_{l=1}^{m(k)} a_{k,l} |v_{k,l}\rangle\langle v_{k,l}|$$

for any indexed set of non-negative scalars $\{v_{k,l} : k \in \mathbb{N}, l \in [1 : m(k)]\}$, provided that all $a_{k,l} \geq 0$, and $\sum_{k,l} a_{k,l} < \infty$.

To handle the multiplicity of $\mathrm{eigensystem}(K)$, we need to make four changes.

1. Generalize the definition of "fast-decaying coefficients" to fast-decaying *segments* of coefficients.

2. Prove that *segments* of adjacent eigenvalues are exponentially separated.

3. Prove the convergence of reducing subspaces (i.e. direct sums of eigenspaces), rather than eigenvectors.

4. Prove the convergence of whole segments of $\mathrm{eigensystem}(K)$, rather than individual entries like $(\lambda_n(K), |v_n(K)\rangle)$.

A fast-decaying sequence of coefficient segments with the specified multiplicity $m$ is obtained by slightly loosening a fast-decaying sequence of coefficients, so that instead of individual coefficients, it is segments of coefficients that are now decaying exponentially. This will force segments of the eigenvalues to be well-separated as well, and thus their corresponding reducing subspaces.

**Definition 10** (fast-decay with multiplicity). A coefficient system $c_{k,l}$ is fast-decaying iff there exists a sequence of numbers $\delta_{low,n} \in (0, 1]$, and a number $\epsilon \in [0, 1)$, and a sequence of numbers $\bar{c}_n$, such that

$$c_{n,l} \in [\bar{c}_n \delta_{low,n}, \bar{c}_n], \quad \bar{c}_{n+1} \leq \epsilon \bar{c}_n, \quad \forall n \in \mathbb{N}$$

We say that such a system is a fast-decaying coefficient system with parameters $(\epsilon, \delta_{low,n}, \bar{c}_n)$.

For example, a fast-decaying sequence is a fast-decaying system where the multiplicity counting function is $m(k) = 1$, and all $\delta_{low,n} = 1$. We will show that the coefficient system of a dot-product kernel $\sum_n \frac{c_n}{n!}(\boldsymbol{x}^\top \Gamma \boldsymbol{y})^n$ is fast-decaying in Appendix K.6.

Given a polynomially bounded multiplicity $m$, a vector system for $m$, and a fast-decaying coefficient system, define

$$K = \sum_k \sum_l a_{k,l} |v_{k,l}\rangle \langle v_{k,l}|$$

It is positive semidefinite and has finite trace. Therefore, its spectrum is discrete, and except for zero, each of its eigenvalue is positive and has only finite multiplicity. Therefore, its eigensystem is well-defined.

**Definition 11** (eigensegment). Given $K$, a positive semidefinite operator with finite trace, enumerate its eigenvalues as $\lambda_{0,1} \geq \cdots \geq \lambda_{0,m(0)} \geq \lambda_{1,1} \geq \cdots \geq 0$, counting eigenvalue multiplicity. Construct a corresponding orthonormal eigenbasis $|v_{k,l}\rangle$.

If all $\lambda_{n,1}, \ldots, \lambda_{n,m(n)}$ are distinct, then the $n$-th eigensegment of eigensystem$(K)$ is the set $\{(\lambda_{n,l}, \text{Span}(|v_{n,l}\rangle)) : l \in [1 : m(n)]\}$. Otherwise, if $\lambda_{n,1} = \lambda_{n,2}$, and all others are distinct, then the $n$-th eigensegment of eigensystem$(K)$ is the set

$$\{(\lambda_{n,1}, \text{Span}(|v_{n,1}\rangle, |v_{n,2}\rangle)), (\lambda_{n,2}, \text{Span}(|v_{n,1}\rangle, |v_{n,2}\rangle))\} \cup \{(\lambda_{n,l}, \text{Span}(|v_{n,l}\rangle)) : l \in [3 : m(n)]\}$$

In general, the eigensegment is obtained by merging degenerate eigenspaces.

Despite the valiant effort in removing ambiguity, the above definition still has some residual ambiguity: If unluckily, $\lambda_{k,m(k)} = \lambda_{k+1,1}$, then the reducing subspaces of the $k$-th and $(k+1)$-th eigensegments are not uniquely defined, since we can rotate the eigenvector pair $|v_{k+1,1}\rangle, |v_{k,m(k)}\rangle$ arbitrarily.

Fortunately, we will demonstrate that the residual ambiguity disappears in the $\epsilon \to 0$ limit for fast-decaying kernels.

Finally, we need to define what it means for two eigensegments to be close together:

**Definition 12** (eigensegment bulk closeness). Given two eigensegments of equal length, let $\lambda_{n,1}, \ldots, \lambda_{n,m(n)}$ be the eigenvalues from the first segment, and $\lambda'_{n,1}, \ldots, \lambda'_{n,m(n)}$ from the second. We say that they are $\epsilon$-close in bulk if

$$|\lambda_{n,l} - \lambda'_{n,l}| < \epsilon \min(\lambda_{n,l}, \lambda'_{n,l}) \quad \forall l \in [1 : m(n)]$$

In bulk? Indeed, due to the annoyances of degeneracy, we would first show that the eigenvalues converge as $O(\epsilon)$. After that is done, we can set $\epsilon$ small enough so that it will force the eigenspaces to match up as well, and thus does bulk closeness resolve into detailed closeness.

**Definition 13** (eigensegment detailed closeness). Given two eigensegments of equal length, we say that the two eigensegments are $\epsilon$-close in detail if they are $\epsilon$-close in bulk, and for each eigenspace $V'$ in the second eigensegment, there exists one or more eigenspaces $V_1, \ldots, V_s$ in the first, such that there exists a unitary angle operator $\Theta$, such that $\begin{bmatrix} \cos\Theta & \sin\Theta \\ -\sin\Theta & \cos\Theta \end{bmatrix}$ rotates $V_1 \oplus \cdots \oplus V_s$ to $V'$, and $\|\sin\Theta\|_{op} < \epsilon$.

See Appendix J.1 and (Davis & Kahan, 1970) for details on the meaning of the angle operator. Intuitively, the operator $\begin{bmatrix} \cos\Theta & \sin\Theta \\ -\sin\Theta & \cos\Theta \end{bmatrix}$ is the generalization of multidimensional rotation to a Hilbert space. It performs simultaneous rotation in many (potentially infinitely many) 2-dimensional planes. To say that $\|\sin\Theta\|_{op} < \epsilon$ means that in every single one of these planes, the angle of rotation is $< \arcsin(\epsilon)$.

The definition is by design *asymmetric*, because we will show that if one source eigensegment (think of the dot-product kernel's eigensystem) is always $O(\epsilon)$-close to a target eigensegment (think of the Hermite eigensystem) in the bulk sense, then as $\epsilon \to 0$, the source eigensegment will be forced to be $O(\epsilon)$-close to the target eigensegment in the detailed sense. We would rather not hit a moving target with a static gun, but hit a static target with a moving gun.

We point out that even "convergence in detail" does *not* imply convergence of every and each eigenvector, because a degeneracy in the target eigensegment may stubbornly remain broken in the

source segment. For example, it is possible that $\lambda'_{n,1} = \lambda'_{n,2}$ exactly in the target eigensegment, but $\lambda_{n,1} \neq \lambda_{n,2}$ in the source eigensegment. In this case, the eigenvectors corresponding to $\lambda_{n,1}, \lambda_{n,2}$ may be rotated by 45° compared to the chosen eigenvectors for $\lambda'_{n,1}, \lambda'_{n,2}$.

Concretely for our case of the dot-product kernels, this means that the kernel may have no degenerate eigenvectors, but the Hermite eigensystem *has* degenerate eigenvectors. In such a case, the best we can possibly do is proving that each eigenspace of the Hermite eigensystem corresponds to a *direct sum* of the kernel's eigensystem that is a small angle's rotation away.

Concretely, suppose that we have $\gamma_1 = \gamma_2$, then the Hermite eigensystem is degenerate, but we may find that the kernel's eigensystem stubbornly remains both nondegenerate *and* rotated by 45° askew of the Hermite basis. For example, it might stubbornly insist on containing two non-degenerate eigenvectors close to $\sqrt{\frac{1}{2}}(|h_{(0,1)}\rangle + |h_{(1,0)}\rangle)$ and $\sqrt{\frac{1}{2}}(|h_{(0,1)}\rangle - |h_{(1,0)}\rangle)$. This is fine, since their direct sum *does* converge to $\mathrm{Span}\left(|h_{(0,1)}\rangle, |h_{(1,0)}\rangle\right)$, and that is the best that can be proven. One cannot expect more to be proven given such degeneracy in the target eigensystem.

In the case of Theorem 2, before proving it, we saw why it should be true using a 2-dimensional picture with ellipses (Figure 22). Here, we can also see why it should be true using a 3-dimensional picture with ellipsoids.

Consider the case of three dimensions. We only have $|v_{0,1}\rangle, |v_{1,1}\rangle, |v_{1,2}\rangle$. In this case, the kernel $K = a_{0,1}|v_{0,1}\rangle\langle v_{0,1}| + a_{1,1}|v_{1,1}\rangle\langle v_{1,1}| + a_{1,2}|v_{1,2}\rangle\langle v_{1,2}|$. Diagonalizing the kernel is equivalent to finding the principal axes of the contour ellipsoid defined by $\{|x\rangle : \langle x|K|x\rangle = 1\}$. This ellipsoid is the unique ellipse tangent to the 6 planes defined by $a_{0,1}|v_{0,1}\rangle\langle v_{0,1}| = 1, a_{1,1}|v_{1,1}\rangle\langle v_{1,1}| = 1, a_{1,2}|v_{1,2}\rangle\langle v_{1,2}| = 1$.

Suppose we fix $a_{1,1}, a_{1,2}$, and let $a_{0,1} \to 0$. Then, the planes of $a_{1,1}|v_{1,1}\rangle\langle v_{1,1}| = 1, a_{1,2}|v_{1,2}\rangle\langle v_{1,2}| = 1$ remain constant, but the planes of $a_{0,1}|v_{0,1}\rangle\langle v_{0,1}| = 1$ diverge to infinity. The ellipsoid degenerates to a parallelogram prism defined by the 4 planes. Two of its principal axes rotate to become perpendicular to the 2 pairs of parallel planes, and essentially ignore $a_{0,1}$.

Suppose we fix $a_{1,1}, a_{1,2}$, and let $a_{0,1} \to \infty$. Then, the planes of $a_{1,1}|v_{1,1}\rangle\langle v_{1,1}| = 1, a_{1,2}|v_{1,2}\rangle\langle v_{1,2}| = 1$ remain constant, but the planes of $a_{0,1}|v_{0,1}\rangle\langle v_{0,1}| = 1$ converge to the origin. The ellipsoid degenerates to two flat parallelograms. Two of its principal axes rotate to fall within the parallelogram, perpendicular to its 2 edges.

Intuitively, we see that for a given $n$, the effect of all $a_{n+1,1}, \ldots, a_{n+1,m(n+1)}, a_{n+2,1}, \ldots$ terms in the kernel is a small perturbation on the $n$-th eigenspace, and negligible because the parallel planes diverge to infinity. The effect of all $a_{n-1,m(n-1)}, a_{n-1,m(n-1)-1}, \ldots, a_{0,1}$ is a large but *fixed* perturbation, forcing the $n$-th eigenspace to be perpendicular to all of $|v_{n-1}, m(n-1)\rangle, \ldots, |v_{0,0}\rangle$, but once that is done, their effects are also negligible because the parallel planes converge to the origin.

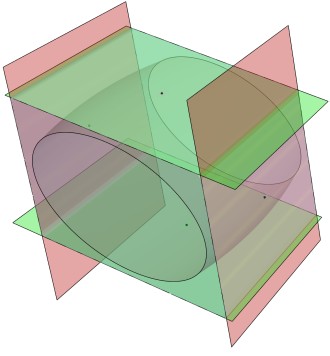
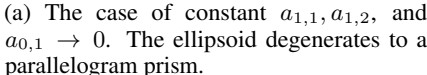
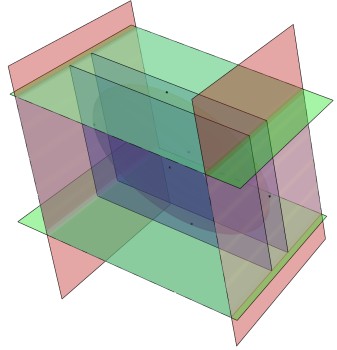

(a) The case of constant $a_{1,1}, a_{1,2}$, and $a_{0,1} \to 0$. The ellipsoid degenerates to a parallelogram prism.

(b) The case of constant $a_{1,1}, a_{1,2}$, and $a_{0,1} \to \infty$. The ellipsoid degenerates to two repeated parallelograms.

Figure 23: Diagonalizing the kernel in 3 dimensions at two limits.

Figure 24 shows the eigenstructure of $K$. On the coarse level, it is divided into exponentially separated segments. The $N$-th segment contains $m(N)$ eigenvalues clustered within an interval with order of magnitude $\Theta(\bar{a}_N)$. As $\epsilon \to 0$, the segments become ever cleanly separated, and also converging closer and closer to the $N$-th segment of a target eigenstructure. The target eigensegment may contain multiple eigenvalues with varying multiplicity. If a target eigenvalue $\lambda$ has multiplicity 3, then there will be precisely 3 eigenvalues (counting multiplicity) falling within $\lambda(1 \pm O(\epsilon))$.

Notably, these 3 eigenvalues may be degenerate, or not degenerate. In either case, the direct sum of their eigenspaces will have the same dimension as the eigenspace corresponding to $\lambda$, and it will make an angle of size $O(\epsilon)$.

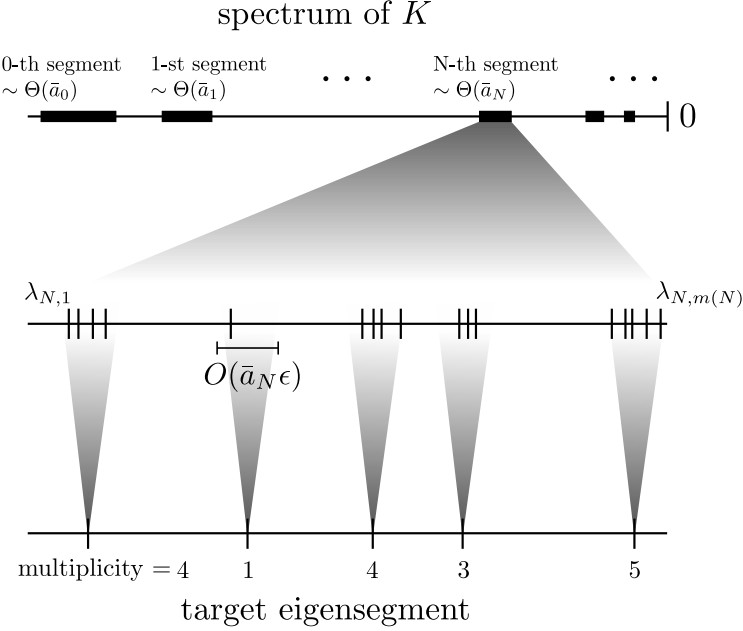

Figure 24: Structure of the kernel spectrum, zooming into one particular $N$-th segment. They converge to a segment of the target spectrum according to multiplicity of the target spectrum.

### K.3   STATEMENT OF THE THEOREM

The statement of the theorem is really unwieldy, so we write it out in a separate section.

A dot-product kernel

$$K(\boldsymbol{x}, \boldsymbol{y}) := \sum_{n=0}^{\infty} \frac{c_n}{n!} (\boldsymbol{x}^\top \boldsymbol{y})^n$$

is defined by its level-coefficients $c_0, c_1, \cdots \geq 0$. Given $\epsilon > 0$, it is $\epsilon$-fast-decaying if $c_{n+1} \leq \epsilon c_n$ for all $n \in \mathbb{N}$.

The multiplicity counting function for it is $m(n) = \binom{n+d-1}{d-1}$. It is polynomially bounded.

We enumerate the eigensystem of $K$ as

$$
\begin{aligned}
&(\lambda_{0,1}, V_{0,1}), \\
&(\lambda_{1,1}, V_{1,1}), \ldots, (\lambda_{1,d}, V_{1,d}), \\
&(\lambda_{2,1}, V_{2,1}), \ldots, (\lambda_{2,\binom{d+1}{2}}, V_{2,\binom{d+1}{2}}), \\
&\qquad\qquad \vdots \\
&(\lambda_{n,1}, V_{n,1}), \ldots, (\lambda_{n,\binom{n+d-1}{d-1}}, V_{n,\binom{n+d-1}{d-1}}), \\
&\qquad\qquad \vdots
\end{aligned}
$$

where the eigenvalues are arranged in decreasing order:

$$\lambda_{0,1} \geq \lambda_{1,1} \geq \cdots$$

In case of multiplicity, the same eigenvalue is written repeatedly. For example, if $\lambda$ has multiplicity 2, then it is written out twice.

Each $V_{k,l}$ is the eigenspace of $\lambda_{k,l}$. In case of eigenvalue multiplicity, repeat the eigenspace. For example, if $\lambda_{1,1} = \lambda_{1,2}$ is an eigenvalue of multiplicity 2, then it corresponds to a 2-dimensional eigenspace $V$. In this case, define $V_{1,1} = V_{1,2} = V$.

We also need to perform a *similar* "merging" on the Hermite eigensystem:

$$\left\{ \left( c_n \prod_{i=1}^{d} \gamma_i^{\alpha_i}, \operatorname{Span}\left( |h_{\boldsymbol{\alpha}}\rangle^{(\boldsymbol{\Sigma})} \right) \right) : n \in \mathbb{N}, |\boldsymbol{\alpha}| = n \right\}$$

This merging is, unfortunately, *not* exactly the same, because we can't just say that if $c_n \prod_{i=1}^{d} \gamma_i^{\alpha_i} = c_{n'} \prod_{i=1}^{d} \gamma_i^{\alpha_i'}$, then merge their eigenspaces, because we must *not* merge if $n \neq n'$, even if the eigenvalues happen to be the same in this case. This is because, as $\epsilon \to 0$, eventually $c_n$ and $c_{n'}$ will differ so greatly, that this *accidental* degeneracy is broken. We must only merge eigenspaces that are *non-accidentally* degenerate.

So why didn't we do this for $K$? Because $K$ is much harder to control for! We know everything there could be known about the Hermite eigenstructure, but $K$ is a big unknown that we must laboriously control. One thing that is a big unknown about $K$ is that we don't know which eigenvalues are accidentally the same, and which eigenvalues are non-accidentally the same. So we treat them without special considerations. We know which are accidental and which are not for the Hermite eigenstructure, so we treat them differently.

So, we perform a non-accidental merge of the Hermite eigensystem. For each $n \in \mathbb{N}$, we say that the $n$-th segment of the Hermite eigensystem is

$$\left\{ \left( c_n \prod_{i=1}^{d} \gamma_i^{\alpha_i}, \operatorname{Span}\left( |h_{\boldsymbol{\alpha}}\rangle^{(\boldsymbol{\Sigma})} \right) \right) : |\boldsymbol{\alpha}| = n \right\}$$

and *within* each segment, we merge the degenerate eigenspaces. For example, if it happens that $\prod_{i=1}^{d} \gamma_i^{\alpha_i} = \prod_{i=1}^{d} \gamma_i^{\alpha_i'}$ for $|\boldsymbol{\alpha}| = |\boldsymbol{\alpha}'| = n$, but no other $\boldsymbol{\alpha}''$ with $|\boldsymbol{\alpha}''| = n$, then replace both $\operatorname{Span}\left( |h_{\boldsymbol{\alpha}}^{(\boldsymbol{\Sigma})}\rangle \right)$ and $\operatorname{Span}\left( |h_{\boldsymbol{\alpha}'}^{(\boldsymbol{\Sigma})}\rangle \right)$ by their direct sum $\operatorname{Span}\left( |h_{\boldsymbol{\alpha}}^{(\boldsymbol{\Sigma})}\rangle, |h_{\boldsymbol{\alpha}'}^{(\boldsymbol{\Sigma})}\rangle \right)$.

With all this set up, we can now state Theorem 2, this time with full rigor.

---

**Theorem 7** (The HEA holds for a fast-decaying dot-product kernel on Gaussian measure). *Let*

1. $\Sigma$ *be a covariance matrix;*
2. $\Sigma = U\Gamma U^\top$ *be its SVD, with* $\Gamma = \text{diag}(\gamma_1, \ldots, \gamma_d)$;
3. $\mu = \mathcal{N}(0, \Sigma)$;

*For any* $N \in \mathbb{N}$, *there exists constants* $C_N, D_N, \epsilon_N$, *such that if* $\epsilon \in [0, \epsilon_N]$, *and* $K$ *is an* $\epsilon$-*fast-decaying dot-product kernel, the following happens.*

*For any element* $(\lambda, V)$ *within the merged* $N$-*th segment of the Hermite eigenstructure, there exists exactly* $\dim V$ *eigenvalues (counting multiplicity) of* $K$ *that are in the interval* $\lambda(1 \pm C_N\epsilon)$. *Let their corresponding eigenspaces be* $V_1, \ldots, V_{\dim V}$, *then there exists a unitary operator* $\begin{bmatrix} \cos\Theta & \sin\Theta \\ -\sin\Theta & \cos\Theta \end{bmatrix}$ *that rotates* $V_1 \oplus \cdots \oplus V_{\dim V}$ *to* $V$, *such that* $\| \sin\Theta \|_{op} < D_N\epsilon$.

---

## K.4 PART 1: EXPONENTIAL SEPARATION OF EIGENSYSTEM SEGMENTS

For each segment $N$, we use the min-max principle to construct upper and lower bounds on its eigenvalues.

---

**Lemma 1** (exponential separation of eigensystem segments). *Given*

1. *a polynomially bounded multiplicity counting function* $m$,
2. *a fast-decaying coefficient system* $c_{k,l}$ *with parameters* $(\epsilon, \delta_{low,n}, \bar{a}_n)$,
3. *a linearly independent vector system* $|v_{k,l}\rangle$,
4. *operator*

$$K := \sum_{k=0}^{\infty} \sum_{l=1}^{m(l)} a_{k,l} |v_{k,l}\rangle\langle v_{k,l}|$$

*then for any* $n$, *there exists constants* $\epsilon_n, C_n, D_n > 0$, *such that*

$$\lambda_{n,1}, \ldots, \lambda_{n,m(n)} \in [C_n\bar{a}_n, D_n\bar{a}_n]$$

*for all* $\epsilon \in [0, \epsilon_n]$.
*The constants* $\epsilon_n, C_n, D_n$ *depend only on* $\delta_{low,0}, \ldots, \delta_{low,n}$ *and the Gram matrix of the vectors* $|v_{0,1}\rangle, \ldots, |v_{n,m(n)}\rangle$.

---

We bound the entire eigensegment $\lambda_{N,1}(K) \geq \cdots \geq \lambda_{N,m(N)}(K)$ by the min-max principle. This is analogous to what we did in Appendix J.3.1, but simplified, because we do not need to produce sharp bounds. That comes later.

For the lower bound, we use $V = \text{Span}\left(|v_{0,1}\rangle, \ldots, |v_{N,m(N)}\rangle\right)$

$$\lambda_{N,m(N)}(K) \geq \min_{\substack{x \in \mathrm{Span}\big(|v_{0,1}\rangle,...,|v_{N,m(N)}\rangle\big) \\ \|x\|=1}} \langle x|K|x\rangle$$

$$= \min_{\substack{x \in \mathrm{Span}\big(|v_{0,1}\rangle,...,|v_{N,m(N)}\rangle\big) \\ \|x\|=1}} \sum_{k=0}^{\infty} \sum_{l=1}^{m(k)} a_{k,l}|\langle x|v_{k,l}\rangle|^2$$

$$\geq \min_{\substack{x \in \mathrm{Span}\big(|\hat{v}_{0,1}\rangle,...,|\hat{v}_{N,m(N)}\rangle\big) \\ \|x\|=1}} \sum_{k=0}^{N} \sum_{l=1}^{m(k)} a_{k,l}|\langle x|v_{k,l}\rangle|^2$$

$$\geq \frac{\bar{a}_N}{\delta_{min,N}} \min_{\substack{x \in \mathrm{Span}\big(|\hat{v}_{0,1}\rangle,...,|\hat{v}_{N,m(N)}\rangle\big) \\ \|x\|=1}} \sum_{k=0}^{N} \sum_{l=1}^{m(k)} |\langle x|v_{k,l}\rangle|^2$$

$$= \frac{\bar{a}_N}{\delta_{min,N}} \lambda_{\min}(G)$$

$$= \Omega(\bar{a}_N)$$

where $G$ is the Gram matrix of the vectors $|v_{0,1}\rangle, \ldots, |v_{N,m(N)}\rangle$. By assumption, these vectors are linearly independent, so the Gram matrix is positive definite.

For the upper bound, we use $V = \mathrm{Span}\big(|v_{0,1}\rangle, \ldots, |v_{N-1,m(N-1)}\rangle\big)$.

$$\lambda_{N,1}(K) \leq \sup_{\substack{x \in \mathrm{Span}\big(|v_{0,1}\rangle,...,|v_{N-1,m(N-1)}\rangle\big)^{\perp} \\ \|x\|=1}} \langle x|K|x\rangle$$

$$= \sup_{\substack{x \in \mathrm{Span}(|\hat{v}_{N,1}\rangle,...) \\ \|x\|=1}} \langle x|K|x\rangle$$

$$\leq \sup_{\substack{x \in \mathrm{Span}(|\hat{v}_{N,1}\rangle,...) \\ \|x\|=1}} \sum_{l=1}^{m(N)} a_{N,l}\langle x|v_{N,l}\rangle\langle v_{N,l}|x\rangle + \lambda_{\max}\left(\sum_{k=N+1}^{\infty}\sum_{l=1}^{m(k)} a_{k,l}|v_{k,l}\rangle\langle v_{k,l}|\right)$$

$$\leq \sup_{\substack{x \in \mathrm{Span}\big(|\hat{v}_{N,1}\rangle,...,|\hat{v}_{N,m(N)}\rangle\big) \\ \|x\|=1}} \sum_{l=1}^{m(N)} a_{N,l}\langle x|v_{N,l}\rangle\langle v_{N,l}|x\rangle + \mathrm{Tr}\left[\sum_{k=N+1}^{\infty}\sum_{l=1}^{m(k)} a_{k,l}|v_{k,l}\rangle\langle v_{k,l}|\right]$$

$$= \lambda_{\max}\left(\left[\sum_{l=1}^{m(N)} a_{N,l}\langle\hat{v}_{N,i}|v_{N,l}\rangle\langle v_{N,l}|\hat{v}_{N,j}\rangle\right]_{i,j=1}^{m(N)}\right) + \underbrace{\sum_{k=N+1}^{\infty}\sum_{l=1}^{m(k)} a_{k,l}}_{\text{use polynomial multiplicity}}$$

$$\leq \mathrm{Tr}\left[\left[\sum_{l=1}^{m(N)} a_{N,l}\langle\hat{v}_{N,i}|v_{N,l}\rangle\langle v_{N,l}|\hat{v}_{N,j}\rangle\right]_{i,j=1}^{m(N)}\right] + O(\bar{a}_N\epsilon)$$

$$= \sum_{l=1}^{m(N)}\sum_{i=1}^{m(N)} a_{N,l}\langle\hat{v}_{N,i}|v_{N,l}\rangle\langle v_{N,l}|\hat{v}_{N,i}\rangle + O(\bar{a}_N\epsilon)$$

$$\leq \bar{a}_N \sum_{l=1}^{m(N)}\sum_{i=1}^{m(N)} \langle\hat{v}_{N,i}|v_{N,l}\rangle\langle v_{N,l}|\hat{v}_{N,i}\rangle + O(\bar{a}_N\epsilon)$$

$$= O(\bar{a}_N)$$

## K.5 PART 2: CONVERGENCE IN BULK

Now that the spectrum is divided into exponentially separated segments, we can perform a surgical extraction of each $N$-th segment of the spectrum, to cut off both the "head" part $< N$, and the "tail" part $> N$.

---

**Lemma 2** (bulk insensitivity of eigensystem segments). *Under the same assumptions as Appendix K.4, for any $n$, there exists constants $\epsilon_n > 0$, such that the $N$-th eigensegment is $O(\epsilon)$-close in bulk to the spectrum of the matrix*

$$\left[ \sum_{l=1}^{m(N)} a_{N,l} \langle \hat{v}_{N,i} | v_{N,l} \rangle \langle v_{N,l} | \hat{v}_{N,j} \rangle \right]_{i,j=1}^{m(N)}$$

*for all $\epsilon \in [0, \epsilon_n]$.*
*The constant $\epsilon_n$ and the constant in $O(\epsilon)$ depend only on $\delta_{low,0}, \delta_{low,n}$ and the Gram matrix of the vectors $|v_{0,1}\rangle, \ldots, |v_{n,m(n)}\rangle$.*

---

Intuitively, the lemma states that the eigensystem segments separate very cleanly. First, relative to the $N$-th eigensegment, all the higher-order segments are $O(\epsilon)$-small, and thus ignorable. Second, relative to the $N$-th segment, the *only* effect of the lower-order segments is to force the $N$-th eigensegment into a safe subspace very close to the orthogonal subspace $\text{Span}\left( |\hat{v}_{N,1}\rangle, \ldots, |\hat{v}_{N,m(N)}\rangle \right)$, within which the lower-order terms $a_{0,1}|v_{0,1}\rangle\langle v_{0,1}|, \ldots, a_{N-1,m(N-1)}|v_{N-1,m(N-1)}\rangle\langle v_{N-1,m(N-1)}|$ all vanish.

Stated in another way, the lemma states that the $N$-th segment of the spectrum of $K$ is $O(\epsilon)$-close in bulk to $\tilde{K}$. To obtain $\tilde{K}$, we first remove its tail $\sum_{n=N+1}^{i} \sum_{l=1}^{m(n)} a_{n,l}|v_{n,l}\rangle\langle v_{n,l}|$, then project to the space orthogonal to $|v_{0,1}\rangle, \ldots, |v_{N-1,m(N-1)}\rangle$ to remove its head, to obtain

$$\tilde{K} = \sum_{i,j=1}^{m(N)} \sum_{l=1}^{m(N)} a_{N,l} |\hat{v}_{N,i}\rangle \langle \hat{v}_{N,i} | v_{N,l} \rangle \langle v_{N,l} | \hat{v}_{N,j} \rangle \langle \hat{v}_{N,j} |$$

The key of the proof is to ensure that each cut perturbs the eigenvalues by $O(\bar{a}_N \epsilon)$, so that we would extract something that is $O(\epsilon)$-close in bulk to the original eigensegment.

### K.5.1 CUTTING OFF THE TAIL
We need:

---

**Theorem 8** (Wielandt–Hoffman inequality (Kato, 1987)). *Let $A, B$ be self-adjoint operators, such that $C := B - A$ is a trace-class operator, then we can enumerate the eigenvalues of $A, B, C$ as $\alpha_i, \beta_i, \gamma_i$ (including eigenvalue multiplicity) such that*

$$\sum_i |\alpha_i - \beta_i| \leq \sum_i |\gamma_i| \tag{54}$$

---

The tail of the operator $K$ is the part that comes after the $a_{N,l}$ coefficients. It is bounded in trace norm:

$$\text{Tr}\left[ \sum_{k=N+1}^{\infty} \sum_{l=1}^{m(k)} a_{k,l} |v_{k,l}\rangle\langle v_{k,l}| \right] = \sum_{k=N+1}^{\infty} \sum_{l=1}^{m(k)} a_{k,l}$$

$$\leq \sum_{k=N+1}^{\infty} \bar{a}_N \epsilon^{k-N} m(k)$$

$$\leq \bar{a}_N \sum_{k=N+1}^{\infty} \epsilon^{k-N} A k^D$$

$$= O(\bar{a}_N \epsilon)$$

where we use the polynomial bound $m(k) \leq Ak^D$.

Thus, by Wielandt–Hoffman, removing the tail perturbs the spectrum by only $O(\bar{a}_N \epsilon)$. Note particularly:

1. all segments from the 0-th to the $N$-th remain exponentially separated;

2. the perturbed $N$-th segment is $O(\epsilon)$-close in bulk to the original $N$-th segment.

### K.5.2 CUTTING OFF THE HEAD

Cutting off the head is simply cutting off the inverted tail.

Having cut off the tail, we have a finite-rank operator

$$\tilde{K} := \sum_{k=0}^{N} \sum_{l=1}^{m(k)} a_{k,l} |v_{k,l}\rangle \langle v_{k,l}|$$

that splits into two reducing subspaces $V, V^{\perp}$, where $V = \mathrm{Span}\,(|0, 1\rangle, \ldots, |N, m(N)\rangle)$. It is zero on $V^{\perp}$ and positive-definite on $V$. Therefore, we drop down from the full Hilbert space to just $V$, where we can reason with matrices.

We express $\tilde{K}$ in matrix form in the orthonormal basis $|\hat{v}_{k,l}\rangle$, ordered so that $|\hat{v}_{N,1}\rangle, \ldots, |\hat{v}_{N,m(N)}\rangle$ come before $|\hat{v}_{0,1}\rangle, \ldots, |\hat{v}_{N-1,m(N-1)}\rangle$:

$$[\tilde{K}] = \begin{bmatrix} A & B \\ B^{\top} & C + D \end{bmatrix}$$

where the four matrices are:

$$A = \left[ \sum_{l=1}^{m(N)} a_{N,l} \langle \hat{v}_{N,i}|v_{N,l}\rangle \langle v_{N,l}|\hat{v}_{N,j}\rangle \right]_{i,j=1}^{m(N)},$$

$$B = \left[ \sum_{l=1}^{m(N)} a_{N,l} \langle \hat{v}_{N,i}|v_{N,l}\rangle \langle v_{N,l}|\hat{v}_{n,j}\rangle \right]_{i \in 1:m(N),\ (n,j) \in [0:N-1,1:m]}$$

$$C = \left[ \sum_{l=1}^{m(N)} a_{N,l} \langle \hat{v}_{n,j}|v_{N,l}\rangle \langle v_{N,l}|\hat{v}_{n',j'}\rangle \right]_{(n,j),(n',j') \in [0:N-1,1:m]}$$

$$D = \left[ \sum_{n=0}^{N-1} \sum_{l=1}^{m(N)} a_{n,l} \langle \hat{v}_{n,j}|v_{n,l}\rangle \langle v_{n,l}|\hat{v}_{n',j'}\rangle \right]_{(n,j),(n',j') \in [0:N-1,1:m]}$$

For a symmetric matrix in block form,

$$\begin{bmatrix} A & B \\ B^{\top} & C + D \end{bmatrix}^{-1} = \begin{bmatrix} A^{-1} & 0 \\ 0 & 0 \end{bmatrix} + \begin{bmatrix} A^{-1}BS^{-1}B^{\top}A^{-1} & -A^{-1}BS^{-1} \\ -S^{-1}B^{\top}A^{-1} & S^{-1} \end{bmatrix}$$

where $S = D + C - B^{\top}A^{-1}B$. We need several crude bounds on $A, B, C, D$, to prove that the first term really is the bulk term, and the second term really is an order $O(\epsilon)$ perturbation upon the bulk term, and thus we can safely cut it off according to the Wielandt–Hoffman inequality.

1. For each of $A, B, C$, their entries are all bounded in absolute values by $O(\bar{a}_N)$. Thus, their spectral radii are bounded by $O(\bar{a}_N)$.

2. By Cauchy interlacing law, $\lambda_{\min}(A) \geq \lambda_{\min}([\tilde{K}]) = \Omega(\bar{a}_N)$. Thus, the spectrum of $A$ is $\Theta(\bar{a}_N)$, and the spectrum of $A^{-1}$ is $\Theta(\bar{a}_N^{-1})$.

3. Notice that the matrix $D$ is constructed similarly to the matrix $[\tilde{K}]$, except with one more truncation. Therefore, by the same argument, its least eigenvalue is on the order of $\Omega(\bar{a}_{N-1}) = \Omega(\bar{a}_N/\epsilon)$.

4. Therefore, $B^\top A^{-1}, A^{-1}B$ have entries bounded in order $O(1)$, and $B^\top A^{-1}B$ has entries bounded in order $O(\bar{a}_N)$.

5. Therefore, $S$ has the same spectrum as $D$ with an order $O(\bar{a}_N)$ perturbation. Since $D$ has smallest eigenvalue $\Omega(\bar{a}_{N-1})$, so does $S$. Therefore, $S^{-1}$ has largest eigenvalue $O(1/\bar{a}_{N-1}) = O(\epsilon/\bar{a}_N)$.

Thus, the $N$-th segment of the eigenstructure of $\tilde{K}$, inverted, is $O(\epsilon)$-close in bulk to the eigenstructure of $A^{-1}$. Inverting again, we find that it is $O(\epsilon)$-close in bulk to the eigenstructure of $A$.

### K.6 Part 3: The special case of dot-product kernels

Recall, we need to show that, in the standard Gaussian space $L^2(\mathcal{N}(0, I_d))$, an operator $K$ defined by kernel $\sum_n \frac{c_n}{n!}(\boldsymbol{x}^\top \Gamma \boldsymbol{y})^n$ converges to the Hermite eigensystem. Here, $\Gamma = \mathrm{diag}(\gamma_1, \ldots, \gamma_d)$ with $\gamma_1, \ldots, \gamma_d > 0$.

Before applying the two generic lemmas. We need to first show that the kernel does have a fast-decaying coefficient system. This is because, even though $c_n$ is a fast-decaying coefficient sequence, it does not automatically imply that $K$ has a fast-decaying coefficient system if we express it in the monomial vector system.

Once this is done, we apply Appendix K.4 to conclude that the kernel's spectrum is divided into exponentially separated segments. Then, we "hopscotch" through several eigenstructures, until we show that the $N$-th eigensegment of $K$ is $O(\epsilon)$-close in bulk to the $N$-th eigensegment of a stretched dot-product kernel $ce^{\epsilon(\boldsymbol{x}^\top \Gamma \boldsymbol{y})}$. This argument is nearly the same as the proof of Appendix K.5, with a small extension to account for the special structure of dot-product kernels. Finally, we hopscotch again, applying Davis–Kahan at every step, to prove that the eigenspaces also converge as $O(\epsilon)$.

#### K.6.1 Combinatorics with the monomial basis

In analogy with the multidimensional Hermite vector system

$$|h_{\boldsymbol{\alpha}}\rangle = \bigotimes_{i=1}^{d} |h_{\alpha_i}\rangle = \prod_{i=1}^{d} h_{\alpha_i}(x_i)$$

we define the multidimensional normalized monomial vector system, by taking the tensor product of the single-dimensional normalized monomial vectors:

$$|v_{\boldsymbol{\alpha}}\rangle = \bigotimes_{i=1}^{d} |v_{\alpha_i}\rangle = \prod_{i=1}^{d} \frac{x_i^{\alpha_i}}{\sqrt{(2\alpha_i - 1)!!}}$$

Both of them are vector systems over the polynomially bounded multiplicity counting function

$$m(n) := \binom{n + d - 1}{d - 1}$$

The $n$-th segment of the monomial vector system consists of $\{|v_{\boldsymbol{\alpha}}\rangle : |\boldsymbol{\alpha}| = n\}$, and similarly, for Hermite, $\{|h_{\boldsymbol{\alpha}}\rangle : |\boldsymbol{\alpha}| = n\}$.

The Hermite vector system is obtained by the Gram–Schmidt process on the monomial vector system.

Now, consider the form of the dot-product kernel function:

$$K(x, y) := \sum_{n=0}^{\infty} \frac{c_n}{n!} \left( \sum_{i=1}^{d} \gamma_i x_i y_i \right)^n = \sum_{n=0}^{\infty} \frac{c_n}{n!} \sum_{|\boldsymbol{\alpha}|=n} \frac{n!}{\alpha_1! \cdots \alpha_d!} \left( \prod_{i=1}^{d} \gamma_i^{\alpha_i} x_i^{\alpha_i} y_i^{\alpha_i} \right)$$

In bra-ket notation, the operator is

$$K = \sum_{n=0}^{\infty} \sum_{\boldsymbol{\alpha}:|\boldsymbol{\alpha}|=n} c_n \prod_{i=1}^{d} \left( \frac{\gamma_i^{\alpha_i}(2\alpha_i - 1)!!}{\alpha_i!} \right) |v_{\boldsymbol{\alpha}}\rangle\langle v_{\boldsymbol{\alpha}}|$$

Since $c_n$ is a fast-decaying sequence, it remains to show that the quantity

$$\max_{\boldsymbol{\alpha}:|\boldsymbol{\alpha}|=n} \prod_{i=1}^{d} \left( \frac{\gamma_i^{\alpha_i}(2\alpha_i - 1)!!}{\alpha_i!} \right)$$

grows *at most* exponentially with $n$. Once we show that, we know that the coefficient system is also fast-decaying. Equivalently, we need to show that

$$\max_{\boldsymbol{\alpha}:|\boldsymbol{\alpha}|=n} \sum_{i=1}^{d} \left( \alpha_i \ln(\gamma_i/2) + \ln \binom{2\alpha_i}{\alpha_i} \right)$$

grows *at most* linearly with $n$. Note that, because such a bound on does not depend on the choice of the fast-decaying sequence $c_n$, it allows us to change $c_n$ to any other $c_n'$ with the same decay rate $\epsilon$, and still get a fast-decaying coefficient system.

First term is trivial:

$$\sum_{i=1}^{d} \alpha_i \ln(\gamma_i/2) \in \left[ n \min_{i\in[1:d]} \ln(\gamma_i/2), n \max_{i\in[1:d]} \ln(\gamma_i/2) \right] = \Theta(n)$$

To bound the second term, we do some combinatorics. Let $a_k := \binom{2k}{k}$, then we have $\frac{f(k+1)}{f(k)} = 4 - \frac{2}{k+1}$ strictly monotonically increasing. Therefore, $f$ is strictly log-convex. Therefore, $\boldsymbol{\alpha} \mapsto \sum_{i=1}^{d} \ln f(\alpha_k)$ is strictly Schur-convex. Thus,

$$\sum_{i=1}^{d} \ln f(\alpha_k)$$

achieves its upper bound at $(n, 0, \ldots, 0)$, and lower bound when all $\alpha_k$ are as close to equal as possible.

Upper bound:

$$\ln f(n) = \ln \binom{2n}{n} = 2n \ln 2 - \frac{1}{2} \ln(\pi n) + O(1/n) = \Theta(n)$$

Lower bound:

$$d \ln f(n/d) = d \ln \binom{2n/d}{n/d} = 2n \ln 2 - \frac{1}{2} \ln(\pi n/d) + O(1/n) = \Theta(n)$$

In fact, we have shown that the coefficient block is very tightly clustered:

$$\sum_{i=1}^{d} \ln f(\alpha_k) = 2n \ln 2 - \frac{1}{2} \ln(\pi n) + [-\ln \sqrt{d}, 0] + O(1/n)$$

for all $\boldsymbol{\alpha}$ satisfying $|\boldsymbol{\alpha}| = n$.

### K.6.2 Convergence in bulk

Like the general case, the proof has multiple parts. First, we apply Appendix K.4 to conclude that $K$ has exponentially separated eigensegments. Then, we hopscotch through multiple eigensystem segments to show eigensegment convergence in bulk, as in Appendix K.5.

$$N\text{-th segment of } K(\boldsymbol{x}, \boldsymbol{y}) = \sum_{n=0}^{\infty} \frac{c_n}{n!} (\gamma_1 x_1 y_1 + \cdots + \gamma_d x_d y_d)^n$$

$$\left\| O(\epsilon) \right.$$

$$N\text{-th segment of } \tilde{K}(\boldsymbol{x}, \boldsymbol{y}) = \sum_{n=0}^{N} \frac{c_n}{n!} (\gamma_1 x_1 y_1 + \cdots + \gamma_d x_d y_d)^n$$

$$\|$$

$$(0\text{-th segment of } [\tilde{K}]^{-1})^{-1}$$

$$\left\| O(\epsilon) \right.$$

$$(\text{eigensystem of just the } a_N \text{ part of } [\tilde{K}]^{-1})^{-1}$$

$$\left\| \text{ provided that } c_N' = c_N \right.$$

$$(\text{eigensystem of just the } a_N \text{ part of } [\tilde{K}']^{-1})^{-1}$$

$$\left\| O(\epsilon) \right.$$

$$(0\text{-th segment of } [\tilde{K}']^{-1})^{-1}$$

$$\|$$

$$N\text{-th segment of } \tilde{K}'(\boldsymbol{x}, \boldsymbol{y}) = \sum_{n=0}^{N} \frac{c_n'}{n!} (\gamma_1 x_1 y_1 + \cdots + \gamma_d x_d y_d)^n$$

$$\left\| O(\epsilon) \right.$$

$$N\text{-th segment of } K'(\boldsymbol{x}, \boldsymbol{y}) = \sum_{n=0}^{\infty} \frac{c_n'}{n!} (\gamma_1 x_1 y_1 + \cdots + \gamma_d x_d y_d)^n$$

$$\left\| \text{ set } K'(\boldsymbol{x}, \boldsymbol{y}) = c e^{\epsilon(\boldsymbol{x}^\top \Gamma \boldsymbol{y})}, \text{ where } c = c_N / \epsilon^N \right.$$

$$N\text{-th segment of } K'(\boldsymbol{x}, \boldsymbol{y}) = \frac{c_N}{\epsilon^N} e^{\epsilon(\boldsymbol{x}^\top \Gamma \boldsymbol{y})}$$

$$\left\| O(\epsilon) \right.$$

$$N\text{-th segment of } \left\{ \left( \frac{c_N}{\epsilon^N} \epsilon^{|\boldsymbol{\alpha}|} \prod_{i=1}^{d} \gamma_i^{\alpha_i}, |h_{\boldsymbol{\alpha}}\rangle \right) \text{ for all } \boldsymbol{\alpha} \in \mathbb{N}_0^d \right\}$$

$$\|$$

$$\left\{ \left( c_N \prod_{i=1}^{d} \gamma_i^{\alpha_i}, |h_{\boldsymbol{\alpha}}\rangle \right) \text{ for all } \boldsymbol{\alpha} \in \mathbb{N}_0^d, \ |\boldsymbol{\alpha}| = N \right\}$$

Figure 25: Hopscotching through eigensystems.

The new trick here is that we avoid solving for the $N$-th eigensegment of $\tilde{K}$, by going down then up again, arriving at a previously solved kernel as if taking a subway.

### K.6.3 CONVERGENCE IN DETAIL

We have successfully proven the source eigensegment converges in bulk to the target eigensegment. It remains to prove convergence in detail. This requires breaking degeneracies in the source eigensegment whenever the degeneracy is broken in the target eigensegment (but not vice versa), followed by applying Davis–Kahan once per target eigenspace.

Fix some $N$. The $N$-th target eigensegment is

$$\left\{ \left( c_N \prod_{i=1}^{d} \gamma_i^{\alpha_i}, |h_{\boldsymbol{\alpha}}\rangle \right) \text{ for all } \boldsymbol{\alpha} \in \mathbb{N}_0^d, \; |\boldsymbol{\alpha}| = N \right\}$$

Some of the eigenvalues in it may be equal, because $\{\ln \gamma_1, \ldots, \ln \gamma_d\}$ may be $\mathbb{N}$-linearly dependent.[17] Therefore, merge the eigenvectors with equal eigenvalue into eigenspaces. The eigenspaces have distinct eigenvalues. Note in particular that this constitutes a *static target*: The coefficients $c_0, c_1, \ldots$ may change, but if $c_N \prod_{i=1}^{d} \gamma_i^{\alpha_i} = c_N \prod_{i=1}^{d} \gamma_i^{\alpha'_i}$ for one choice of the coefficients, then it is so for all choices.

Now, fix one such eigenspace $V$ and its corresponding eigenvalue $c_N \zeta$. Let the $N$-th segment of $K$ be

$$\{(c_N \zeta_{N,1}(K), |v_{N,1}(K)\rangle), \ldots, (c_N \zeta_{N,m(N)}(K), |v_{N,m(N)}(K)\rangle)\}$$

where we do not yet demand that the eigenvalues are all distinct. If some of the eigenvalues are unluckily degenerate, we just tolerate an arbitrary choice of the eigenvectors for now.

As previously argued, it is $O(\epsilon)$-close to the target eigensegment in bulk. Therefore, for small enough $\epsilon$, the source eigenvalues $c_N \zeta_{N,1}(K), \ldots, c_N \zeta_{N,m(N)}(K)$ will be corralled around their corresponding target eigenvalues, like iron filings around magnets. Because there are only $m(N)$ source eigenvalues to go around, each target eigenvalue can only grab exactly as many source eigenvalues as its multiplicity. In particular, this means that even if these of eigenvalues are highly degenerate, they still congregate into "herds" that keep a good distance away from each other by $\bar{a}_N \Theta(1)$.

In particular, this means $c_N \zeta$ will be able to grab exactly $\dim V$ source eigenvalues, so that they are all stuck within an interval $c_N(\zeta \pm O(\epsilon))$. Enumerate these as $c_N \zeta_{N,i_1}(K), \ldots, c_N \zeta_{N,i_{\dim V}}(K)$. Let their eigenvectors be $|v_{N,i_1}(K)\rangle, \ldots, |v_{N,i_{\dim V}}(K)\rangle$, and let the vector space spanned by them be $V_K$. This is a reducing subspace of $K$, and one that we wish to show as $O(\epsilon)$-close to $V$.

Follow through the hopscotching diagram again. At each step in the hopscotching, either there is an exact equality, in which case the reducing subspace is unchanged, or there is a perturbation on the operator that is $O(\epsilon)$ relative to the operator, in which case, by Davis–Kahan, the reducing subspace is perturbed by an angle of only $O(\epsilon)$.

We spell this out explicitly for the top half of the diagram. The bottom half is the same.

At the first step, $K = \sum_{n=0}^{\infty} \frac{c_n}{n!}(\boldsymbol{x}^\top \Gamma \boldsymbol{y})^n$ is perturbed by truncating the tail $\sum_{n=N+1}^{\infty} \frac{c_n}{n!}(\boldsymbol{x}^\top \Gamma \boldsymbol{y})^n$. This has operator norm $O(c_N \epsilon)$. Since the gap between $c_N \zeta_{N,i_1}(K), \ldots, c_N \zeta_{N,i_{\dim V}}(K)$ and all other eigenvalues of $K$ is on the order of $\bar{a}_N \Theta(1)$. Thus, by Davis–Kahan, truncating the tail only perturbs the reducing subspace $V_K$ by an angle $O(\epsilon)$.

The second step is an exact identity. Under a matrix inverse, the eigenspaces are preserved, even though the eigenvalues are inverted.

In the third step, the inverted head is truncated. The inverted head has operator norm on the order of $O(\epsilon/\bar{a}_N)$, while the remaining operator has operator norm on the order of $O(1/\bar{a}_N)$. Now, since the gap between $(c_N \zeta_{N,i_1}(K))^{-1}, \ldots, (c_N \zeta_{N,i_{\dim V}}(K))^{-1}$ and all other inverted eigenvalues of $[\tilde{K}]^{-1}$ is on the order $\frac{1}{\bar{a}_N}\Theta(1)$, truncating the inverted head only perturbs the reducing subspace by another angle $O(\epsilon)$.

$\square$

---

[17] Even if they are $\mathbb{N}$-linearly independent, the gaps between successive eigenvalues will get smaller for larger $N$ if $d \geq 3$. By the standard counting argument in Diophantine approximation theory, the gap decays at least as fast as $N^{-(d-2)}$.

