# OpenReview forum: "Predicting Kernel Regression Learning Curves from Only Raw Data Statistics"
_ICLR.cc/2026/Conference — ICLR 2026 Poster_

### Official Review · Reviewer_3sV7 · 2025-10-30

**Soundness:** 3
**Presentation:** 3
**Contribution:** 3
**Rating:** 8
**Confidence:** 2

**Summary:**

This paper presents a theoretical framework for predicting the learning behavior of kernel ridge regression (KRR) on high-dimensional possible complex datasets using rotation-invariant kernels.
It proposes the Hermite Eigenstructure Ansatz (HEA): the kernel's eigensystem (for rotation-invariant kernels) closely matches a "Hermite eigensystem" derived from multivariate Hermite polynomials in the data space, where the latter depends only on low-order statistics of the data distribution and has an explicit analytical form.
For two data models, wide Gaussian kernel on a Gaussian measure and fast-decaying dot-product kernel on a Gaussian measure, the ansatz is proven to hold asymptotically.
Numerical simulations suggest that the ansatz also holds approximately for real datasets like MNIST and CIFAR-10.
Moreover, MLPs in the feature learning regime are shown to learn Hermite polynomials in the order predicted by HEA (lower-degree first), suggesting broader applicability.

**Strengths:**

* The paper is well-written and clearly structured, making it easy to follow the main ideas.
* It is novel to propose the Hermite Eigenstructure Ansatz (HEA) to predict the learning curves of KRR using only low-order statistics of the data distribution. It would help to better understand the generalization behavior of kernel methods and intrinsic structure of data.
* The theoretical analysis seem to be solid for the two specific data models.
* Extensive numerical experiments are provided to validate the proposed theory, including synthetic data and real datasets (MNIST and CIFAR-10). The approximation seems to hold well, justifying the practical relevance of the theory.

**Weaknesses:**

* Most of the theory in this paper is heuristic, relying on the proposed Hermite Eigenstructure Ansatz (HEA) without rigorous proof except for two specific data models.
In addition, it would be helpful to bound the difference between the prediction errors (the original model and the Hermite one) in terms of the approximation error of HEA, giving non-asymptotic guarantees.

* The theoretical results are limited to rotation-invariant kernels and Gaussian data distributions, which may not generalize to more complex real-world scenarios.

**Questions:**

1. What are the connections between the HEA and the Gaussian equivalence assumption (GEA) used in prior works on learning curves of kernel methods?
2. What are the datasets that HEA fails to approximate well? Can you provide some more insights into the limitations of HEA?
3. How will HEA be useful for practical applications, such as kernel selection or hyperparameter tuning?

---

> ### Author Response · Authors · 2025-11-26
> **Author Response**
>
> We thank the reviewer for the feedback on our paper. We are pleased that the reviewer found our paper approachable, as well as noting our use of data structure. To address the reviewer’s comments and questions:
>
> **Dependence on heuristics**: We acknowledge the strength of analytical error bounds, and believe that they are worth pursuing. While we hope follow-up work will create such bounds, this is not our focus.
>
> **Beyond rotation invariant kernels and Gaussian data**: Generalizing beyond rotation-invariant kernels would inevitably apply to more complex scenarios, true. We chose to study rotation invariant kernels in particular as the ReLU NTK is rotation invariant, giving us hope that the HEA applies to MLPs. We believe the connection to ReLU MLPs (further explored in Sec. 6) suggests that the HEA will be useful beyond just the two specific kernel types.
>
> **GEA connections**: The HEA *itself* does not rely on the Gaussian equivalence assumption; Gaussian universality is assumed on the HEA-recovered eigenfeatures such that past KRR eigenframeworks apply. To give further clarity, we have mentioned the GEA as well as other assumptions beyond the HEA in a new Appendix F.
>
> **Datasets where the HEA fails**: The datasets that the HEA fails to approximate well are surprisingly the ones that are the least complex: take the iris, mushroom, or any other categorical dataset (that happen to have many less features than the typical image dataset does). This holds even for image datasets, with MNIST being predicted by the HEA noticeably worse than ImageNet. To make specific notes, the categorical datasets ‘fail’ being approximated by the HEA because the PCA components are highly non-Gaussian: we require that each component be independent and have Gaussian marginals across the dataset, but due to binary or other discrete axes being included, these requirements are not held. We generally find the independence of PCA components to be more important than the marginals being Gaussians. For MNIST being worse than ImageNet, this is due to the lower effective dimensionality of the data, as the top few are vastly more important than later ones. These modes are shown in Figure 18, where the first 4 coordinates can be visually seen to be extremely similar to the numbers ‘0’, ‘9 or 2’, ‘3’, and ‘5 or 6’. Our last condition for the HEA to work well (narrow kernel width) can be avoided for all datasets.
>
> **HEA practicality**: We speculate the HEA could be useful in performing kernel regression faster than the native algorithm. In particular, we believe hyperparameter sweeps to be vastly easier when using the HEA, as the eigenfunction and target coefficient via Gram-Schmidt steps are kernel-independent. Once those are found, estimating the test error for a particular kernel will be much faster than performing KRR.
>
> Thank you for your time.

---

### Official Review · Reviewer_Cuso · 2025-10-31

**Soundness:** 4
**Presentation:** 4
**Contribution:** 1
**Rating:** 0
**Confidence:** 4

**Summary:**

This paper introduces the "hermite eigensystem" as an approximate formula for the eigenvalues and eigenfunctions of dot-product kernels, expressed in terms of the data covariance matrix. The authors present two limiting cases where their approximation holds exactly, then show that the learning curves obtained by applying the standard theory of kernel ridge regression under their approximation captureswell the empirical learning curves of kernel ridge regression on real data.

**Strengths:**

- The paper is well written, and the results are clearly and logically organised.
- Developing an end-to-end theory of learning based directly on data statistics is a highly valuable goal.
- The experimental validation appears convincing and supports the proposed theoretical framework.

**Weaknesses:**

1. Theoretical learning curves for kernel regression have been established in prior work (e.g., *Optimal Rates for the Regularized Least-Squares Algorithm* by Caponnetto and De Vito). It is unclear what new insights an approximate, simplified formula based on Gaussian assumptions contributes to this existing body of knowledge.
2. Gaussian Equivalence Principles (e.g., *The Gaussian Equivalence of Generative Models for Learning with Shallow Neural Networks* by Goldt et al.) already characterise when Gaussian approximations like the one made by the authors are valid or not. The paper does not sufficiently discuss its relation to these frameworks.
3. The proposed approximation appears *uncontrolled*, as the limits of its validity are neither explored nor commented upon, leaving uncertainty about its general applicability.

**Questions:**

Presumably, if the data are actually assumed to come from the Gaussian distribution with covariance matrix $\Sigma$, then the eigenfunctions should be multivariate Hermite polynomials. Have the authors tried to relate the kernel eigenvalues to the Taylor coefficients of the Kernel expansion into powers of $\mathbf{x}\cdot\mathbf{y}$ exactly? Is capturing the right eigenvalues even important for the prediction of learning curves, or is it really the order of the eigenfunctions that matters?

---

> ### Author Response · Authors · 2025-11-26
> **Author Response**
>
> Hello, and thank you for the review.
>
> We appreciate your rating our paper as very sound in its results and very well-presented. It does make a significant contribution, as correctly identified by the other four reviewers. In response to your three comments:
>
> 1. Yes, we cite and use the existing theory of KRR/LRR generalization. The problem we solve is that one needs the kernel eigenfunctions to use this theory. Before this paper, a user would have to solve for these numerically, which makes theoretical insight into what is learned difficult or impossible. After this paper, users can just use our HEA formulae. This enables downstream theory.
>
> 2. Yes, we cite this literature. Goldt’s work in part inspired this paper. Our work is more quantitative: we not only claim “the data may be approximated as Gaussian,” but *provide a detailed and powerful downstream theory based on that approximation.*
>
> 3. We give theory that works on *real datasets.* How would you control an approximation you wish to apply to e.g. CIFAR-10? It’s a very interesting and difficult question. We take two approaches:
> * We empirically explore *and* comment on the limits of the validity of the HEA in Section 4.2 in terms of both the choice of kernel and the choice of dataset.
> * We give convergence rates for Theorem 2 (and in the proof of Theorem 1) in $\epsilon$ and $\sigma$, respectively.
>
> To your question: no, even with exactly Gaussian data, the eigenfunctions are not exactly Hermite polynomials. This is why limits are required in Theorems 1 and 2, and this is why our theoretical results are interesting. Our HEA does exactly what the reviewer suggests, relating kernel eigenvalues to the Taylor coefficients of a dot-product kernel. To the question of whether capturing the right eigenvalues is important: getting the right eigenvalues is important for good learning curve agreement, but yes, the eigenfunctions are more important.
>
> You yourself say that “developing an end-to-end theory of learning based directly on data statistics is a highly valuable goal.” We do this. It is difficult to see how you then justify a score of zero. Thank you for your time.

---

> ### Comment · Reviewer_Cuso · 2025-11-27
>
> Dear authors,
>
> Please do not misinterpret my words: the clarity of the exposition and soundness of the results do not imply significance of the contribution---that depends purely on the scientific content. The theory of generalisation/learning curves for kernel ridge regression is well established, including contributions from statistics, theoretical computer science and physics (e.g., Optimal Rates for the Regularized Least-Squares Algorithm by Caponnetto and De Vito, which you did not comment on, Asymptotic learning curves of kernel methods: empirical data versus teacher–student paradigm by Spigler et al., Spectrum dependent learning curves in kernel regression and wide neural networks by Bordelon et al. and many others ). You can look at this recent preprint for a modern overview: https://arxiv.org/pdf/2510.04780, by Wortsman and Loureiro. Concerning my initial comments:
>
> **1.** You are not solving a problem, but rather proposing an uncontrolled, approximate solution to a problem that can already be solved exactly. I don't understand the addition that this perspective provides to the existing literature. Previous results already enable downstream theories, either in solvable models where the eigenfunctions can be determined, or in cases where Gaussian universality applies (see next comment).
>
> **2.** Previous works on Gaussian universality, in particular Loureiro et al. "Learning curves of generic feature maps for
> realistic datasets with a teacher-student model" and follow-up works (e.g. Cui et al. "Generalisation Error Rates in Kernel Regression: The Crossover from the Noiseless to Noisy Regime"), have already explored the theoretical implications for learning curves and generalisation of Gaussian universality (i.e. the possibility to replace real data with Gaussian approximations having the same mean and variance). Moreover, the works I refer to characterise the boundaries of validity of Gaussian universality in a mathematically precise way, as opposed to your approximation, which is entirely uncontrolled. In this respect, your work can be seen as a vague restatement of previous results.
>
> **3.** The theorems you provide reduce the problem to such simplicity that it's hardly of any use. In practice, in both limits you consider, the kernel converges to a constant $c$. The constant $c$ is exactly diagonalised with one eigenfunction (the number $1$) and one eigenvalue (the constant $c$). No need to define "the hermite eigenstructure ansatz" for this, and I don't see how these two theorems make the theory interesting. The fact that your "theory" approximates well the empirical learning curves on large datasets might be due to many contributing factors, e.g. only the order of the eigenvalues matters and not their absolute value, you are only exploring the small-data regime (#data<=10^4).
>
> To my questions:
>
> - There is at least a case where I know this is true, and that's the high-dimensional limit with identity covariance. The Gaussian distribution converges to the uniform distribution on the sphere, and the kernel is always diagonalised by spherical harmonics.
>
> - What I meant by `exactly' is to plug the Taylor expansion into the eigenvalue problem, solve term by term, and resum.
>
> - On the question of capturing the right eigenvalues: why? The analyses of "Asymptotic learning curves of kernel methods: empirical data versus teacher–student paradigm" by Spigler et al., and "Spectrum dependent learning curves in kernel regression and wide neural networks" by Bordelon et al. imply that only the order matters for scaling.
>
> Anyway, there are at least two cases that you present in your experiments where your approximation for the eigenvalues should be completely wrong: the Laplace kernel and the ReLU NTK. These kernels are singular: their derivatives diverge when $x\cdot x'\to 1$. In cases like this, using the Taylor expansion of the kernel in $x\cdot x'$ (expanded around $x\cdot x'=0$) is tantamount to exchanging sums and integrals when the sum does not converge, which is fundamentally wrong.
>
> I do believe that developing an end-to-end theory of learning based directly on data statistics is a highly valuable goal. I don't believe that this work brings us any closer to the goal, hence my rating remains.

---

> > ### Author Response · Authors · 2025-12-02
> > **final response to reviewer Cuso**
> >
> > Thanks for the detailed reply! We weren't misinterpreting you, just politely disagreeing. If our paper, our rebuttal, and the four other positive reviews haven't convinced you our ideas here are useful, then we probably won't be able to. That's okay :) It's difficult to get to the bottom of such discussions over OpenReview, and we appreciate your trying. We'll still go through and respond to some of your key points.
> >
> > Re: Caponetto + De Vito (2007) and friends, we've got a whole Appendix C that discusses this body of prior work on the generalization of LRR/KRR. We didn't cite that particular paper, but we've now added it for completeness.
> >
> > Re: your point (2) on Gaussian universality, we think you may be conflating Gaussian universality in the kernel feature space with GU in the data space. In the papers you cite (and lots of others!), the Gaussian replacement takes place in the kernel feature space. (There's potential for confusion here, since when studying LRR, these spaces are of course the same. We hoped to clarify this with our depiction of the different spaces in Figure 1.) We believe the Loureiro paper you point to (which we also cite) applies GU in the feature space when studying KRR, not the data space. That paper does not discuss Hermite polynomials at all, so it's difficult for us to see how our HEA idea could be contained there.
> >
> > Re: point (3), this is a clever objection! Yeah, we should probably discuss this one explicitly. It's similar to the study of NN training from small init: one could object that in that limit, training takes infinite time, so the dynamics are trivial --- but of course, when you rescale time appropriately, you recover interesting (and newly tractable) dynamics. In our limit, sure, the constant eigenmode dwarfs the others, but when using an inverse method like KRR, exactly how small the small eigenmodes are (and what the eigenfunctions are) makes a big difference in the learning behavior once you have $n > 1$ samples. Remember also that the limits here are really just for theory --- our Figure 2 shows that the HEA works really well even for e.g. a Gaussian kernel of width one!
> >
> > You say that our theory could be correctly predicting learning curves for trivial reasons. We don't really know how to respond to this other than to say that we did a lot of experimentation, and it's not --- if you do something wrong in the theory, you predict the wrong learning curves. If you modify the HEA in any of numerous ways, the predictions become wrong. We're not sure what to say here besides challenging you to predict learning curves from only the statistics we use without using something like our HEA!
> >
> > Re: your questions:
> > * Yup, that high-dimensional isotropic Gaussian setting was a useful solved case for pointing us in the right direction here. The high-dim limit is doing a lot of work there! It's not clear how to apply it to real data, which is always anisotropic and often has a modest effective dimension. As far as we know, you can't exactly diagonalize the general Taylor-series dot-product-kernel-on-low-dim-Gaussian-data case in closed form. We tried pretty hard using precisely this Taylor series idea! The underlying commonality we came up with was the HEA.
> > * Re: why eigenvalues matter, you might look at e.g. [Mallinar et al (2022)](https://arxiv.org/abs/2207.06569): their trichotomy theorem is an example of how the eigenspectrum, independent of the eigenfns, can affect e.g. the strength of overfitting. Aside from this, we're not sure what to say besides (a) the KRR eigenframework does depend on the eigenvalues, so of course they affect the generalization error at least a little, and (b) empirically we find you usually need to get them right in order to get good learning curves.
> >
> > Re: singular kernels, yes, we agree it's surprising that the HEA works in cases with singular kernels! (To get some intuition for why it does, imagine that the data all lie on a sphere, which is sometimes a reasonable approximation in high dim. Then the singularity at zero doesn't matter.) The fact remains that the HEA predicts very well for these kernels nonetheless. To us, this is a sign that there's something fundamental about this Hermite eigenstructure idea. Hopefully we don't overstep in suggesting that the fact that our theory works even when you think it shouldn't should *raise* your evaluation, not lower it!
> >
> > Thanks again for the back-and-forth. A shame the OpenReview fiasco's prevented further exchange. Take care!

---

### Official Review · Reviewer_16Lq · 2025-11-01

**Soundness:** 4
**Presentation:** 4
**Contribution:** 3
**Rating:** 8
**Confidence:** 3

**Summary:**

Authors study kernel ridge regression on anisotropic data for rotation-invariant kernels.
They introduce a Hermite eigenstructure ansatz (HEA), that allows to approximately compute the eigendecomposition of the kernel function.
The main finding is that the covariance of the data and the coefficients of the target function in the basis of this eigendecomposition are enough to predict the test error of the kernel ridge regression.
The result holds when the data is 'Gaussian enough', which is empirically verified for certain real datasets.
In particular, the width of the kernel needs to be sufficiently large, as well as the effective dimension of the data.
The cases when the predicted test errors are further from the real values, are thoroughly studied.
Theoretical results prove that the predictions hold when data is Gaussian under two limiting regimes.

One of the technical issue that the authors had to deal with is how to obtain the coefficients of the expansion of the target function in the HEA of the kernel. They suggest to iteratively project the target function on the top eigenfunctions and use Gram-Schmidt orthogonalization to remove correlations between the Hermite polynomials that empirically works well even when the data is non-Gaussian, but the target is sufficiently smooth.

**Strengths:**

The paper is very well-written and constitutes a timely contribution as the role of the data in learning is extremely important.

Hermite eigenstructure ansatz is a clean and theoretically motivated tool to understand KRR.
The authors present a thorough analysis of when the proposed HEA method holds, both on synthetic and real datasets.
They empirically study the effects of data dimension, target function, and various kernels.
Interestingly, the authors also study the regime 'beyond kernel ridge regression' and find that their predictions hold in that scenario too, to some extent.

For the case of Gaussian data, two theorems regarding different scaling regimes (very wide kernel and very fast coefficient decay) are present.
This work is likely to motivate further research to understand rigorously where the limits of the use of the Gaussianity assumption are, possibly beyond kernel regimes.

**Weaknesses:**

I don't find serious weaknesses in the methodology, experiments or theoretical results of the paper, although I did not check the proofs in detail.

Extended related work discussion would help the reader to position current work:
1. In (Refinetti et al., 2023), the authors claim that the neural networks, trained on 'Gaussian versions' of CIFAR-10 (with the same mean and covariance), perform worse than on the real dataset. This can be interpreted that CIFAR-10 is not 'Gaussian enough'. Perhaps whether the data is 'Gaussian enough' depends on the class of learning algorithms, with KRR not being able 'to see further' than the covariance of the data. Is such interpretation correct?
2. There is a line of work that studies 'staircase property' of learning (e.g., Abbe et al., 2021). Is there a relation between this property and the results in Figure 4? Why both Figure 3 and Figure 4 only use monomials as a target and what would happen when polynomials are used instead?

Also, the notation of $h_i$ (eg in line 1135) can be confused with $h_{\mathbf{\alpha}}$.

Refinetti, Maria, Alessandro Ingrosso, and Sebastian Goldt. "Neural networks trained with sgd learn distributions of increasing complexity." International Conference on Machine Learning. PMLR, 2023.

Abbe, Emmanuel, et al. "The staircase property: How hierarchical structure can guide deep learning." Advances in Neural Information Processing Systems 34 (2021): 26989-27002.

**Questions:**

See the section above. Furthermore:
1. Throughout the text, both 'population data covariance, $\Sigma$' (eg line 47, Figure 1) and 'empirical data covariance, $\hat \Sigma$' (eg line 13, Figure 3) are used. From the description in Section 4, it seems that $\Sigma$ is the main quantity of study. Could the authors clarify to what extent they expect these results to hold when having access only to $\hat \Sigma$?
2. Line 1643 and Figure 18: what is $\gamma$? Should it be $\zeta$?
3. Figure 5: in the legend, shouldn't the labels for predicted and empirical be swapped?

---

> ### Author Response · Authors · 2025-11-26
> **Author Response**
>
> We thank the reviewer for the feedback on our paper. We are pleased that the reviewer sees the motivation for our framework potentially beyond the kernel regime, as well as noting our use of data within the theory. Here, we address the reviewer’s comments and questions:
>
> **Gaussianization of CIFAR-10/Image datasets**: Broadly yes, your interpretation is correct. It would appear that “Gaussian enough” as a condition depends on the class of learning algorithms, and that the interpretation of KRR not being able to ‘see further’ than the covariance is probably right. As we’re now able to show, KRR (at least within our rotation-invariant kernel setup) is only sensitive to the data’s covariance, whereas this is clearly not true for other networks (take ResNets per the Refinetti et al. paper). *For KRR specifically*, CIFAR-10 appears “Gaussian enough” as (1) PCA components are roughly independent and (2) they have approximately Gaussian marginals.
>
> **Polynomial targets**: The rationale behind Figure 4 (the MLP experiments) only using monomials as opposed to polynomials is precisely because of this: the HEA eigenvalues would no longer be representative of the effective optimization time if there are any staircasing components of a polynomial. It’s currently unclear how the staircase experiment can be interpreted in the HEA context. We are investigating that now, and we praise the reviewer’s intuition for what the most promising next step is! Polynomials weren’t used in Figure 3 (the kernel experiments) due to the target functions we were training on being pure Hermite terms—both linearity of the target function and properties of eigenstructures  would cause each monomial in the polynomial to be learned as if there were separate kernels learning each monomial term.
>
> **Hermite function notation**: We apologize about any confusion. We used different notation to denote the Hermites recovered numerically $h_i$ and those computed theoretically $h_{\alpha_i}$.
>
> **Population vs empirical data covariance**: Much of the main text of the paper focuses on the theoretical backing of the HEA, where the true population covariance $\Sigma$ is known. Within our empirics and figures, we largely assume that we have enough samples $n$ such that a PCA will reliably recover a sample covariance $\hat{\Sigma}$ that is close to the true underlying covariance $\Sigma$. Fortunately, real vision datasets commonly have sample size much larger than their dimensionality, which means that the empirical covariance matrix will indeed be a good estimate of the population covariance.
>
> **Gamma in line 1643**: Yes, good catch. This was a typo and it should be a $\zeta$.
>
> **Legend label mismatch**: The legend is correct, but we admit it is confusing due to the empirics being the line with the theory being points instead of vice-versa. In our updated version, we now explicitly note the change.
>
> Thank you for the review!

---

### Official Review · Reviewer_msN8 · 2025-11-01

**Soundness:** 3
**Presentation:** 3
**Contribution:** 4
**Rating:** 8
**Confidence:** 5

**Summary:**

The paper introduces the hermite eigenstructure ansatz (HEA), which gives a closed form expression for the eigensystem of rotation invariant kernels. Using the eigenframework, this gives the learning curvevs for KRR on real image datasets, which are empirically shown to approximate the learning curves well using only the covariance matrix and the hermite decomposition of the target function. They also show that HEA holds for gaussian data and two limiting cases of the kernel function. Finally, the papers empirically shows that MLPs learn hermite polynomials on real datasets in the order predicted by HEA.

**Strengths:**

The hermite eigenstructure ansatz solves one of the main difficulties of using the eigenframework predictions in predicting the learning curves of KRR on real datasets: estimating the eigenvalues and eigenfunctions of the kernel under the given data distribution. This a very important contribution and expands the applicability of this line of work.


Further, the paper provides extensive empirical evidence that the approximate closed form expression for eigenvalues and eigenfunctions can be used to very accurately predict the learning curves of KRR on a number of real image dataset and various synthetic setups, which is needed given the harndess of verfying the for real datasets HEA.


Finally, the paper clearly demonstrates the usefulness of insights from HEA, which they use to predict how MLPs learn functions.


Overall, I think that this is a very strong contribution that is presented clearly and concisely and should get accepted.

**Weaknesses:**

1. The paper is unnecessarily and sometimes confusingly written in an underdefined and informal way. For example, the use of undefined approximate sign $\approx$, which is crucial to the Hermite eigenstructure ansatz lines 251-260, is at time confusing. Since we are interested in predicting the learning curves, it is quite easy to define this approximate equivalence between two eigenstructures as having learning curves differing by at most something small. It is also unclear whatt does the final prediction mismatch depend on after using a number of these approximations. I think the whole framework would greatly benefit if some care was taken to quantify or at least characterize the dependence of the prediction error of using the hermite eigenstructure ansatz.
2. As explained in lines 968-969, and if I understand correctly, the hermite ansatz framework depends on eigenlearning framework to map predictions of eigenstructure to predictions of learning curves. Since the hermite ansatz is not formal (and not quantitative, see #1), the applicability of hermite eigenstructure ansatz also depends on the applicability of the eigenframework to the case considered (so it also depends on Gaussian Universality Ansatz). This way, the error of the final prediction aggregates the error of the hermite eigenstructure prediction and the eigenframework prediction. So it’s a bit unclear whether the conditions for success (Section 4.2) are sufficient for both of these errors to be small. I feel like this “hidden” dependence on the eigenframework should be more transparently discussed in the main body. This is especially given some recent work questioning whether the eigenframework actually applies in the case of NTK.
3. The paper only shows that HEA holds for the Gaussian data with a Gaussian kernel or other fast-decaying dot product kernels in a certain limit of the parameters of those kernels. It’s unclear what is the interpretation of this limit (e.g. taking the width of the Gaussian kernel to infinity). This section would also be much more clear if the limit of is more quantitatively defined.

**Questions:**

1. Can the approximate relationship in the definition of HEA and elsewhere be formalized in terms of the error of the prediction of the learning curves?
2. Is the HEA expected to hold whenever the Gaussian Universality Ansatz holds? How the two interact?
3. What is the interpretation of the infinite width limit of Gaussian kernel?

---

> ### Author Response · Authors · 2025-11-26
> **Author Response**
>
> Thank you for the strong praise of our paper. You made our day. We broadly agree with your assessment of its strengths and weaknesses.
>
> In response to weakness 1, about approximation: yes, this is a fair point. The suggestion to define “$\approx$” in terms of closeness of learning curves is a good one! The problem is that we really do want to talk about the similarity of the eigenvalues and eigenfunctions directly, not about a downstream consequence – our Theorems 1 + 2 are claims about the raw eigenstructure, not about downstream learning curves, and it’d add layers of complication to make it about that. (The claims are probably still true, but much more difficult to prove. See [this concurrent paper](https://arxiv.org/abs/2510.04780) for a positive example of a work which *does* go all the way to learning curves in its results. Their claims are rather narrower in scope, having focused their effort on making the full stack precise + rigorous. We think it’s worthwhile to push our Theorems 1 + 2 to claims about generalization error, but it is difficult.)
>
> What we really mean by “$\approx$” in the HEA is roughly: the $i$-th eigenvalue matches within a small multiplicative factor, and the $i$-th Hermite eigenfunction is almost entirely captured in the empirical kernel subspace associated with eigenvalues within a small multiplicative factor of $\lambda_i$. This is still not quite precise, but we will make a note of it after we introduce the HEA. While not mathematically precise, this is hopefully clear enough to serve as a guide for readers and future researchers who wish to formalize what we’ve found here.
>
> In response to weakness 2, about chained sources of error: this is also a subtle and interesting point. We use a sequence of many approximations and assumptions required to get learning curve predictions. Each such step is a place where error gets introduced. In our opinion, the fact that it all works so well at the end – that the errors do not accumulate unduly – is rather surprising, and much of the scientific work here was identifying the right settings and controls for this to be the case. While we cannot analytically control for every source of error, what we *can* and should do here is list them. They are as follows:
> 1. We model the data as Gaussian.
> 2. We assume the data has high dimension.
> 3. Even so, the HEA is an approximation at finite kernel width, and we pretend it is exact.
> 4. We assume our naive Gram-Schmidt procedure finds the correct Hermite coefficients.
> 5. Assuming the HEA gives the correct eigenfeatures, we assume Gaussian universality on those eigenfeatures so that the KRR eigenframework applies.
> 6. Even with Gaussian universality, the KRR eigenframework has finite error at finite sample size $n$. We assume that $n$ is large enough that this error is small.
>
> These are roughly our sources of error. Fortunately, the errors introduced in steps 5 and 6 have been pretty robustly shown to be small in prior work, so we aren’t too worried about them. Most of our debugging effort in this paper was in identifying + controlling sources 1-4. Note that the Gaussian universality assumption on kernel features (step 5) is independent of the HEA (steps 1-3), which is about the distribution in the data space, not feature space. We have added mention of this chain in the main text and a new Appendix F that gives this list and discusses each point.
>
> Re: wide kernels, what we’re really saying here is that the width of the kernel is much wider than the largest data eigenvalue $\gamma_1$. The “wide kernel” limit is the same as the “narrow data” limit. When the data has high effective dimension, one usually has that $\gamma_1 \ll \sum_i \gamma_i = \Theta(1)$, which explains why this works in practice. We’ll add mention of this in Section 4.
>
> Thanks again for the constructive review.

---

> > ### Comment · Reviewer_msN8 · 2025-11-27
> >
> > Thanks for the clear and concise answer.
> >
> > I agree with the authors that it's hard to make "$\approx$" mathematically precise and that there are many ways to do so. I still think being explicit and transparent about the sources of error (e.g., by discussing 1-6 in the paper) would help the reader understand the HEA framework better. Also, it would make the dependence on eigenframework and related results more explicit, which in turn would make the contribution more visible. Further, I still think that providing an outline of one particular way of tracking the final error would provide further justification for the framework and make the result more easily digestible to a wide audience of readers.
> >
> > Overall, I like the paper, and I think the contribution is strong.

---

> > > ### Author Response · Authors · 2025-12-02
> > >
> > > Yes, we agree. We've added some discussion of the meaning of "$\approx$" and steps 1-6 to the paper. (Doesn't seem we can upload a new version, otherwise we'd show you.)
> > >
> > > Re: tracking the final error, we think this is interesting and worthwhile, but the techniques required are outside our area of expertise, and at present we don't have a clear vision of how you'd do this. We do agree it would widen the audience and will take this suggestion into consideration in our future work.

---

### Official Review · Reviewer_MH8X · 2025-11-05

**Soundness:** 3
**Presentation:** 3
**Contribution:** 3
**Rating:** 8
**Confidence:** 3

**Summary:**

This paper proposes a principled framework for predicting learning curves of kernel ridge regression (KRR) using only empirical data statistics, specifically the data covariance matrix and the Hermite polynomial decomposition of the target function.
The central theoretical idea is the Hermite Eigenstructure Ansatz (HEA), which posits that for rotation-invariant kernels on approximately Gaussian datasets, the kernel’s eigenfunctions coincide with multivariate Hermite polynomials of the data, and the corresponding eigenvalues factorize as simple monomials in the data covariance eigenvalues.



This work offers a conceptually novel and practically valuable contribution: predicting learning curves from minimal dataset statistics without explicit kernel computation.
It bridges data geometry, kernel methods, and neural tangent kernel theory, showing a concrete mapping from data covariance to performance prediction.
The observed similarity between MLP learning order and HEA predictions further strengthens its relevance.

**Strengths:**

- Introduces the Hermite Eigenstructure Ansatz, a compact analytical surrogate for kernel eigenfunctions and eigenvalues grounded in probabilistic data geometry.

- Combines proofs, ablations, and large-scale experiments (CIFAR-5m, SVHN, ImageNet-32) with excellent agreement between theory and observation.

- The conceptual pipeline links data covariance to feature-space statistics and test error.

- Provides a data-to-performance theory that is both interpretable and predictive.

- Clearly delineates success and failure regimes (e.g., Figures 12–16).

**Weaknesses:**

- No analytical error bounds are provided for the eigenvalue approximation.

- The Gram–Schmidt coefficient estimation step can be computationally heavy for large datasets.

- Minor comment: Dependence on Gaussianity: the HEA requires data to be "Gaussian enough"; it fails for structured or discrete datasets (MNIST, tabular). Extending it beyond this regime would strengthen the framework, but would be very difficult though (but does not undermine the results though).

**Questions:**

- Could the Gaussian assumption be relaxed to a mixture-of-Gaussians or empirical-moment approach to handle non-Gaussian data?


- Can the authors provide quantitative error bounds between predicted and empirical eigenvalues as a function of effective dimension or kernel width?

---

> ### Author Response · Authors · 2025-11-26
> **Author Response**
>
> We thank the reviewer for the feedback on our paper. We are glad that the reviewer saw the novelty in our approach to understand how both kernels and MLPs “see the data.” Here, we address the reviewer’s major comments and questions:
>
> **No analytical error bounds**: We acknowledge the strength of analytical error bounds, and agree they will be worthwhile to obtain. While we hope follow-up work will obtain such bounds, this is not our present focus. Instead, we provided heuristic regimes under which the HEA fails to be a good approximation.
>
> **Computational cost of Gram-Schmidt procedure**: That is correct, although it should be noted that (1) the Gram-Schmidt step is no more computationally expensive than native KRR and (2) we’re not trying to solve KRR in a more time-efficient manner. Nonetheless, we do find one notable way to speed up our KRR fits: we only need to run the G-S procedure once per dataset, irrespective of our choice of kernel or hyperparameters. We have attempted various different methods for analyzing the target function’s coefficients (see the updated App. D.3) that are computationally much faster, however the accuracy of the recovered coefficients noticeably drops. Further details about why we chose to use a G-S procedure can be found in App. D.3 “Third Pass.”
>
> **Non-Gaussianity**: Yes, we agree that pushing to non-Gaussianity would be useful. It would be difficult – with our approach, we’d need some other analytical family of distributions that’s a better model for these non-Gaussian datasets, and we don’t as yet know what that would be. It’s a good idea, though.
>
> **Could a mixture of Gaussians or empirical moment approach handle non-Gaussian data?** Maybe, though as we mention above, it would be difficult since we are not aware of analytical eigenfunctions for arbitrary Gaussian mixtures. We’re unsure of the finer details, though.
>
> **Can the authors provide bounds between predicted and empirical eigenvalues?** While we’re fairly confident that bounds could be proven, especially for Gaussian data, we haven’t done any such analysis.
>
> Thank you again for the review.

---

### Meta-Review · Area_Chair_TU3o · 2026-01-12

**Summary:**

This submission proposes a Hermite ansatz to predict the generalization error of kernel ridge regression on anisotropic data.
The reviews are very much polarized. After reading the manuscript, the area chair believes that this submission is an interesting contribution to the precise asymptotics and universality literature.
However, reviewer Cuso provides two valuable feedback points that should not be overlooked in the revision: (I) since the proposed approximation is uncontrolled (which other reviewers also pointed out), the individual steps of the approximation need to be highlighted — the authors should include the details in the response to reviewer msN8; (II) the two theorems handle somewhat uninteresting regimes where the limiting kernel doesn’t fluctuate — this limitation should be explicitly discussed.

**Reviewer Concerns:**

See above.

**Reviewer Scores:**

Since this submission received polarized scores, the recommendation is based on the area chair’s own reading of the manuscript.

---

### Decision · Program_Chairs · 2026-01-26

Accept (Poster)